# Stability and Generalization Analysis of Decentralized SGD: Sharper Bounds Beyond Lipschitzness and Smoothness

**Shuang Zeng** [1]   **Yunwen Lei** [1]

## Abstract

Decentralized SGD (D-SGD) is a popular optimization method to train large-scale machine learning models. In this paper, we study the generalization behavior of D-SGD for both smooth and nonsmooth problems by leveraging the algorithm stability. For convex and smooth problems, we develop stability bounds involving the training errors to show the benefit of optimization in generalization. This improves the existing results by removing the Lipschitzness assumption and implying fast rates in a low-noise condition. We also develop the first optimal stability-based generalization bounds for D-SGD applied to nonsmooth problems. We further develop optimization error bounds which imply minimax optimal excess risk rates. Our novelty in the analysis consists of an error decomposition to use the co-coercivity of functions as well as the control of a neighboring-consensus error.

## 1. Introduction

Modern machine learning often involves large-scale datasets which contain private information of users. This asks for the development of scalable optimization methods that can preserve the private information (Lian et al., 2017; 2018). An efficient methodology to meet these requirements is to distribute the datasets over several agents, each of which only processes its own local datasets. In this way, the private information of a local dataset is not revealed to other agents. Decentralized stochastic gradient descent (D-SGD) is a representative distributed optimization method where each agent updates its own local model by building stochastic gradient using its own local dataset. These agents are connected via a communication graph which shows the de-

gree of the connection. Then, each local model iteratively conducts a weighted average and a gradient descent.

The impressive success of D-SGD in practice motivates a lot of theoretical studies to understand their behaviour, for which a challenge lies in the control of a consensus error. Most of these studies focus on the convergence analysis to show the behavior on the training datasets (Lian et al., 2017; 2018; Koloskova et al., 2020; Xin et al., 2020). As a comparison, there is far less work on the generalization analysis to understand the performance on testing datasets, which is a central topic in learning theory. Indeed, generalization analysis sheds insights on how different factors contribute to the prediction, which helps the construction of early-stopping rule to avoid overfitting (Hardt et al., 2016).

Richards and Rebeschini (2020) initialized the generalization analysis of D-SGD by leveraging the concept of uniform stability. Several works improve the generalization analysis by considering another stability concept called on-average model stability (Zhu et al., 2022; Le Bars et al., 2024; Taheri and Thrampoulidis, 2023). While these results are interesting to understand the generalization behavior of D-SGD, there is still room for further improvements. For example, these stability analyses often impose restrictive assumptions such as bounded gradient assumption and bounded variance assumption. Furthermore, they fail to imply optimal generalization bounds that can fully capture low-noises conditions for fast rates. In this paper, we aim to improve these discussions by removing restrictive assumptions and getting optimistic risk bounds. Our contributions are summarized as follows.

• We present the generalization and convergence analysis for D-SGD on smooth and convex problems, for both of which we remove the existing Lipschitzness assumptions and get fast rates under a low noise condition. We build on-average model stability bounds, which involve the training errors to show the benefit of optimization in generalization. We present a clear comparison (Table 1) to show the advantage of our stability analysis as compared to existing results.

• We also consider nonsmooth problems with Hölder continuous gradients, for which we present a comprehensive analysis to show the effect of a smoothness parameter and

---

[1]Department of Mathematics, The University of Hong Kong, Hong Kong, China. Correspondence to: Yunwen Lei <leiyw@hku.hk>.

*Proceedings of the 42$^{nd}$ International Conference on Machine Learning*, Vancouver, Canada. PMLR 267, 2025. Copyright 2025 by the author(s).

the spectral gap on stability and generalization. For the specific Lipschitz problems, we present the first optimal generalization bounds for D-SGD.

- A challenge in the stability analysis of D-SGD is the control of a *neighboring consensus error*. Our key observation is that this error can be offset by the co-coercivity of a gradient map, towards which we introduce a new error decomposition.

The paper is organized as follows. Section 2 gives the related work. Section 3 formulates the problem. We present our result for smooth and nonsmooth problems in Section 4 and Section 5, respectively. Conclusions are given in Section 6.

## 2. Related Work

### 2.1. Algorithmic Stability

Algorithmic stability is a fundamental concept in learning theory to measure the sensitivity of an algorithm with respect to a perturbation on the training dataset. Various stability concepts have been introduced in the literature, including the uniform stability (Bousquet and Elisseeff, 2002), on-average stability (Shalev-Shwartz et al., 2010), argument stability (Liu et al., 2017) and on-average model stability (Lei and Ying, 2020). The uniform stability is arguably the strongest stability measure, which can imply high-probability generalization bounds (Bousquet and Elisseeff, 2002; Bousquet et al., 2020; Klochkov and Zhivotovskiy, 2021; Feldman and Vondrak, 2019; Fan and Lei, 2024; Li et al., 2024). The on-average stability has a close connection to the learnability (Shalev-Shwartz et al., 2010). The influential work (Hardt et al., 2016) pioneered the stability analysis of SGD for Lipschitz and smooth problems, which motivates a lot of interesting studies on the stability of stochastic optimization algorithms (Kuzborskij and Lampert, 2018; Neu et al., 2021; Lei and Ying, 2020; Charles and Papailiopoulos, 2018; Bassily et al., 2020; Nikolakakis et al., 2022; Schliserman et al., 2025). For example, the Lipschitzness and smoothness assumption in Hardt et al. (2016) were weaken based on the on-average model stability. While a convexity assumption is often imposed in the stability analysis, recent progress leverages the stability concept to study the generalization behavior of gradient descent methods to train neural networks (Richards and Kuzborskij, 2021; Wang et al., 2025; Taheri et al., 2025; Deora et al., 2024; Zhang et al., 2024a). Algorithmic stability was widely used to study the generalization behavior for various learning tasks such as adversarial training (Xiao et al., 2022; Zhang et al., 2024a), federated learning (Sun et al., 2024; Chen et al., 2024), differential privacy (Bassily et al., 2020; Wang et al., 2024) and meta-learning (Maurer, 2005).

### 2.2. Decentralized SGD

The exploration of decentralized optimization algorithms dates back to the work of Tsitsiklis (1984). Decentralized gradient descent (DGD) was studied by Nedic et al. (2009), where the consensus error was introduced as an important ingredient in the analysis of decentralized algorithms. The stochastic variants were also extensively studied due to their simplicity and effectiveness in addressing large-scale machine learning challenges, mostly focusing on the optimization properties (Sundhar Ram et al., 2010; Xu et al., 2015; Duchi et al., 2011; Lian et al., 2017; Vlaski and Sayed, 2021; Pu and Nedić, 2021; Koloskova et al., 2020; Le Bars et al., 2023; Zhang et al., 2024b). Generalization issues of D-SGD are also drawing increasing attention recently. Richards and Rebeschini (2020) considered a different variant of D-SGD and derived generalization bounds by using uniform stability and Rademacher complexity (Bartlett and Mendelson, 2002). Sun et al. (2021) considered the same D-SGD as ours, and developed uniform stability bounds. On-average model stability of D-SGD was recently studied, where topology-aware (Zhu et al., 2022) and topology-independent (Le Bars et al., 2024) generalization bounds were developed. These stability analyses were extended to other decentralized algorithms such as DGD (Taheri and Thrampoulidis, 2023) and decentralized stochastic gradient descent ascent (Zhu et al., 2024).

## 3. Problem Setup

Let $\mathbb{P}$ be a probability measure defined on a sample space $\mathcal{Z} := \mathcal{X} \times \mathcal{Y}$, where $\mathcal{X}$ is an input space and $\mathcal{Y} \subseteq \mathbb{R}$ is an output space. We consider a decentralized learning setting where the training examples are distributed over $m$ agents. Specifically, for the $k$-th agent, we assume that a training dataset $S_k = \{z_{1,k}, \ldots, z_{n,k}\}$ of size $n$ is independently drawn from $\mathbb{P}$. We collect all these $m$ datasets into $S := \cup_{k=1}^m S_k$, based on which we want to build a prediction function $h : \mathcal{X} \mapsto \mathbb{R}$. We consider parametric learning where the prediction function $h$ is parameterized by $\theta \in \mathcal{W} \subset \mathbb{R}^d$. The performance of $\theta$ on an example $z$ is measured by $\ell(\theta; z)$, where $\ell : \mathcal{W} \times \mathcal{Z} \mapsto \mathbb{R}_+$ is a loss function. Our objective is to find a global model $\theta \in \mathcal{W}$ minimizing the population risk defined by

$$R(\theta) = \frac{1}{m} \sum_{k=1}^m \mathbb{E}_{z \sim \mathcal{Z}}[\ell(\theta, z)],$$

which quantifies the behavior of the model on a testing dataset. Here $\mathbb{E}_z[\cdot]$ denotes the expectation w.r.t. $z$. We also define the empirical risk as

$$R_S(\theta) = \frac{1}{m} \sum_{k=1}^m R_{S_k}(\theta), \text{ where } R_{S_k}(\theta) = \frac{1}{n} \sum_{i=1}^n \ell(\theta, z_{i,k}),$$

which quantifies the behavior of the model on training.

Given $S$, we apply a randomized learning algorithm $A$ to build a model as an approximation of $\theta^* = \arg\min_{\theta \in \mathcal{W}} R(\theta)$. We denote $A(S)$ as the model produced by running $A$ to $S$. We are interested in studying the excess risk $\mathbb{E}[R(A(S)) - R(\theta^*)]$, which can be decomposed as

$$
\begin{aligned}
\mathbb{E}[R(A(S))] - R(\theta^*) &= \mathbb{E}[R(A(S)) - R_S(A(S))] \\
&\quad + \mathbb{E}[R_S(A(S)) - R_S(\theta^*)], \quad (3.1)
\end{aligned}
$$

where we use $\mathbb{E}[R_S(\theta^*)] = R(\theta^*)$. We refer to $\mathbb{E}[R(A(S)) - R_S(A(S))]$ as the generalization gap, which measures the difference between training and testing for the output model. Moreover, we call $\mathbb{E}[R_S(A(S)) - R_S(\theta^*)]$ the optimization error which measures the difference between the training error of the output model versus the best model.

### 3.1. Decentralized SGD

In this paper, we apply D-SGD (Lian et al., 2017) to minimize $R_S(\theta)$. We assume there are $m$ agents communicated via a communication graph represented by a weight matrix $W \in [0,1]^{m \times m}$, where $W_{kl}$ reflects the weight of information sent from agent $l$ to agent $k$. D-SGD initializes all the local models with the same point $\theta^{(0)}$. At the $t$-th round, each agent $k$ independently samples a datapoint $z_{I_k^t, k}$ uniformly from its local dataset $S_k$ (i.e., $I_k^t$ follows the uniform distribution over $[n] := \{1, 2, \ldots, n\}$), based on which it computes a local gradient at its own model $\theta_l^{(t)}$. Then, it receives local models from other agents for a weighted aggregation, after which it moves along the negative direction of local gradient with the step size $\eta_t$. This process can be formulated as

$$
\theta_k^{(t+1)} \leftarrow \sum_{l=1}^{m} W_{kl} \theta_l^{(t)} - \eta_t \nabla \ell(\theta_k^{(t)}; z_{I_k^t, k}). \quad (3.2)
$$

In this paper, we always consider *non-increasing* step sizes, i.e., $\eta_{t+1} \leq \eta_t$. We summarize the procedure of Decentralized SGD in Algorithm 1.

To study D-SGD, we impose an assumption on the mixing matrix, which is standard and has been widely used in the analysis of decentralized learning algorithms (Sun et al., 2021; Deng et al., 2023; Le Bars et al., 2024; Taheri and Thrampoulidis, 2023; Zhu et al., 2022; Richards and Rebeschini, 2020).

**Assumption 3.1** (Mixing matrix). We assume $W$ is doubly stochastic, i.e., $W^\top \mathbf{1} = W\mathbf{1} = \mathbf{1}$ where $\mathbf{1} \in \mathbb{R}^m$ is the vector that contains only ones, and $\max\{|\lambda_2(W)|, |\lambda_m(W)|\} < 1$, where $\lambda_i(W)$ denotes the $i$-th largest eigenvalue of $W$.

Let $\lambda = \max\{\lambda_2^2(W), \lambda_m^2(W)\}$. Then, $1 - \sqrt{\lambda}$ is referred to as the *spectral gap*, which is defined as the difference

between the moduli of the two largest singular values of a matrix associated with a graph. In decentralized learning, the spectral gap measures the connectivity of the underlying communication graph (Zhu et al., 2022). For instance, $\lambda = 0$ for a fully connected graph, where each element of the communication matrix is $\frac{1}{m}$. Conversely, in a ring graph of varying sizes, the value of $\lambda$ increases with the size of the graph and approaches 1 as the size of the ring approaches infinity (Vogels et al., 2022). The inverse of the spectral gap, i.e. $\frac{1}{1-\sqrt{\lambda}}$, can significantly depend on the number of agents $m$. For one of the most standard communication matrices, the inverse of the spectral gap is $O(1)$ in the fully connected graph, $O(m)$ in the grid graph and $O(m^2)$ in the cycle graph (Lu and De Sa, 2023).

---

**Algorithm 1** Decentralized SGD

---

**Input:** Initialize $\forall k, \theta_k^{(0)} = \theta^{(0)} \in \mathbb{R}^d$, iteration number $T$, stepsizes $\{\eta_t\}_{t=0}^{T-1}$, weight matrix $W$.
  **for** $t = 0, \ldots, T-1$ **do**
    **for** each agent $k = 1, \ldots, m$ **do**
      Sample $I_k^t$ by the uniform distribution over $[n]$
      $\theta_k^{(t+1)} \leftarrow \sum_{l=1}^{m} W_{kl} \theta_l^{(t)} - \eta_t \nabla \ell(\theta_k^{(t)}; z_{I_k^t, k})$
    **end for**
  **end for**

---

### 3.2. Stability and Generalization

Algorithmic stability measures how the output model will change if we change a single example in the training dataset (Bousquet and Elisseeff, 2002). Various stability concepts have been proposed in the literature. In this paper, we consider the on-average model stability which can relax the smoothness and the Lipschitzness assumption. Below, we adapt the stability concept in Lei and Ying (2020) to the decentralized learning setting.

**Definition 3.2** (On-average model stability). Let $\epsilon > 0$. Let $S = (S_1, \ldots, S_m)$ with $S_k = \{z_{1,k}, \ldots, z_{n,k}\}$ and $\tilde{S} = (\tilde{S}_1, \ldots, \tilde{S}_m)$ with $\tilde{S}_k = \{\tilde{z}_{1,k}, \ldots, \tilde{z}_{n,k}\}$ be two independent copies such that $z_{i,k} \sim \mathbb{P}$ and $\tilde{z}_{i,k} \sim \mathbb{P}$. For any $i \in [n]$ and $j \in [m]$, denote by $S^{(ij)}$ the dataset formed from $S$ by replacing the $i$-th element of the $j$-th agent's dataset by $\tilde{z}_{i,j}$:

$$
S^{(ij)} = (S_1, \ldots, S_{j-1}, S_j^{(i)}, S_{j-1}, \ldots, S_m),
$$

where $S_j^{(i)} = \{z_{1,j}, \ldots, z_{i-1,j}, \tilde{z}_{i,j}, z_{i+1,j}, \ldots, z_{n,j}\}$. We say an algorithm $A$ is on-average $\epsilon$-model stable if

$$
\mathbb{E}_{S, \tilde{S}, A}\left[ \frac{1}{mn} \sum_{i=1}^{n} \sum_{j=1}^{m} \|A(S) - A(S^{(ij)})\|_2^2 \right] \leq \epsilon^2.
$$

To study the connection between stability and generalization, we need to introduce standard concepts on the Hölder

continuity and Lipscthizness. Smooth loss functions include the logistic loss for classification and the least square loss for regression. Lipschitz loss functions include the logistic loss, hinge loss and Huber loss (Mohri et al., 2018). Examples of loss functions with Hölder continuous gradients include the $q$-norm hinge loss and $q$-th power absolute distance loss with $q \in [1, 2]$ (Steinwart and Christmann, 2008).

**Definition 3.3** (Hölder continuity). Let $\alpha \in [0, 1]$ and $L > 0$. We say a function $g$ has $(\alpha, L)$-Hölder continuous gradients if

$$\|\nabla g(\theta) - \nabla g(\theta')\|_2 \le L \|\theta - \theta'\|_2^\alpha, \quad \forall \theta, \theta'.$$

Especially, when $\alpha = 1$, we say the function $g$ is $L$-smooth.

**Definition 3.4** (Lipschitzness). Let $G > 0$. We say a function $g$ is $G$-Lipschitz continuous if $|g(\theta) - g(\theta')| \le G\|\theta - \theta'\|_2$ for any $\theta, \theta'$.

The following lemma is a direct extension of a result in Lei and Ying (2020) to the decentralized learning setting. We give the proof in Section A for completeness.

**Lemma 3.5.** *Let $S, \tilde{S}$ and $S^{(ij)}$ be constructed as in Definition 3.2, and let $\gamma > 0$. Let $A$ be a randomized algorithm.*

a) *Suppose for any $z$, the function $\mathbf{w} \mapsto \ell(\mathbf{w}; z)$ is non-negative and $L$-smooth. Then*

$$\mathbb{E}_{S,A}[R(A(S)) - R_S(A(S))] \le \frac{L}{\gamma} \mathbb{E}_{S,A}[R_S(A(S))] +$$

$$\frac{L + \gamma}{2mn} \sum_{i=1}^{n} \sum_{j=1}^{m} \mathbb{E}_{S,\tilde{S},A}[\|A(S^{(ij)}) - A(S)\|_2^2].$$

b) *If $\mathbf{w} \mapsto \ell(\mathbf{w}; z)$ is $G$-Lipschitz continuous, then*

$$|\mathbb{E}_{S,A}[R(A(S)) - R_S(A(S))]|$$

$$\le \frac{G}{mn} \sum_{i=1}^{n} \sum_{j=1}^{m} \mathbb{E}_{S,\tilde{S},A}[\|A(S^{(ij)}) - A(S)\|_2].$$

# 4. Convex and Smooth Problems

In this section, we present the generalization analysis in the convex and smooth case.

## 4.1. Stability Analysis

We first present our main result on the stability of D-SGD, which control the stability by the magnitude of gradients encountered in the trajectory. The proof is given in Section B.1. We denote $A \lesssim B$ if there exists a universal constant $C$ such that $A \le CB$. We denote $A \asymp B$ if $A \lesssim B$ and $B \lesssim A$.

**Theorem 4.1** (Stability bound ). *Let Assumption 3.1 hold. Let $S, S^{(ij)}$ be defined in Definition 3.2. Assume the loss*

*function $\ell(\cdot; z)$ is convex and $L$-smooth. Let $\theta_1^{(t)}, \ldots, \theta_m^{(t)}$ be the $t$-th iterates of D-SGD run over $S$, and $\theta_1^{(t,ij)}, \ldots, \theta_m^{(t,ij)}$ be the $t$-th iterates of D-SGD run over $S^{(ij)}$. Denote $\bar{\theta}^{(t)} = \frac{1}{m} \sum_{k=1}^{m} \theta_k^{(t)}$ and $\bar{\theta}^{(t,ij)} = \frac{1}{m} \sum_{k=1}^{m} \theta_k^{(t,ij)}$. Let*

$$A(S) = \bar{\theta}^{(T)} \quad and \quad A(S^{(ij)}) = \bar{\theta}^{(T,ij)}. \quad (4.1)$$

*If*

$$\left(\frac{2(1 + \lambda)L\eta_t}{(1 - \lambda)^2} + 2\right)\eta_t - \frac{1}{L} \le 0, \quad (4.2)$$

*then*

$$\frac{1}{mn} \sum_{i=1}^{n} \sum_{j=1}^{m} \mathbb{E}[\|A(S) - A(S^{(ij)})\|_2^2] \lesssim$$

$$\frac{1}{m^2 n^2} \sum_{i=1}^{n} \sum_{j=1}^{m} \sum_{t=0}^{T-1} \left(\frac{L\eta_t}{(1-\lambda)^2} + \frac{1}{m}\right)\eta_t^2 \mathbb{E}[\|\nabla \ell(\theta_j^{(t)}; z_{i,j})\|_2^2]$$

$$+ \frac{1}{m^3 n^3} \sum_{i=1}^{n} \sum_{j=1}^{m} \left(\sum_{t=0}^{T-1} \eta_t \left(\mathbb{E}[\|\nabla \ell(\theta_j^{(t)}; z_{i,j})\|_2^2]\right)^{\frac{1}{2}}\right)^2.$$

*Remark* 4.2. If we impose a $G$-Lipschitzness condition, Theorem 4.1 implies (we assume $\eta_t = \eta$ for brevity)

$$\frac{1}{mn} \sum_{i=1}^{n} \sum_{j=1}^{m} \mathbb{E}[\|A(S) - A(S^{(ij)})\|_2^2] \lesssim$$

$$\frac{G^2 \eta^2 T}{mn}\left(\frac{L\eta}{(1-\lambda)^2} + \frac{1}{m}\right) + \frac{G^2 \eta^2 T^2}{m^2 n^2}. \quad (4.3)$$

If $\eta \lesssim m(1-\lambda)^2/L$, which is widely used in both the existing convergence analysis (Richards and Rebeschini, 2020) and stability analysis (Taheri and Thrampoulidis, 2023) of decentralized algorithms, then Eq. (4.3) further implies

$$\frac{1}{mn} \sum_{i=1}^{n} \sum_{j=1}^{m} \mathbb{E}[\|A(S) - A(S^{(ij)})\|_2^2] \lesssim \frac{G^2 \eta^2}{mn}\left(\frac{T}{m} + \frac{T^2}{mn}\right). \quad (4.4)$$

If we impose a $G$-Lipschtizness condition in Lei and Ying (2020), then it was shown that the vanilla SGD is on-average $\epsilon$-model stable with $\epsilon^2 \lesssim \frac{G^2 \eta^2}{n}\left(T + \frac{T^2}{n}\right)$. Therefore, our on-average model stability analysis recovers the existing result for the specific single-machine serial case. Eq. (4.4) can be rewritten as $\frac{G^2 \eta^2}{N}\left(\frac{T}{m} + \frac{T^2}{N}\right)$ with $N = mn$, showing that increasing $m$ improves stability even when the total number of samples $N$ is fixed. The underlying reason is that we consider $\ell_2$ on-average model stability, which is the second moment of a random variable and can be decomposed as a bias and a variance term. Since the average operator reduces the variance by a factor of $m$, there is a factor of $1/m$ in our stability bound (Theorem 4.1 gives a bound on the square of the $\ell_2$ on-average model stability). As a comparison, the previous discussions either consider local iterates or the

$\ell_1$ version of stability, which may not show the effect of variance reduction of D-SGD with $m$ agents. Our results also match recent findings in Lei et al. (2023), which show that the variance decreases by a factor of batch size or local machines.

*Remark* 4.3 (Novelty). As compared to the stability analysis for the vanilla SGD, a challenge in the decentralized case is to control the *neighboring consensus errors*: $\sum_{k=1}^{m} \mathbb{E}\big[\big\|\bar{\theta}^{(t)} - \theta_k^{(t)} - \bar{\theta}^{(t,ij)} + \theta_k^{(t,ij)}\big\|_2^2\big]$. Existing works (Sun et al., 2021; Taheri and Thrampoulidis, 2023) decompose this term into two consensus errors and use the estimate for the consensus error as follows

$$
\sum_{t=0}^{T-1} \sum_{k=1}^{m} \mathbb{E}\big[\big\|\bar{\theta}^{(t)} - \theta_k^{(t)} - \bar{\theta}^{(t,ij)} + \theta_k^{(t,ij)}\big\|_2^2\big]
$$

$$
\lesssim \sum_{t=0}^{T-1} \sum_{k=1}^{m} \mathbb{E}\big[\|\bar{\theta}^{(t)} - \theta_k^{(t)}\|_2^2\big] + \sum_{t=0}^{T-1} \sum_{k=1}^{m} \big[\|\bar{\theta}^{(t,ij)} - \theta_k^{(t,ij)}\|_2^2\big]
$$

$$
\lesssim \frac{Lm}{(1-\lambda)^2} \sum_{t=0}^{T-1} \eta_t^2 \mathbb{E}[R_S(\bar{\theta}^t)].
$$

This decomposition ignores the important property that $\theta_k^{(t)}$ and $\theta_k^{(t,ij)}$ are produced based on two neighboring datasets, which should be close. Due to the ignorance of this important property, the existing stability analysis implies somewhat crude bounds (Sun et al., 2021; Taheri and Thrampoulidis, 2023). Instead, we give a bound of *neighboring consensus errors* which can capture the closeness of $\theta_k^{(t)}$ and $\theta_k^{(t,ij)}$ (see in Lemma B.3)

$$
\sum_{t=0}^{T-1} \sum_{k=1}^{m} \mathbb{E}\big[\big\|\bar{\theta}^{(t)} - \theta_k^{(t)} - \bar{\theta}^{(t,ij)} + \theta_k^{(t,ij)}\big\|_2^2\big] \lesssim
$$

$$
\frac{1}{(1-\lambda)^2} \sum_{t=0}^{T-1} \eta_t^2 \sum_{k=1}^{m} \big\|\nabla\ell(\theta_k^{(t)}; Z_{I_k^t,k}) - \nabla\ell(\theta_k^{(t,ij)}; \tilde{Z}_{I_k^t,k}^{(ij)})\big\|_2^2,
$$

where $Z_{I_k^t,k} = z_{I_k^t,k}$, and $\tilde{Z}_{I_k^t,k}^{(ij)} = \tilde{z}_{i,j}$ if ($k = j$ & $I_k^t = i$) and $z_{I_k^t,k}$ otherwise. Our key observation is that the term $\big\|\nabla\ell(\theta_k^{(t)}; Z_{I_k^t,k}) - \nabla\ell(\theta_k^{(t,ij)}; \tilde{Z}_{I_k^t,k}^{(ij)})\big\|_2^2$ can be offset by using the co-coercive property of the convex and smooth loss functions. To this aim, we introduce an error decomposition to get $\langle \theta_k^{(t)} - \theta_k^{(t,ij)}, \nabla\ell(\theta_k^{(t)}; Z_{I_k^t,k}) - \nabla\ell(\theta_k^{(t,ij)}; \tilde{Z}_{I_k^t,k}^{(ij)})\rangle$, which offsets $\big\|\nabla\ell(\theta_k^{(t)}; Z_{I_k^t,k}) - \nabla\ell(\theta_k^{(t,ij)}; \tilde{Z}_{I_k^t,k}^{(ij)})\big\|_2^2$ by

$$
\langle \theta_k^{(t)} - \theta_k^{(t,ij)}, \nabla\ell(\theta_k^{(t)}; Z_{I_k^t,k}) - \nabla\ell(\theta_k^{(t,ij)}; \tilde{Z}_{I_k^t,k}^{(ij)})\rangle
$$

$$
\geq \frac{1}{L}\big\|\nabla\ell(\theta_k^{(t)}; Z_{I_k^t,k}) - \nabla\ell(\theta_k^{(t,ij)}; \tilde{Z}_{I_k^t,k}^{(ij)})\big\|_2^2.
$$

We control the gradient norm with a self-bounding property (Srebro et al., 2010), and derive the following stability

bounds involving training errors, which shows the benefit of optimization in stability. Our analysis requires step sizes to satisfy Eq. (4.2) and $\eta_t \leq \big(\frac{(1-\lambda)^2}{8L^2(1+\lambda)} + \frac{\eta_1^2}{2}\big)^{\frac{1}{2}}$, which are satisfied if $\eta_t \lesssim (1-\lambda)/L$. This is a mild assumption since for D-SGD we often choose $\eta_t \asymp 1/\sqrt{T}$, which vanishes as $T \to \infty$. The proof is given in Section B.1.

**Corollary 4.4.** *Let Assumption 3.1 hold. Let* $\ell(\cdot; z)$ *be convex and L-smooth. Let* $A(S), A(S^{(ij)})$ *be defined in Eq.* (4.1). *If* (4.2) *holds and* $\eta_t \leq \big(\frac{(1-\lambda)^2}{8L^2(1+\lambda)} + \frac{\eta_1^2}{2}\big)^{\frac{1}{2}}$, *then*

$$
\frac{1}{mn} \sum_{i=1}^{n} \sum_{j=1}^{m} \mathbb{E}[\|A(S) - A(S^{(ij)})\|_2^2]
$$

$$
\lesssim \frac{L}{mn} \sum_{t=0}^{T-1} \Big(\frac{L\eta_t}{(1-\lambda)^2} + \frac{1}{m}\Big) \eta_t^2 \mathbb{E}[R_S(\bar{\theta}^{(t)})]
$$

$$
+ \frac{L^2 \sum_{t=0}^{T-1} \eta_t^2}{m^2 n^2} \sum_{t=0}^{T-1} \mathbb{E}[R_S(\bar{\theta}^{(t)})]. \tag{4.5}
$$

*Remark* 4.5 (Comparison). For brevity, we consider constant step sizes. We now compare our stability bounds with existing results. Richards and Rebeschini (2020) considered a different variant of D-SGD, and derived uniform stability bounds of order $\frac{G\eta T}{mn}$, which was extended to the D-SGD in Algorithm 1 (Sun et al., 2021). Under a Gaussian weight difference assumption, a topology-aware stability bound of order $\frac{\eta}{\sqrt{n}}\big(\frac{1}{m}\sum_{k=1}^{m} R_{S_k}(\theta_k^{(T)})\big)^{\frac{1}{2}} + \frac{1}{\sqrt{m}} + \lambda$ was derived (Zhu et al., 2022). Le Bars et al. (2024) developed an improved stability bound as follows

$$
\frac{1}{mn} \sum_{i=1}^{n} \sum_{j=1}^{m} \mathbb{E}[\|A(S) - A(S^{(ij)})\|_2] \lesssim \frac{\sigma\eta T + GT\eta C_W}{mn}
$$

$$
+ \frac{\sqrt{T}\eta}{mn}\Big(\frac{1}{m}\sum_{k=1}^{m} \mathbb{E}[R_{S_k}(\theta_k^{(0)}) - R_{S_k}^*]\Big)^{\frac{1}{2}}, \tag{4.6}
$$

where $R_{S_j}^* = \min_\theta R_{S_j}(\theta), C_W = \sum_{t=0}^{T-1} \|W^t - W^{t+1}\|_2$ and a bounded variance assumption was imposed

$$
\sup_\theta \max_{k\in[m]} \frac{1}{n}\sum_{i=1}^{n} \|\nabla\ell(\theta; Z_{i,k} - \nabla R_{S_k}(\theta))\|_2^2 \leq \sigma^2. \tag{4.7}
$$

Taheri and Thrampoulidis (2023) showed that DGD is on-average $\epsilon$-model stable with (detailed calculations are given in section B.2)

$$
\epsilon \lesssim \Big(\frac{\eta\sqrt{LT}}{mn} + \frac{L^{\frac{3}{2}}\eta^2\sqrt{T}}{1-\lambda}\Big)\Big(\sum_{t=0}^{T-1} \mathbb{E}[R_S(\bar{\theta}^{(t)})]\Big)^{\frac{1}{2}}. \tag{4.8}
$$

Under a mild assumption that $n \lesssim TL$, our analysis shows that D-SGD is on-average $\epsilon$-model stable with

$$
\epsilon \lesssim \Big(\frac{L\eta\sqrt{T}}{mn} + \frac{L\eta^{\frac{3}{2}}}{\sqrt{mn}(1-\lambda)}\Big)\Big(\sum_{t=0}^{T-1} \mathbb{E}[R_S(\bar{\theta}^{(t)})]\Big)^{\frac{1}{2}}. \tag{4.9}
$$

Table 1: Stability Bounds for Convex and Smooth Problems. $C, S, L$ denote the assumptions of convexity, $L$-smoothness and $G$-Lipschitzness respectively. Here, A in the column "Type" means the variant A of D-SGD, i.e., $\theta_k^{(t+1)} \leftarrow \sum_{l=1}^{m} W_{kl}(\theta_l^{(t)} - \eta_t \nabla \ell(\theta_l^{(t)}; z_{I_l^t, l}))$, B means the variant B of D-SGD, i.e., $\theta_k^{(t+1)} \leftarrow \sum_{l=1}^{m} W_{kl} \theta_l^{(t)} - \eta_t \nabla \ell(\theta_k^{(t)}; z_{I_k^t, k})$. For the bound in Le Bars et al. (2024), we use the notation $R_{S_j}^* = \min_\theta R_{S_j}(\theta), C_W = \sum_{t=0}^{T-1} \|W^t - W^{t+1}\|_2$ and impose a bounded variance assumption in Eq. (4.7).

| Type | Reference | C | S | L | Stability Bounds |
|------|-----------|---|---|---|------------------|
| SGD/A | Richards and Rebeschini (2020) | ✓ | ✓ | ✓ | $\frac{G\eta T}{mn}$ |
| SGD/B | Sun et al. (2021) | ✓ | ✓ | ✓ | $\frac{G\eta T}{mn} + \frac{G\eta T}{1-\lambda}$ |
| SGD/B | Zhu et al. (2022) | ✓ | ✓ | ✗ | $\frac{\eta}{\sqrt{n}} \sqrt{\frac{1}{m} \sum_{k=1}^m R_{S_k}(\theta_k^{(T)})} + \frac{1}{\sqrt{m}} + \lambda$ |
| SGD/B | Le Bars et al. (2024) | ✓ | ✓ | ✓ | $\frac{\sqrt{T\eta}}{mn} \sqrt{\frac{1}{m} \sum_{k=1}^m \mathbb{E}[R_{S_k}(\theta_k^{(0)}) - R_{S_k}^*]} + \frac{\sigma\eta T + GT\eta C_W}{mn}$ |
| GD/B | Taheri and Thrampoulidis (2023) | ✓ | ✓ | ✗ | $\left(\frac{\eta\sqrt{LT}}{mn} + \frac{L^{\frac{3}{2}}\eta^2\sqrt{T}}{1-\lambda}\right) \sqrt{\sum_{t=0}^{T-1} \mathbb{E}[R_S(\bar{\theta}^{(t)})]}$ (Section B.2) |
| SGD/B | Ours | ✓ | ✓ | ✗ | $\left(\frac{L\eta\sqrt{T}}{mn} + \sqrt{\frac{L\eta^2}{mn}\left(\frac{L\eta}{(1-\lambda)^2} + \frac{1}{m}\right)}\right) \sqrt{\sum_{t=0}^{T-1} \mathbb{E}[R_S(\bar{\theta}^{(t)})]}$ |

Our analysis improves the stability analysis in Le Bars et al. (2024) by removing the bounded variance assumption and the Lipschitzness assumption. While Eq. (4.6) also involves training errors, the dominant term in the bound is $\frac{\sigma\eta T + GT\eta C_W}{mn}$ if $\sigma$ is not sufficiently small. As a comparison, our stability analysis can imply fast rates in a low-noise condition. Our analysis also outperforms that in Taheri and Thrampoulidis (2023), as it involves a factor $\frac{\eta^2\sqrt{T}}{1-\lambda}$ (we ignore $L$ for brevity), which is replaced by $\frac{\eta^{\frac{3}{2}}}{\sqrt{mn}(1-\lambda)} + \frac{\eta}{m\sqrt{n}}$ in Eq. (4.9). If $\eta \gtrsim \frac{1}{mnT}$ and $\eta \gtrsim \frac{1-\lambda}{\sqrt{T}nm}$, then our stability bound is better since

$$\frac{\eta^2\sqrt{T}}{1-\lambda} \gtrsim \frac{\eta^{\frac{3}{2}}}{\sqrt{mn}(1-\lambda)} + \frac{\eta}{m\sqrt{n}}. \quad (4.10)$$

Our analysis suggests $\eta \asymp 1/\sqrt{mn}$ and $T \asymp mn$ in Remark 4.9, and in this case the left-hand side of Eq. (4.10) is significantly larger than the right-hand side. Then, Eq. (4.9) is sharper than Eq. (4.8). Finally, it should be emphasized that we consider D-SGD, while Taheri and Thrampoulidis (2023) considered DGD. We summarize the comparison in Table 1. According to the table, it is also clear that our stability analysis also improves Sun et al. (2021) and Zhu et al. (2022) by removing the term $G\eta T/(1-\lambda)$ and $\lambda$, respectively, which do not converge to 0 even if $nm$ goes to infinity.

## 4.2. Convergence Analysis

Theorem 4.6 to be proved in Section B.3 gives convergence rates of D-SGD for smooth and convex problems. It should be mentioned that our assumption on step size in Eq (4.11) is consistent with Eq. (4.2) and that in Corollary 4.4.

**Theorem 4.6** (Optimization error)**.** *Assume $\ell(\cdot; z)$ is convex and $L$-smooth. Define $\bar{\theta}^{(t)} = \frac{1}{m}\sum_{k=1}^m \theta_k^{(t)}$. If Assumption 3.1 holds and $\eta_t = \eta$ with*

$$\frac{8L^2(1+\lambda)\eta^2}{(1-\lambda)^2} + 2L\eta \leq \frac{1}{2}, \quad (4.11)$$

*then*

$$\frac{1}{T}\sum_{t=1}^T \mathbb{E}[R_S(\bar{\theta}^{(t)}) - R_S(\theta^*)] \leq \frac{\mathbb{E}[\|\bar{\theta}^{(1)} - \theta^*\|_2^2]}{\eta T} +$$

$$\left(\frac{16L^2(1+\lambda)\eta^2}{(1-\lambda)^2} + 4L\eta\right)\mathbb{E}[R_S(\theta^*)]. \quad (4.12)$$

*Remark* 4.7 (Comparison)**.** For convex, $L$-smooth and $G$-Lipschitz problems, it was shown (Richards and Rebeschini, 2020; Sun et al., 2021)

$$\frac{1}{T}\sum_{t=1}^T \mathbb{E}[R_S(\bar{\theta}^{(t)}) - R_S(\theta^*)] \lesssim \frac{\|\theta^*\|_2^2}{\eta T} + \frac{GL\eta}{1-\lambda} + \frac{G^2\eta^2}{(1-\lambda)^2}. \quad (4.13)$$

A key difference between our result and Eq. (4.13) is that our optimization error bound involves the empirical risk of $\theta^*$. Therefore, it implies fast rates in an interpolation setting. In more detail, by choosing $\eta \asymp 1/\sqrt{T}$, Theorem 4.6 implies convergence rates of order $O(1/\sqrt{T})$ in a general case. In the interpolation setting with a vanishing $R_S(\theta^*)$, Theorem 4.6 implies fast convergence rates of order $O(1/T)$ by choosing $\eta \asymp 1$. As a comparison, Eq. (4.13) requires an additional Lipschitzness assumption and can only imply convergence rates of order $O(1/\sqrt{T})$. Convergence of DGD was also recently studied in Taheri and Thrampoulidis (2023). Their analysis considers learning with separable data. As a comparison, our convergence analysis covers a more general case and focus on D-SGD, which is more computationally-efficient than DGD.

### 4.3. Excess Risk Analysis

We combine the stability and convergence analysis together, and derive the following excess risk bounds for D-SGD on convex and smooth problems. Our bound is optimistic in the sense of involving the risk of the best model (Srebro et al., 2010). The proof is given in Section B.4.

**Theorem 4.8.** *Let assumptions in Theorem 4.1 and Theorem 4.6 hold. Then for any $\gamma > 0$ we have*

$$
\mathbb{E}[R(A(S))] - R(\theta^*) \lesssim \Big(R(\theta^*) + \frac{\|\theta^*\|_2^2}{\eta T}\Big) \times
$$
$$
\Big(\frac{L}{\gamma} + \frac{T(L+\gamma)L\big(\frac{L\eta}{(1-\lambda)^2} + \frac{1}{m}\big)\eta^2}{mn} + \frac{(L+\gamma)L^2\eta^2 T^2}{m^2 n^2}\Big)
$$
$$
+ \frac{\|\theta^*\|_2^2}{\eta T} + \Big(\frac{L^2\eta^2}{(1-\lambda)^2} + L\eta\Big)\mathbb{E}[R(\theta^*)].
$$

*Remark* 4.9 (Illustration). In the standard setting with $R(\theta^*) + \|\theta^*\|_2^2/(\eta T) \lesssim 1$, Theorem 4.8 shows that

$$
\epsilon_{\text{risk}} := \mathbb{E}[R(A(S))] - R(\theta^*) \lesssim \frac{\|\theta^*\|_2^2}{\eta T} + \frac{L^2\eta^2}{(1-\lambda)^2} + L\eta
$$
$$
+ \frac{L}{\gamma} + \frac{T(L+\gamma)L\big(\frac{L\eta}{(1-\lambda)^2} + \frac{1}{m}\big)\eta^2}{mn} + \frac{(L+\gamma)L^2\eta^2 T^2}{m^2 n^2}.
$$

We choose the optimal $\gamma$ to minimize this bound and get

$$
\epsilon_{\text{risk}} \lesssim \frac{\|\theta^*\|_2^2}{\eta T} + \frac{L^2\eta^2}{(1-\lambda)^2} + L\eta + \frac{L\sqrt{T}\eta}{\sqrt{mn}}\Big(\frac{L\eta}{(1-\lambda)^2} + \frac{1}{m}\Big)^{\frac{1}{2}} + \frac{L^{\frac{3}{2}}\eta T}{mn}.
$$

Since $\frac{L\sqrt{T}\eta}{\sqrt{mn}}\big(\frac{L\eta}{(1-\lambda)^2}\big)^{\frac{1}{2}} \lesssim \frac{L^2\eta^2}{(1-\lambda)^2} + \frac{L^{\frac{3}{2}}\eta T}{mn}$ and assuming $n \lesssim TL$, we further get

$$
\epsilon_{\text{risk}} \lesssim \frac{\|\theta^*\|_2^2}{\eta T} + \frac{L^2\eta^2}{(1-\lambda)^2} + L\eta + \frac{L^{\frac{3}{2}}\eta T}{mn}.
$$

Taking $\eta \asymp \frac{\|\theta^*\|_2}{\sqrt{T}(L + L^{3/2}T/(mn))^{\frac{1}{2}}}$[1] gives

$$
\epsilon_{\text{risk}} \lesssim \frac{L\|\theta^*\|_2^2}{T(1-\lambda)^2\big(1 + \sqrt{L}T/(mn)\big)} + \|\theta^*\|_2\Big(\frac{L}{T} + \frac{L^{\frac{3}{2}}}{mn}\Big)^{\frac{1}{2}}.
$$

Setting $T \asymp mn/\sqrt{L}$, we get

$$
\epsilon_{\text{risk}} \lesssim \frac{L^{\frac{3}{2}}\|\theta^*\|_2^2}{mn(1-\lambda)^2} + \frac{L^{\frac{3}{4}}\|\theta^*\|_2}{\sqrt{mn}}.
$$

The minimax statistical error for learning with a convex and smooth function is $O(1/\sqrt{n})$ (e.g., Theorem 7 in Chen

---

[1]For simplicity, we assume $T$ is large so that such $\eta$ satisfy the condition $\eta \lesssim (1-\lambda)/L$ in Eqs. (4.2) and (4.11). For example, a suggested choice is $T \asymp mn/\sqrt{L}$. Our assumption on step sizes holds if $mn$ is large.

et al. (2018)), where $n$ is the sample size. Since our training set has $mn$ examples in total, our stability analysis implies minimax optimal excess risk bounds of the order $1/\sqrt{mn}$.

We now consider a low-noise setting where $R(\theta^*) + \|\theta^*\|_2^2/(\eta T) \lesssim 1/(\eta T)$. Theorem 4.8 with $\gamma = L$ shows

$$
\epsilon_{\text{risk}} \lesssim \frac{1 + L^2\eta^2(1-\lambda)^{-2} + L\eta}{\eta T} +
$$
$$
\frac{1}{\eta T}\Big(1 + \frac{TL^2\big(\frac{L\eta}{(1-\lambda)^2} + \frac{1}{m}\big)\eta^2}{mn} + \frac{L^3\eta^2 T^2}{m^2 n^2}\Big),
$$

which can be simplified as

$$
\epsilon_{\text{risk}} \lesssim \frac{L^2\eta}{T(1-\lambda)^2} + \frac{L}{T} + \frac{1}{\eta T} + \frac{L^3\eta^2}{mn(1-\lambda)^2} + \frac{L^3\eta T}{m^2 n^2}.
$$

We choose $\eta \asymp (1-\lambda)/L$ to meet our assumption on step sizes, and get

$$
\epsilon_{\text{risk}} \lesssim \frac{L}{T(1-\lambda)} + \frac{L}{mn} + \frac{L^2(1-\lambda)T}{m^2 n^2}.
$$

We choose $T \asymp \frac{mn}{\sqrt{L}(1-\lambda)}$ and get $\epsilon_{\text{risk}} \lesssim \frac{L^{\frac{3}{2}}}{mn}$. That is, we get fast rates of order $1/(mn)$ independent of the spectral gap in a low-noise case.

*Remark* 4.10 (Comparison). Excess risk bounds of order $1/\sqrt{mn}$ were also developed in Richards and Rebeschini (2020) for a different variant of D-SGD, where the model aggregation is performed after local update. The work (Sun et al., 2021) considered our D-SGD and developed excess risk bounds of order $\frac{1}{\eta T} + \frac{G^2\eta T}{mn} + \frac{G^2\eta T}{1-\lambda}$, which, however, will not converge due to the term $\frac{G^2\eta T}{1-\lambda}$. The recent work (Le Bars et al., 2024) considered the generalization gap without discussions on optimization error. Furthermore, their generalization bound involves $\frac{\sqrt{T}\eta}{mn}\big(\frac{1}{m}\sum_{k=1}^m \mathbb{E}[R_{S_k}(\theta_k^{(0)}) - R_{S_k}^*]\big)^{\frac{1}{2}}$. It is not clear how to control this term by optimization theory. Therefore, their analysis fails to imply fast rates of order $1/(mn)$ even combined with our optimization error bounds in a low-noise setting. As a comparison, our stability bound involves $\big(\sum_{t=0}^{T-1}\mathbb{E}[R_S(\bar{\theta}^{(t)})]\big)^{\frac{1}{2}}$, which can be controlled by optimization theory and implies the first fast rates of order $O(1/(mn))$ under a low-noise condition. Moreover, these works (Richards and Rebeschini, 2020; Sun et al., 2021; Le Bars et al., 2024) require a Lipschitzness assumption, which is removed in our analysis.

## 5. Convex and Nonsmooth Problems

In this section, we study the stability and generalization of D-SGD applied to convex and nonsmooth problems. Instead, we impose a Hölder continuity assumption on the gradients.

## 5.1. Stability Analysis

We first present a stability bound for D-SGD applied to non-smooth problems. Our bound incorporates the magnitudes of gradients encountered in the learning process, and shows the benefit of optimization in improving generalization. The proof is given in Section C.1.

**Theorem 5.1** (Stability bound). *Let Assumption 3.1 hold and $b = \frac{2(1+\lambda)L}{(1-\lambda)^2} + 2L$. Let $A(S)$ and $A(S^{(ij)})$ be defined in Eq. (4.1). If $\ell(\cdot;z)$ is convex and has $(\alpha, L)$-Hölder continuous gradients with $\alpha \in [0,1)$, then*

$$\frac{1}{mn} \sum_{i=1}^{n} \sum_{j=1}^{m} \mathbb{E}[\|A(S) - A(S^{(ij)})\|_2^2]$$

$$\leq \frac{8}{m^2 n^2} \sum_{i=1}^{n} \sum_{j=1}^{m} \sum_{t=0}^{T-1} \left(\frac{1}{L} + \frac{2}{m}\right) \eta_t^2 \mathbb{E}[\|\nabla \ell(\theta_j^{(t)}; z_{i,j})\|_2^2]$$

$$+ \frac{16}{m^3 n^3} \sum_{i=1}^{n} \sum_{j=1}^{m} \left(\sum_{t=0}^{T-1} \eta_t \left(\mathbb{E}\left[\|\nabla \ell(\theta_j^{(t)}; z_{i,j})\|_2^2\right]\right)^{\frac{1}{2}}\right)^2$$

$$+ \frac{4L(1-\alpha)}{1+\alpha} \sum_{t=0}^{T-1} b^{\frac{1+\alpha}{1-\alpha}} \eta_t^{\frac{2}{1-\alpha}}. \tag{5.1}$$

For Lipschitz continuous problems, we get the following simplified stability bounds to be proved in Section C.1.

**Corollary 5.2** (Stability bound for Lipschitz problems). *Let Assumption 3.1 hold. Let $A(S)$ and $A(S^{(ij)})$ be defined in (4.1). If $\ell(\cdot;z)$ is convex and $G$-Lipschitz continuous, then*

$$\frac{1}{mn} \sum_{i=1}^{n} \sum_{j=1}^{m} \mathbb{E}[\|A(S) - A(S^{(ij)})\|_2^2]$$

$$\lesssim \frac{G^2}{m^2 n^2} \left(\sum_{t=0}^{T-1} \eta_t\right)^2 + \frac{G^2}{(1-\lambda)^2} \sum_{t=0}^{T-1} \eta_t^2. \tag{5.2}$$

*Remark* 5.3. For convex and Lipschitz problems, it was shown that the vanilla SGD is on-average $\epsilon$-model stable with $\epsilon^2 \lesssim (T + T^2/n^2) \eta^2 G^2$. Our analysis recovers their bound in the single machine case.

## 5.2. Convergence Analysis

We now study the convergence for nonsmooth problems. Our optimization error bound involves the consensus errors $\|\Theta^{(t)} - \bar{\Theta}^{(t)}\|_F$, which reflect the variation of the D-SGD iterates around their agent center. The idea of using consensus errors to study the convergence of D-SGD dates back to Nedic et al. (2009). The proof is given in Section C.2.

**Lemma 5.4** (Optimization error). *Let Assumption 3.1 hold. Assume $\ell(\cdot;z)$ is convex and has $(\alpha, L)$-Hölder continuous gradients with $\alpha \in [0,1)$. Define $\bar{\theta}^{(t)} = \frac{1}{m} \sum_{k=1}^{m} \theta_k^{(t)}$.*

*Then,*

$$\frac{1}{T} \sum_{t=1}^{T} \mathbb{E}[R_S(\bar{\theta}^{(t)}) - R_S(\theta)] \leq \frac{c_{\alpha,1}^2 \eta}{T} \sum_{t=1}^{T} \left(\mathbb{E}[R_S(\bar{\theta}^{(t)})]\right)^{\frac{2\alpha}{\alpha+1}}$$

$$+ \frac{\mathbb{E}[\|\bar{\theta}^{(1)} - \theta\|_2^2]}{2\eta T} + \frac{L^2 \eta}{m^\alpha T} \sum_{t=1}^{T} \mathbb{E}[\|\Theta^{(t)} - \bar{\Theta}^{(t)}\|_F^{2\alpha}]$$

$$+ \frac{L}{(\alpha+1)mT} \sum_{t=1}^{T} \mathbb{E}[\|\Theta^{(t)} - \bar{\Theta}^{(t)}\|_F^{\alpha+1}]. \tag{5.3}$$

We can control the above consensus errors and derive the explicit convergence rates. For simplicity, we only consider the Lipschitz case. By choosing $\eta \asymp 1/\sqrt{T}$, Theorem 5.5 implies convergence rates of order $O(1/\sqrt{T})$, which matches that of vanilla SGD (Bottou et al., 2018).

**Theorem 5.5** (Optimization Error for Lipschitz problems). *Let Assumption 3.1 hold. Assume $\ell(\cdot;z)$ is convex and $G$-Lipschitz continuous. Define $\bar{\theta}^{(t)} = \frac{1}{m} \sum_{k=1}^{m} \theta_k^{(t)} \in \mathbb{R}^d$. If we choose a constant step size $\eta$ and $\beta > 0$ such that $(1+\beta)^{\frac{1}{2}} \lambda^{\frac{1}{2}} < 1$, then*

$$\frac{1}{T} \sum_{t=1}^{T} \mathbb{E}[R_S(\bar{\theta}^{(t)}) - R_S(\theta^*)] \leq \frac{\mathbb{E}[\|\bar{\theta}^{(1)} - \theta^*\|_2^2]}{2\eta T} + 13 G^2 \eta$$

$$+ \frac{10\sqrt{2}(1+1/\beta)^{\frac{1}{2}} G^2 \eta}{(1-(1+\beta)^{\frac{1}{2}} \lambda^{\frac{1}{2}})\sqrt{m}}. \tag{5.4}$$

## 5.3. Excess Risk Analysis

In this subsection, we combine the above stability and convergence analysis together to derive excess risk bounds for D-SGD. For simplicity, we only consider the Lipschitz case. The proof is given in Section C.3.

**Theorem 5.6** (Excess Risk for Lipschitz problems). *Let assumptions in Corollary 5.2 and Theorem 5.5 hold. Then*

$$\mathbb{E}[R(A(S))] - R(\theta^*) \lesssim \frac{G^2 \eta T}{mn} + \frac{G^2 \eta T^{\frac{1}{2}}}{1-\lambda} + \frac{\|\theta^*\|_2^2}{\eta T} + \frac{G^2 \eta}{(1-\lambda)^{\frac{3}{2}} m^{\frac{1}{2}}}.$$

*Remark* 5.7. We choose

$$\eta \asymp \frac{\|\theta^*\|_2}{G\sqrt{T}} \left(\frac{T}{mn} + \frac{\sqrt{T}}{1-\lambda} + \frac{1}{(1-\lambda)^{\frac{3}{2}} \sqrt{m}}\right)^{-\frac{1}{2}}$$

and derive

$$\mathbb{E}[R(A(S))] - R(\theta^*) \lesssim \frac{G\|\theta^*\|_2}{\sqrt{T}} \left(\frac{T}{mn} + \frac{\sqrt{T}}{1-\lambda} + \frac{1}{(1-\lambda)^{\frac{3}{2}} m^{\frac{1}{2}}}\right)^{\frac{1}{2}}.$$

We can choose $T \asymp (mn)^2/(1-\lambda)^2$ to get

$$\mathbb{E}[R(A(S))] - R(\theta^*) \lesssim \frac{G\|\theta^*\|_2}{\sqrt{mn}}.$$

Excess risk bounds were also developed for D-SGD applied to convex and Lipschitz problems based on a uniform convergence approach (Richards and Rebeschini, 2020), where it was assumed that ($\sigma_i$ are Rademacher random variables, i.e., taking values in $\{-1, +1\}$ with the same probability)

$$\mathbb{E}_{\sigma_i, i \in [n]} \Big[ \sup_\theta \frac{1}{mn} \sum_{i=1}^{mn} \sigma_i \ell(\theta; \mathbf{z}_i) \Big] \lesssim \frac{1}{\sqrt{mn}}. \qquad (5.5)$$

Under this assumption, D-SGD with appropriate parameters was shown to satisfy $\mathbb{E}[R(A(S))] - R(\theta^*) \lesssim 1/(mn)^{\frac{1}{3}}$ (Richards and Rebeschini, 2020). As a comparison, our stability analysis removes the assumption in Eq. (5.5) and implies minimax optimal bounds of order $O(1/\sqrt{mn})$ (Chen et al., 2018).

## 6. Conclusions

We present a comprehensive stability and generalization analysis of D-SGD for convex problems, covering both the smooth and nonsmooth setting. Our stability bounds involve the training errors of the D-SGD iterations, which establish a connection between generalization and optimization. We also get convergence rates that can interpolate between the slow rate $O(1/\sqrt{T})$ and the fast rate $O(1/T)$, depending on whether there exist models with small training errors. We combine the generalization and convergence analysis to develop optimistic excess risk bounds that can be of order $1/(mn)$ in a low-noise setting. We also give the first excess risk bounds of order $O(1/\sqrt{mn})$ for D-SGD applied to nonsmooth problems. Our studies remove several assumptions in the literature such as the Lipschitzness condition and the bounded variance assumption.

There remain several interesting problems for further studies. First, we only consider convex problems in this paper. It is interesting to investigate whether our stability analysis can be extended to a nonconvex setting. For example, it is interesting to study the stability and generalization analysis of decentralized algorithms for training overparameterized neural networks, where we can exploit some weak-convexity (Richards and Kuzborskij, 2021; Wang et al., 2025) and self-bounding weak-convexity (Taheri et al., 2025; Deora et al., 2024) to develop stability bounds. Second, we only develop excess risk bounds in expectation. It is interesting to develop high-probability bounds to understand the robustness of D-SGD.

## Acknowledgement

We are grateful to the area chair and reviewers for their constructive comments and suggestions. The work of Yunwen Lei is partially supported by the Research Grants Council of Hong Kong [Project Nos. 22303723, 17302624].

## Impact Statement

This paper presents work whose goal is to advance the field of Machine Learning. There are many potential societal consequences of our work, none which we feel must be specifically highlighted here.

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

# A. Proofs on Stability and Generalization

In this section, we present the proof of Lemma 3.5 on the connection between on-average model stability and generalization for decentralized methods. The following lemma (Srebro et al., 2010) builds the self-bounding property for a smooth and nonnegative function, meaning that the gradient can be bounded by function values. The self-bounding property has been widely used to improve the theoretical guarantee of various learning algorithms (Srebro et al., 2010; Lei and Ying, 2020; Taheri et al., 2025; Zhao et al., 2024).

**Lemma A.1** (Self-bounding property). *If $g$ is $L$-smooth and nonnegative, then for any $\theta$ we have $\|\nabla g(\theta)\|_2^2 \leq 2Lg(\theta)$.*

*Proof of Lemma 3.5.* Similar to the proof of Theorem 2 in Lei and Ying (2020), due to the symmetry, we know

$$\mathbb{E}_{S,A}\big[R(A(S)) - R_S(A(S))\big] = \mathbb{E}_{S,\widetilde{S},A}\Big[\frac{1}{mn}\sum_{i=1}^{n}\sum_{j=1}^{m}(R(A(S^{(ij)})) - R_S(A(S)))\Big]$$

$$= \mathbb{E}_{S,\widetilde{S},A}\Big[\frac{1}{mn}\sum_{i=1}^{n}\sum_{j=1}^{m}(\ell(A(S^{(ij)}); z_{i,j}) - \ell(A(S); z_{i,j}))\Big], \tag{A.1}$$

where the last identity holds since $A(S^{(ij)})$ is independent of $z_{i,j}$. The stated bound in part b) then follows directly from the Lipschitzness condition.

We now prove part a). According to the $L$-smoothness of $\mathbf{w} \mapsto \ell(\mathbf{w}; z)$, we know (Nesterov, 2015)

$$\ell(\mathbf{w}; z) \leq \ell(\tilde{\mathbf{w}}; z) + \langle \mathbf{w} - \tilde{\mathbf{w}}, \nabla\ell(\tilde{\mathbf{w}}; z)\rangle + \frac{L\|\mathbf{w} - \tilde{\mathbf{w}}\|_2^2}{2},$$

which, together with Eq. (A.1), implies that

$$\mathbb{E}_{S,A}[R(A(S)) - R_S(A(S))]$$
$$\leq \frac{1}{mn}\sum_{i=1}^{n}\sum_{j=1}^{m}\mathbb{E}_{S,\widetilde{S},A}\Big[\langle A(S^{(ij)}) - A(S), \nabla\ell(A(S); z_{i,j})\rangle + \frac{L}{2}\|A(S^{(ij)}) - A(S)\|_2^2\Big]. \tag{A.2}$$

According to the Schwarz's inequality we know

$$\langle A(S^{(i)}) - A(S), \nabla\ell(A(S); z_{i,j})\rangle \leq \|A(S^{(ij)}) - A(S)\|_2\|\nabla\ell(A(S); z_{i,j})\|_2$$

$$\leq \frac{\gamma}{2}\|A(S^{(ij)}) - A(S)\|_2^2 + \frac{1}{2\gamma}\|\nabla\ell(A(S); z_{i,j})\|_2^2$$

$$\leq \frac{\gamma}{2}\|A(S^{(ij)}) - A(S)\|_2^2 + \frac{L}{\gamma}\ell(A(S); z_{i,j}), \tag{A.3}$$

where the last inequality is due to the self-bounding property of smooth functions (Lemma A.1). Combining Eq. (A.2) and Eq. (A.3), we derive

$$\mathbb{E}_{S,A}[R(A(S)) - R_S(A(S))]$$
$$\leq \frac{L+\gamma}{2mn}\sum_{i=1}^{n}\sum_{j=1}^{m}\mathbb{E}_{S,\widetilde{S},A}\Big[\|A(S^{(ij)}) - A(S)\|_2^2\Big] + \frac{L}{\gamma mn}\sum_{i=1}^{n}\sum_{j=1}^{m}\mathbb{E}_{S,A}\Big[\ell(A(S); z_{i,j})\Big]$$

$$= \frac{L+\gamma}{2mn}\sum_{i=1}^{n}\sum_{j=1}^{m}\mathbb{E}_{S,\widetilde{S},A}\Big[\|A(S^{(ij)}) - A(S)\|_2^2\Big] + \frac{L}{\gamma}\mathbb{E}_{S,A}[R_S(A(S))],$$

where we use the fact that $\frac{1}{mn}\sum_{i=1}^{n}\sum_{j=1}^{m}\ell(A(S); z_{i,j}) = R_S(A(S))$. $\qquad\square$

# B. Proofs for Convex and Smooth Problems

## B.1. Proofs on Stability Analysis

Before our stability analysis, we first introduce some useful notations which will be used throughout the paper. Let $\theta_1^{(t)}, \ldots, \theta_m^{(t)}$ be the $t$-th iterates of D-SGD run over $S$. Let $\theta_1^{(t,ij)}, \ldots, \theta_m^{(t,ij)}$ be the $t$-th iterates of D-SGD run over $(S^{(ij)})$.

Denote $\bar{\theta}^{(t)} = \frac{1}{m}\sum_{k=1}^{m}\theta_k^{(t)}$, $\bar{\theta}^{(t,ij)} = \frac{1}{m}\sum_{k=1}^{m}\theta_k^{(t,ij)}$. For any $k \in [m]$, denote $Z_{I_k^t,k} = z_{I_k^t,k}$ and

$$\tilde{Z}_{I_k^t,k}^{(ij)} = \begin{cases} \tilde{z}_{i,j}, & \text{if } k = j \text{ and } I_k^t = i, \\ z_{I_k^t,k}, & \text{else.} \end{cases}$$

That is, $\tilde{Z}_{I_k^t,k}^{(ij)}$ is the example selected by the $k$-th local machine at the $t$-th iteration when applied D-SGD to $S^{(ij)}$. We collect all the weights in matrices

$$\begin{aligned} \Theta^{(t)} &= (\theta_1^{(t)}, \ldots, \theta_m^{(t)})^\top \in \mathbb{R}^{m \times d}, \quad \Theta^{(t,ij)} = (\theta_1^{(t,ij)}, \ldots, \theta_m^{(t,ij)})^\top \in \mathbb{R}^{m \times d}, \\ \bar{\Theta}^{(t)} &= (\bar{\theta}^{(t)}, \ldots, \bar{\theta}^{(t)})^\top \in \mathbb{R}^{m \times d}, \quad \bar{\theta}^{(t,ij)} = (\bar{\theta}^{(t,ij)}, \ldots, \bar{\theta}^{(t,ij)})^\top \in \mathbb{R}^{m \times d}, \end{aligned} \tag{B.1}$$

where $\bar{\theta} = \frac{1}{m}\sum_{j=1}^{m}\theta_j$ is an average over $m$ agents. Furthermore, denote

$$\nabla\ell(\Theta^{(t)}, \mathbf{z}_{I^t}) = (\nabla\ell(\theta_1^{(t)}, Z_{I_1^t,1}), \ldots, \nabla\ell(\theta_m^{(t)}, Z_{I_m^t,m}))^\top \in \mathbb{R}^{m \times d},$$

$$\nabla\ell(\bar{\Theta}^{(t)}, \mathbf{z}_{I^t}) = (\nabla\ell(\bar{\theta}_1^{(t)}, Z_{I_1^t,1}), \ldots, \nabla\ell(\bar{\theta}_m^{(t)}, Z_{I_m^t,m}))^\top \in \mathbb{R}^{m \times d},$$

$$\nabla\ell(\Theta^{(t,ij)}, \tilde{\mathbf{z}}_{I^t}) = (\nabla\ell(\theta_1^{(t,ij)}, \tilde{Z}_{I_1^t,1}^{(ij)}), \ldots, \nabla\ell(\theta_m^{(t,ij)}, \tilde{Z}_{I_m^t,m}^{(ij)}))^\top \in \mathbb{R}^{m \times d},$$

$$\nabla\ell(\bar{\Theta}^{(t,ij)}, \tilde{\mathbf{z}}_{I^t}) = (\nabla\ell(\bar{\theta}_1^{(t,ij)}, \tilde{Z}_{I_1^t,1}^{(ij)}), \ldots, \nabla\ell(\bar{\theta}_m^{(t,ij)}, \tilde{Z}_{I_m^t,m}^{(ij)}))^\top \in \mathbb{R}^{m \times d}.$$

Then, we have

$$\Theta^{(t)} = W\Theta^{(t-1)} - \eta_{t-1}\nabla\ell(\Theta^{(t-1)}; \mathbf{z}_{I^{t-1}}), \quad \Theta^{(t,ij)} = W\Theta^{(t-1,ij)} - \eta_{t-1}\nabla\ell(\Theta^{(t-1,ij)}; \tilde{\mathbf{z}}_{I^{t-1}}). \tag{B.2}$$

Let

$$\alpha_1 = (1+\beta)\lambda + 2(1+\beta^{-1})\eta_1^2 L^2, \quad \alpha_2 = 4(1+\beta^{-1}), \quad \alpha_3 = (1+\beta)\lambda, \quad \alpha_4 = 1 + \beta^{-1}, \tag{B.3}$$

where we choose some $\beta > 0$ such that $\alpha_1, \alpha_3 < 1$. For a matrix $A$, we denote by $\|A\|_F$ its Frobenius norm and $\|A\|_2$ its operator norm.

The following lemma shows how to solve a quadratic inequality. We omit the proof for simplicity.

**Lemma B.1.** *Let* $a, b \geq 0$. *If* $x^2 \leq ax + b$, *then* $x^2 \leq a^2 + 2b$.

The following lemma shows the co-coercivity of convex functions with Hölder continuous gradients (Hardt et al., 2016; Lei and Ying, 2020), which plays an important role in our stability analysis.

**Lemma B.2** (Co-coercivity). *If* $\ell(\cdot; z)$ *is convex and has* $(\alpha, L)$-*Hölder continuous gradients with* $\alpha \in [0, 1]$, *then*

$$\langle \theta - \theta', \nabla\ell(\theta; z) - \nabla\ell(\theta'; z)\rangle \geq \frac{2L^{-\frac{1}{\alpha}}\alpha}{1+\alpha}\|\nabla\ell(\theta; z) - \nabla\ell(\theta'; z)\|_2^{\frac{1+\alpha}{\alpha}}.$$

Note if $\alpha = 0$, the right-hand side means 0.

We first present a lemma to control the *neighboring-consensus error* in terms of the difference of gradients.

**Lemma B.3.** *Let Assumption 3.1 hold and* $\alpha_3, \alpha_4$ *be defined in Eq.* (B.3). *Let* $\Theta^{(t)}$, $\Theta^{(t,ij)}$, $\bar{\Theta}^{(t)}$ *and* $\bar{\Theta}^{(t,ij)}$ *be defined as in Eq.* (B.1). *Then,*

$$\sum_{t=0}^{T-1}\frac{\eta_t}{m}\sum_{k=1}^{m}\|\bar{\theta}^{(t)} - \theta_k^{(t)} - \bar{\theta}^{(t,ij)} + \theta_k^{(t,ij)}\|_2^2 \leq \frac{\alpha_4}{(1-\alpha_3)m}\sum_{t=0}^{T-1}\eta_t^3\sum_{k=1}^{m}\|\nabla\ell(\theta_k^{(t)}; Z_{I_k^t,k}) - \ell(\theta_k^{(t,ij)}; \tilde{Z}_{I_k^t,k}^{(ij)})\|_2^2.$$

*Proof.* Denote

$$\mathbf{1} = \begin{pmatrix} 1 \\ \vdots \\ 1 \end{pmatrix} \in \mathbb{R}^{m \times 1}, \quad W^\infty = \frac{1}{m}\mathbf{1}\mathbf{1}^\top = \frac{1}{m}\begin{bmatrix} 1 & \cdots & 1 \\ \vdots & & \vdots \\ 1 & \cdots & 1 \end{bmatrix} \in \mathbb{R}^{m \times m}. \tag{B.4}$$

It is clear that $\bar{\Theta}^{(t)} = W^\infty \Theta^{(t)}$. By Eq. (B.2) and the standard inequality $(a+b)^2 \leq (1+\beta)a^2 + (1+1/\beta)b^2$ for any $\beta > 0$, we know

$$
\begin{aligned}
&\left\|\Theta^{(t)} - \bar{\Theta}^{(t)} - \Theta^{(t,ij)} + \bar{\Theta}^{(t,ij)}\right\|_F^2 \\
&= \left\|\Theta^{(t)} - \Theta^{(t-1)} - \bar{\Theta}^{(t)} + \bar{\Theta}^{(t-1)} - \Theta^{(t,ij)} + \Theta^{(t-1,ij)} + \bar{\Theta}^{(t,ij)} - \bar{\Theta}^{(t-1,ij)}\right\|_F^2 \leq \left\|\Theta^{(t)} - \Theta^{(t-1)} - \Theta^{(t,ij)} + \Theta^{(t-1,ij)}\right\|_F^2 \\
&= \left\|W\Theta^{(t-1)} - \eta_{t-1}\nabla\ell(\Theta^{(t-1)}; \mathbf{z}_{I^{t-1}}) - \bar{\Theta}^{(t-1)} - W\Theta^{(t-1,ij)} + \eta_{t-1}\nabla\ell(\Theta^{(t-1,ij)}; \tilde{\mathbf{z}}_{I^{t-1}}) + \bar{\Theta}^{(t-1,ij)}\right\|_F^2 \\
&\leq (1+\beta)\left\|W(\Theta^{(t-1)} - \Theta^{(t-1,ij)}) - (\bar{\Theta}^{(t-1)} - \bar{\Theta}^{(t-1,ij)})\right\|_F^2 + (1+\tfrac{1}{\beta})\eta_{t-1}^2\left\|\nabla\ell(\Theta^{(t-1)}; \mathbf{z}_{I^{t-1}}) - \nabla\ell(\Theta^{(t-1,ij)}; \tilde{\mathbf{z}}_{I^{t-1}})\right\|_F^2,
\end{aligned}
$$

where we have used $\|\Theta - \bar{\Theta}\|_F \leq \|\Theta\|_F$ for all $\Theta \in \mathbb{R}^{m \times d}$ (Taheri and Thrampoulidis, 2023). For any $\Theta \in \mathbb{R}^{m \times d}$, we know

$$
\begin{aligned}
\|W\Theta - \bar{\Theta}\|_F^2 = \|(W - W^\infty)(\Theta - \bar{\Theta})\|_F^2 &\leq \|W - W^\infty\|_2^2 \|\Theta - \bar{\Theta})\|_F^2 \\
&\leq \max\{\lambda_2^2(W), \lambda_m^2(W)\}\|\Theta - \bar{\Theta}\|_F^2 = \lambda\|\Theta - \bar{\Theta})\|_F^2,
\end{aligned}
\tag{B.5}
$$

where we have used the inequality that $\|W - W^\infty\|_2 \leq \max\{|\lambda_2(W)|, |\lambda_m(W)|\}$ (Taheri and Thrampoulidis, 2023). It then follows that

$$
\begin{aligned}
&\left\|\Theta^{(t)} - \bar{\Theta}^{(t)} - \Theta^{(t,ij)} + \bar{\Theta}^{(t,ij)}\right\|_F^2 \\
&\leq (1+\beta)\lambda\left\|\Theta^{(t-1)} - \Theta^{(t-1,ij)} - \bar{\Theta}^{(t-1)} + \bar{\Theta}^{(t-1,ij)}\right\|_F^2 + (1+\tfrac{1}{\beta})\eta_{t-1}^2\left\|\nabla\ell(\Theta^{(t-1)}; \mathbf{z}_{I^{t-1}}) - \nabla\ell(\Theta^{(t-1,ij)}; \tilde{\mathbf{z}}_{I^{t-1}})\right\|_F^2 \\
&= (1+\beta)\lambda\left\|\Theta^{(t-1)} - \Theta^{(t-1,ij)} - \bar{\Theta}^{(t-1)} + \bar{\Theta}^{(t-1,ij)}\right\|_F^2 + (1+\tfrac{1}{\beta})\eta_{t-1}^2 \sum_{k=1}^m \left\|\nabla\ell(\theta_k^{(t-1)}; Z_{I_k^{t-1},k}) - \ell(\theta_k^{(t-1,ij)}; \tilde{Z}_{I_k^{t-1},k}^{(ij)})\right\|_2^2.
\end{aligned}
$$

By the definition of $\alpha_3$ and $\alpha_4$ in Eq. (B.3), we further get

$$
\begin{aligned}
&\left\|\Theta^{(t)} - \bar{\Theta}^{(t)} - \Theta^{(t,ij)} + \bar{\Theta}^{(t,ij)}\right\|_F^2 \\
&\leq \alpha_3\left\|\Theta^{(t-1)} - \Theta^{(t-1,ij)} - \bar{\Theta}^{(t-1)} + \bar{\Theta}^{(t-1,ij)}\right\|_F^2 + \alpha_4\eta_{t-1}^2 \sum_{k=1}^m \left\|\nabla\ell(\theta_k^{(t-1)}; Z_{I_k^{t-1},k}) - \ell(\theta_k^{(t-1,ij)}; \tilde{Z}_{I_k^{t-1},k}^{(ij)})\right\|_2^2.
\end{aligned}
$$

Applying this inequality repeatedly implies

$$
\left\|\Theta^{(t)} - \bar{\Theta}^{(t)} - \Theta^{(t,ij)} + \bar{\Theta}^{(t,ij)}\right\|_F^2 \leq \alpha_4 \sum_{\tau=1}^{t-1} \alpha_3^{\tau-1} \eta_{t-\tau}^2 \sum_{k=1}^m \left\|\nabla\ell(\theta_k^{(t-\tau)}; Z_{I_k^{t-\tau},k}) - \nabla\ell(\theta_k^{(t-\tau,ij)}; \tilde{Z}_{I_k^{t-\tau},k}^{(ij)})\right\|_2^2.
$$

Since the step size sequence is non-increasing, i.e., $\eta_{t+\tau} \leq \eta_t$, we further get

$$
\begin{aligned}
\sum_{t=0}^{T-1} \frac{\eta_t}{m} \sum_{k=1}^m \left\|\bar{\theta}^{(t)} - \theta_k^{(t)} - \bar{\theta}^{(t,ij)} + \theta_k^{(t,ij)}\right\|_2^2 &\leq \sum_{t=0}^{T-1} \frac{\alpha_4\eta_t}{m} \sum_{\tau=1}^{t-1} \alpha_3^{\tau-1} \eta_{t-\tau}^2 \sum_{k=1}^m \left\|\nabla\ell(\theta_k^{(t-\tau)}; Z_{I_k^{t-\tau},k}) - \ell(\theta_k^{(t-\tau,ij)}; \tilde{Z}_{I_k^{t-\tau},k}^{(ij)})\right\|_2^2 \\
&\leq \frac{\alpha_4}{m} \sum_{\tau=1}^{T-1} \sum_{t=\tau+1}^{T-1} \eta_t \alpha_3^{\tau-1} \eta_{t-\tau}^2 \sum_{k=1}^m \left\|\nabla\ell(\theta_k^{(t-\tau)}; Z_{I_k^{t-\tau},k}) - \ell(\theta_k^{(t-\tau,ij)}; \tilde{Z}_{I_k^{t-\tau},k}^{(ij)})\right\|_2^2 \\
&\leq \frac{\alpha_4}{m} \sum_{\tau=1}^{T-1} \alpha_3^{\tau-1} \sum_{t=0}^{T-\tau} \eta_{t+\tau} \eta_t^2 \sum_{k=1}^m \left\|\nabla\ell(\theta_k^{(t)}; Z_{I_k^t,k}) - \ell(\theta_k^{(t,ij)}; \tilde{Z}_{I_k^t,k}^{(ij)})\right\|_2^2 \\
&\leq \frac{\alpha_4}{m} \sum_{\tau=1}^{T-1} \alpha_3^{\tau-1} \sum_{t=0}^{T-1} \eta_{t+\tau} \eta_t^2 \sum_{k=1}^m \left\|\nabla\ell(\theta_k^{(t)}; Z_{I_k^t,k}) - \ell(\theta_k^{(t,ij)}; \tilde{Z}_{I_k^t,k}^{(ij)})\right\|_2^2 \\
&\leq \frac{\alpha_4}{m} \sum_{\tau=1}^{T-1} \alpha_3^{\tau-1} \sum_{t=0}^{T-1} \eta_t^3 \sum_{k=1}^m \left\|\nabla\ell(\theta_k^{(t)}; Z_{I_k^t,k}) - \ell(\theta_k^{(t,ij)}; \tilde{Z}_{I_k^t,k}^{(ij)})\right\|_2^2 \\
&\leq \frac{\alpha_4}{(1-\alpha_3)m} \sum_{t=0}^{T-1} \eta_t^3 \sum_{k=1}^m \left\|\nabla\ell(\theta_k^{(t)}; Z_{I_k^t,k}) - \ell(\theta_k^{(t,ij)}; \tilde{Z}_{I_k^t,k}^{(ij)})\right\|_2^2.
\end{aligned}
$$

The proof is completed. □

We first present a general stability bound with a general step size. As we will see, Theorem 4.1 follows as a direct corollary of Theorem B.4.

**Theorem B.4** (Stability bound). *Let Assumption 3.1 hold. Let $A(S)$ and $A(S^{(ij)})$ be defined in Eq. (4.1). Let $\ell(\cdot; z)$ be convex and smooth. For any $\beta_1 > 0, \beta_2 > \frac{L}{2}$, assume the step size sequence satisfies*

$$\Big(\frac{\alpha_4 \beta_2}{(1-\alpha_3)}\eta_t + \frac{(1+\beta_1)(m-1)}{m}\Big)\eta_t - \Big(\frac{2}{L} - \frac{1}{\beta_2}\Big) \leq 0 \tag{B.6}$$

*and*

$$\Big(\frac{\alpha_4 \beta_2}{(1-\alpha_3)}\eta_t + \frac{1+\frac{1}{\beta_1}}{m}\Big)\eta_t - \Big(\frac{2}{L} - \frac{1}{\beta_2}\Big) \leq 0. \tag{B.7}$$

*Then, we have*

$$\frac{1}{mn}\sum_{i=1}^{n}\sum_{j=1}^{m}\mathbb{E}[\|A(S) - A(S^{(ij)})\|_2^2] \leq \frac{8}{m^2 n^2}\sum_{i=1}^{n}\sum_{j=1}^{m}\sum_{t=0}^{T-1}\Big(\frac{\alpha_4 \beta_2}{1-\alpha_3}\eta_t + \frac{1+\frac{1}{\beta_1}}{m}\Big)\eta_t^2\mathbb{E}[\|\nabla\ell(\theta_j^{(t)}; z_{i,j})\|_2^2]$$

$$+ \frac{16}{m^3 n^3}\sum_{i=1}^{n}\sum_{j=1}^{m}\Big(\sum_{t=0}^{T-1}\eta_t\Big(\mathbb{E}\big[\|\nabla\ell(\theta_j^{(t)}; z_{i,j})\|_2^2\big]\Big)^{\frac{1}{2}}\Big)^2. \tag{B.8}$$

*Proof.* We first temporarily fix $i \in [n], j \in [m]$ and control $\|A(S) - A(S^{(ij)})\|_2$. Recall the notation in Eq. (B.1). According to the implementation of D-SGD, we know

$$\bar{\theta}^{(t+1)} = \bar{\theta}^{(t)} - \frac{\eta_t}{m}\sum_{k=1}^{m}\nabla\ell(\theta_k^{(t)}; Z_{I_k^t, k}) \quad \text{and} \quad \bar{\theta}^{(t+1,ij)} = \bar{\theta}^{(t,ij)} - \frac{\eta_t}{m}\sum_{k=1}^{m}\nabla\ell(\theta_k^{(t,ij)}; \tilde{Z}_{I_k^t, k}^{(ij)}).$$

It then follows that

$$\|\bar{\theta}^{(t+1)} - \bar{\theta}^{(t+1,ij)}\|_2^2 = \|\bar{\theta}^{(t)} - \bar{\theta}^{(t,ij)} - \frac{\eta_t}{m}\sum_{k=1}^{m}\nabla\ell(\theta_k^{(t)}; Z_{I_k^t, k}) + \frac{\eta_t}{m}\sum_{k=1}^{m}\nabla\ell(\theta_k^{(t,ij)}; \tilde{Z}_{I_k^t, k}^{(ij)})\|_2^2$$

$$= \|\bar{\theta}^{(t)} - \bar{\theta}^{(t,ij)}\|_2^2 + \frac{\eta_t^2}{m^2}\Big\|\sum_{k=1}^{m}\big(\nabla\ell(\theta_k^{(t)}; Z_{I_k^t, k}) - \nabla\ell(\theta_k^{(t,ij)}; \tilde{Z}_{I_k^t, k}^{(ij)})\big)\Big\|_2^2$$

$$- \frac{2\eta_t}{m}\sum_{k=1}^{m}\big\langle\bar{\theta}^{(t)} - \bar{\theta}^{(t,ij)}, \nabla\ell(\theta_k^{(t)}; Z_{I_k^t, k}) - \nabla\ell(\theta_k^{(t,ij)}; \tilde{Z}_{I_k^t, k}^{(ij)})\big\rangle. \tag{B.9}$$

**Part 1: For the second term in Eq. (B.9)**, for any $\beta_1 > 0$, we have

$$\Big\|\sum_{k=1}^{m}\big(\nabla\ell(\theta_k^{(t)}; Z_{I_k^t, k}) - \nabla\ell(\theta_k^{(t,ij)}; \tilde{Z}_{I_k^t, k}^{(ij)})\big)\Big\|_2^2$$

$$\leq (1+\beta_1)\Big\|\sum_{k\neq j}\big(\nabla\ell(\theta_k^{(t)}; Z_{I_k^t, k}) - \nabla\ell(\theta_k^{(t,ij)}; \tilde{Z}_{I_k^t, k}^{(ij)})\big)\Big\|_2^2 + (1+\frac{1}{\beta_1})\big\|\nabla\ell(\theta_j^{(t)}; Z_{I_j^t, j}) - \nabla\ell(\theta_j^{(t,ij)}; \tilde{Z}_{I_j^t, j}^{(ij)})\big\|_2^2$$

$$\leq (1+\beta_1)(m-1)\sum_{k\neq j}\big\|\big(\nabla\ell(\theta_k^{(t)}; Z_{I_k^t, k}) - \nabla\ell(\theta_k^{(t,ij)}; \tilde{Z}_{I_k^t, k}^{(ij)})\big)\big\|_2^2 + (1+\frac{1}{\beta_1})\big\|\nabla\ell(\theta_j^{(t)}; Z_{I_j^t, j}) - \nabla\ell(\theta_j^{(t,ij)}; \tilde{Z}_{I_j^t, j}^{(ij)})\big\|_2^2, \tag{B.10}$$

where the last step uses the Cauchy-Schwartz inequality.

**Part 2: For the last term in Eq. (B.9)**, we first consider two cases to control

$$\big\langle\bar{\theta}^{(t)} - \bar{\theta}^{(t,ij)}, \nabla\ell(\theta_k^{(t)}; Z_{I_k^t, k}) - \nabla\ell(\theta_k^{(t,ij)}; \tilde{Z}_{I_k^t, k}^{(ij)})\big\rangle.$$

(1) if $k = j$ and $I_j^t = i$, then it is clear

$$\langle \bar{\theta}^{(t)} - \bar{\theta}^{(t,ij)}, \nabla\ell(\theta_j^{(t)}; z_{i,j}) - \nabla\ell(\theta_j^{(t,ij)}; \tilde{z}_{i,j})\rangle \geq -\big\|\bar{\theta}^{(t)} - \bar{\theta}^{(t,ij)}\big\|_2 \big\|\nabla\ell(\theta_j^{(t)}; z_{i,j}) - \nabla\ell(\theta_j^{(t,ij)}; \tilde{z}_{i,j})\big\|_2. \quad \text{(B.11)}$$

(2) if ($k = j$ and $I_j^t \neq i$) or if $k \neq j$, then

$$\langle \bar{\theta}^{(t)} - \bar{\theta}^{(t,ij)}, \nabla\ell(\theta_k^{(t)}; Z_{I_k^t,k}) - \nabla\ell(\theta_k^{(t,ij)}; \tilde{Z}_{I_k^t,k}^{(ij)})\rangle = \langle \theta_k^{(t)} - \theta_k^{(t,ij)}, \nabla\ell(\theta_k^{(t)}; Z_{I_k^t,k}) - \nabla\ell(\theta_k^{(t,ij)}; \tilde{Z}_{I_k^t,k}^{(ij)})\rangle$$
$$+ \langle \bar{\theta}^{(t)} - \theta_k^{(t)} - \bar{\theta}^{(t,ij)} + \theta_k^{(t,ij)}, \nabla\ell(\theta_k^{(t)}; Z_{I_k^t,k}) - \nabla\ell(\theta_k^{(t,ij)}; \tilde{Z}_{I_k^t,k}^{(ij)})\rangle. \quad \text{(B.12)}$$

The convexity and $L$-smoothness imply that the gradients are co-coercive (Lemma B.2), namely,

$$\langle \theta_k^{(t)} - \theta_k^{(t,ij)}, \nabla\ell(\theta_k^{(t)}; Z_{I_k^t,k}) - \nabla\ell(\theta_k^{(t,ij)}; \tilde{Z}_{I_k^t,k}^{(ij)})\rangle \geq \frac{1}{L}\big\|\nabla\ell(\theta_k^{(t)}; Z_{I_k^t,k}) - \nabla\ell(\theta_k^{(t,ij)}; \tilde{Z}_{I_k^t,k}^{(ij)})\big\|_2^2. \quad \text{(B.13)}$$

For any $\beta_2 > 0$, we know

$$\langle \bar{\theta}^{(t)} - \theta_k^{(t)} - \bar{\theta}^{(t,ij)} + \theta_k^{(t,ij)}, \nabla\ell(\theta_k^{(t)}; Z_{I_k^t,k}) - \nabla\ell(\theta_k^{(t,ij)}; \tilde{Z}_{I_k^t,k}^{(ij)})\rangle$$
$$\geq -\big\|\bar{\theta}^{(t)} - \theta_k^{(t)} - \bar{\theta}^{(t,ij)} + \theta_k^{(t,ij)}\big\|_2 \big\|\nabla\ell(\theta_k^{(t)}; Z_{I_k^t,k}) - \nabla\ell(\theta_k^{(t,ij)}; \tilde{Z}_{I_k^t,k}^{(ij)})\big\|_2$$
$$\geq -\frac{\beta_2}{2}\big\|\bar{\theta}^{(t)} - \theta_k^{(t)} - \bar{\theta}^{(t,ij)} + \theta_k^{(t,ij)}\big\|_2^2 - \frac{1}{2\beta_2}\big\|\nabla\ell(\theta_k^{(t)}; Z_{I_k^t,k}) - \nabla\ell(\theta_k^{(t,ij)}; \tilde{Z}_{I_k^t,k}^{(ij)})\big\|_2^2. \quad \text{(B.14)}$$

Plugging Eq. (B.13) and Eq. (B.14) into Eq. (B.12), we have, for ($k = j$ and $I_j^t \neq i$) or $k \neq j$:

$$\langle \bar{\theta}^{(t)} - \bar{\theta}^{(t,ij)}, \nabla\ell(\theta_k^{(t)}; Z_{I_k^t,k}) - \nabla\ell(\theta_k^{(t,ij)}; \tilde{Z}_{I_k^t,k}^{(ij)})\rangle$$
$$\geq -\frac{\beta_2}{2}\big\|\bar{\theta}^{(t)} - \theta_k^{(t)} - \bar{\theta}^{(t,ij)} + \theta_k^{(t,ij)}\big\|_2^2 + \big(\frac{1}{L} - \frac{1}{2\beta_2}\big)\big\|\nabla\ell(\theta_k^{(t)}; Z_{I_k^t,k}) - \nabla\ell(\theta_k^{(t,ij)}; \tilde{Z}_{I_k^t,k}^{(ij)})\big\|_2^2. \quad \text{(B.15)}$$

Then, we combine the results above to control

$$-\frac{2\eta_t}{m}\sum_{k=1}^m \langle \bar{\theta}^{(t)} - \bar{\theta}^{(t,ij)}, \nabla\ell(\theta_k^{(t)}; Z_{I_k^t,k}) - \nabla\ell(\theta_k^{(t,ij)}; \tilde{Z}_{I_k^t,k}^{(ij)})\rangle.$$

(1) If $I_j^t \neq i$, by plugging in Eq. (B.15), we have

$$-\frac{2\eta_t}{m}\sum_{k=1}^m \langle \bar{\theta}^{(t)} - \bar{\theta}^{(t,ij)}, \nabla\ell(\theta_k^{(t)}; Z_{I_k^t,k}) - \nabla\ell(\theta_k^{(t,ij)}; \tilde{Z}_{I_k^t,k}^{(ij)})\rangle$$
$$\leq \frac{\beta_2\eta_t}{m}\sum_{k=1}^m \big\|\bar{\theta}^{(t)} - \theta_k^{(t)} - \bar{\theta}^{(t,ij)} + \theta_k^{(t,ij)}\big\|_2^2 - \big(\frac{2}{L} - \frac{1}{\beta_2}\big)\frac{\eta_t}{m}\sum_{k=1}^m \big\|\nabla\ell(\theta_k^{(t)}; Z_{I_k^t,k}) - \nabla\ell(\theta_k^{(t,ij)}; \tilde{Z}_{I_k^t,k}^{(ij)})\big\|_2^2. \quad \text{(B.16)}$$

(2) If $I_j^t = i$, we know

$$-\frac{2\eta_t}{m}\sum_{k=1}^m \langle \bar{\theta}^{(t)} - \bar{\theta}^{(t,ij)}, \nabla\ell(\theta_k^{(t)}; Z_{I_k^t,k}) - \nabla\ell(\theta_k^{(t,ij)}; \tilde{Z}_{I_k^t,k}^{(ij)})\rangle$$
$$= -\frac{2\eta_t}{m}\sum_{k\neq j} \langle \bar{\theta}^{(t)} - \bar{\theta}^{(t,ij)}, \nabla\ell(\theta_k^{(t)}; Z_{I_k^t,k}) - \nabla\ell(\theta_k^{(t,ij)}; \tilde{Z}_{I_k^t,k}^{(ij)})\rangle - \frac{2\eta_t}{m}\langle \bar{\theta}^{(t)} - \bar{\theta}^{(t,ij)}, \nabla\ell(\theta_j^{(t)}; Z_{I_j^t,j}) - \nabla\ell(\theta_j^{(t,ij)}; \tilde{Z}_{I_j^t,j}^{(ij)})\rangle$$
$$\leq -\frac{2\eta_t}{m}\sum_{k\neq j} \langle \bar{\theta}^{(t)} - \bar{\theta}^{(t,ij)}, \nabla\ell(\theta_k^{(t)}; Z_{I_k^t,k}) - \nabla\ell(\theta_k^{(t,ij)}; \tilde{Z}_{I_k^t,k}^{(ij)})\rangle + \frac{2\eta_t}{m}\big\|\bar{\theta}^{(t)} - \bar{\theta}^{(t,ij)}\big\|_2 \big\|\nabla\ell(\theta_j^{(t)}; z_{i,j}) - \nabla\ell(\theta_j^{(t,ij)}; \tilde{z}_{i,j})\big\|_2$$

From Eq. (B.15), we know

$$
-\frac{2\eta_t}{m}\sum_{k\neq j}\big\langle \bar{\theta}^{(t)} - \bar{\theta}^{(t,ij)}, \nabla\ell(\theta_k^{(t)}; Z_{I_k^t,k}) - \nabla\ell(\theta_k^{(t,ij)}; \tilde{Z}_{I_k^t,k}^{(ij)})\big\rangle
$$
$$
\leq \frac{\beta_2\eta_t}{m}\sum_{k\neq j}\big\|\bar{\theta}^{(t)} - \theta_k^{(t)} - \bar{\theta}^{(t,ij)} + \theta_k^{(t,ij)}\big\|_2^2 - \big(\frac{2}{L} - \frac{1}{\beta_2}\big)\frac{\eta_t}{m}\sum_{k\neq j}\big\|\nabla\ell(\theta_k^{(t)}; Z_{I_k^t,k}) - \nabla\ell(\theta_k^{(t,ij)}; \tilde{Z}_{I_k^t,k}^{(ij)})\big\|_2^2
$$
$$
\leq \frac{\beta_2\eta_t}{m}\sum_{k=1}^m\big\|\bar{\theta}^{(t)} - \theta_k^{(t)} - \bar{\theta}^{(t,ij)} + \theta_k^{(t,ij)}\big\|_2^2 - \big(\frac{2}{L} - \frac{1}{\beta_2}\big)\frac{\eta_t}{m}\sum_{k\neq j}\big\|\nabla\ell(\theta_k^{(t)}; Z_{I_k^t,k}) - \nabla\ell(\theta_k^{(t,ij)}; \tilde{Z}_{I_k^t,k}^{(ij)})\big\|_2^2.
$$

Combining these two inequalities above, for the case when $I_j^t = i$, we have

$$
-\frac{2\eta_t}{m}\sum_{k=1}^m\big\langle \bar{\theta}^{(t)} - \bar{\theta}^{(t,ij)}, \nabla\ell(\theta_k^{(t)}; Z_{I_k^t,k}) - \nabla\ell(\theta_k^{(t,ij)}; \tilde{Z}_{I_k^t,k}^{(ij)})\big\rangle
$$
$$
\leq \frac{\beta_2\eta_t}{m}\sum_{k=1}^m\big\|\bar{\theta}^{(t)} - \theta_k^{(t)} - \bar{\theta}^{(t,ij)} + \theta_k^{(t,ij)}\big\|_2^2 - \big(\frac{2}{L} - \frac{1}{\beta_2}\big)\frac{\eta_t}{m}\sum_{k\neq j}\big\|\nabla\ell(\theta_k^{(t)}; Z_{I_k^t,k}) - \nabla\ell(\theta_k^{(t,ij)}; \tilde{Z}_{I_k^t,k}^{(ij)})\big\|_2^2
$$
$$
+ \frac{2\eta_t}{m}\big\|\bar{\theta}^{(t)} - \bar{\theta}^{(t,ij)}\big\|_2\big\|\nabla\ell(\theta_j^{(t)}; z_{i,j}) - \nabla\ell(\theta_j^{(t,ij)}; \tilde{z}_{i,j})\big\|_2. \tag{B.17}
$$

Therefore, combining the cases when $I_j^t \neq i$ and $I_j^t = i$, i.e., Eq. (B.16) and Eq. (B.17), we have

$$
-\frac{2\eta_t}{m}\sum_{k=1}^m\big\langle \bar{\theta}^{(t)} - \bar{\theta}^{(t,ij)}, \nabla\ell(\theta_k^{(t)}; Z_{I_k^t,k}) - \nabla\ell(\theta_k^{(t,ij)}; \tilde{Z}_{I_k^t,k}^{(ij)})\big\rangle
$$
$$
\leq \frac{\beta_2\eta_t}{m}\sum_{k=1}^m\big\|\bar{\theta}^{(t)} - \theta_k^{(t)} - \bar{\theta}^{(t,ij)} + \theta_k^{(t,ij)}\big\|_2^2 - \big(\frac{2}{L} - \frac{1}{\beta_2}\big)\frac{\eta_t}{m}\sum_{k\neq j}\big\|\nabla\ell(\theta_k^{(t)}; Z_{I_k^t,k}) - \nabla\ell(\theta_k^{(t,ij)}; \tilde{Z}_{I_k^t,k}^{(ij)})\big\|_2^2
$$
$$
-\big(\frac{2}{L} - \frac{1}{\beta_2}\big)\frac{\eta_t}{m}\big\|\nabla\ell(\theta_j^{(t)}; Z_{I_j^t,j}) - \nabla\ell(\theta_j^{(t,ij)}; \tilde{Z}_{I_j^t,j}^{(ij)})\big\|_2^2\mathbb{I}_{\{I_j^t\neq i\}} + \frac{2\eta_t}{m}\big\|\bar{\theta}^{(t)} - \bar{\theta}^{(t,ij)}\big\|_2\big\|\nabla\ell(\theta_j^{(t)}; z_{i,j}) - \nabla\ell(\theta_j^{(t,ij)}; \tilde{z}_{i,j})\big\|_2\mathbb{I}_{\{I_j^t=i\}}, \tag{B.18}
$$

where $\mathbb{I}_{\{\cdot\}}$ denotes the indicator function, i.e., returning 1 if the argument holds and 0 otherwise.

**Iteration form:** Plugging Eq. (B.10) and Eq. (B.18) into Eq. (B.9), we have

$$
\|\bar{\theta}^{(t+1)} - \bar{\theta}^{(t+1,ij)}\|_2^2 \leq \|\bar{\theta}^{(t)} - \bar{\theta}^{(t,ij)}\|_2^2 + \frac{\beta_2\eta_t}{m}\sum_{k=1}^m\big\|\bar{\theta}^{(t)} - \theta_k^{(t)} - \bar{\theta}^{(t,ij)} + \theta_k^{(t,ij)}\big\|_2^2
$$
$$
+ \Big((1+\beta_1)(m-1)\frac{\eta_t^2}{m^2} - \big(\frac{2}{L} - \frac{1}{\beta_2}\big)\frac{\eta_t}{m}\Big)\sum_{k\neq j}\big\|\big(\nabla\ell(\theta_k^{(t)}; Z_{I_k^t,k}) - \nabla\ell(\theta_k^{(t,ij)}; \tilde{Z}_{I_k^t,k}^{(ij)})\big)\big\|_2^2
$$
$$
+ \Big((1+\frac{1}{\beta_1})\frac{\eta_t^2}{m^2} - \big(\frac{2}{L} - \frac{1}{\beta_2}\big)\frac{\eta_t}{m}\Big)\big\|\nabla\ell(\theta_j^{(t)}; Z_{I_j^t,j}) - \nabla\ell(\theta_j^{(t,ij)}; \tilde{Z}_{I_j^t,j}^{(ij)})\big\|_2^2\mathbb{I}_{\{I_j^t\neq i\}}
$$
$$
+ (1+\frac{1}{\beta_1})\frac{\eta_t^2}{m^2}\big\|\big(\nabla\ell(\theta_j^{(t)}; z_{i,j}) - \nabla\ell(\theta_j^{(t,ij)}; \tilde{z}_{i,j})\big)\big\|_2^2\mathbb{I}_{\{I_j^t=i\}}
$$
$$
+ \frac{2\eta_t}{m}\big\|\bar{\theta}^{(t)} - \bar{\theta}^{(t,ij)}\big\|_2\big\|\nabla\ell(\theta_j^{(t)}; z_{i,j}) - \nabla\ell(\theta_j^{(t,ij)}; \tilde{z}_{i,j})\big\|_2\mathbb{I}_{\{I_j^t=i\}}. \tag{B.19}
$$

By Lemma B.3, we get

$$
\sum_{t=0}^{T-1} \frac{\beta_2 \eta_t}{m} \sum_{k=1}^{m} \left\| \bar{\theta}^{(t)} - \theta_k^{(t)} - \bar{\theta}^{(t,ij)} + \theta_k^{(t,ij)} \right\|_2^2
$$

$$
+ \sum_{t=0}^{T-1} \left( (1+\beta_1)(m-1)\frac{\eta_t^2}{m^2} - (\frac{2}{L} - \frac{1}{\beta_2})\frac{\eta_t}{m} \right) \sum_{k \neq j} \left\| (\nabla\ell(\theta_k^{(t)}; Z_{I_k^t,k}) - \nabla\ell(\theta_k^{(t,ij)}; \tilde{Z}_{I_k^t,k}^{(ij)})) \right\|_2^2
$$

$$
+ \sum_{t=0}^{T-1} \left( (1+\frac{1}{\beta_1})\frac{\eta_t^2}{m^2} - (\frac{2}{L} - \frac{1}{\beta_2})\frac{\eta_t}{m} \right) \left\| \nabla\ell(\theta_j^{(t)}; Z_{I_j^t,j}) - \nabla\ell(\theta_j^{(t,ij)}; \tilde{Z}_{I_j^t,j}^{(ij)}) \right\|_2^2 \mathbb{I}_{\{I_j^t \neq i\}}
$$

$$
\leq \sum_{t=0}^{T-1} \frac{\eta_t}{m} \left( \left( \frac{\alpha_4\beta_2}{(1-\alpha_3)}\eta_t + \frac{(1+\beta_1)(m-1)}{m} \right)\eta_t - \left(\frac{2}{L} - \frac{1}{\beta_2}\right) \right) \sum_{k \neq j} \left\| (\nabla\ell(\theta_k^{(t)}; Z_{I_k^t,k}) - \nabla\ell(\theta_k^{(t,ij)}; \tilde{Z}_{I_k^t,k}^{(ij)})) \right\|_2^2
$$

$$
+ \sum_{t=0}^{T-1} \frac{\eta_t}{m} \left( \left( \frac{\alpha_4\beta_2}{(1-\alpha_3)}\eta_t + \frac{1+\frac{1}{\beta_1}}{m} \right)\eta_t - \left(\frac{2}{L} - \frac{1}{\beta_2}\right) \right) \left\| \nabla\ell(\theta_j^{(t)}; Z_{I_j^t,j}) - \nabla\ell(\theta_j^{(t,ij)}; \tilde{Z}_{I_j^t,j}^{(ij)}) \right\|_2^2 \mathbb{I}_{\{I_j^t \neq i\}}
$$

$$
+ \sum_{t=0}^{T-1} \frac{\eta_t^3 \alpha_4 \beta_2 \mathbb{I}_{\{I_j^t=i\}}}{(1-\alpha_3)m} \left\| \nabla\ell(\theta_j^{(t)}; Z_{I_j^t,j}) - \nabla\ell(\theta_j^{(t,ij)}; \tilde{Z}_{I_j^t,j}^{(ij)}) \right\|_2^2 \leq \sum_{t=0}^{T-1} \frac{\eta_t^3 \alpha_4 \beta_2 \mathbb{I}_{\{I_j^t=i\}}}{(1-\alpha_3)m} \left\| \nabla\ell(\theta_j^{(t)}; Z_{I_j^t,j}) - \nabla\ell(\theta_j^{(t,ij)}; \tilde{Z}_{I_j^t,j}^{(ij)}) \right\|_2^2,
$$

where we have used Eq. (B.6) and Eq. (B.7) in the last inequality. Therefore, taking a summation for $t = 0, \ldots, T-1$ to both sides of Eq. (B.19) and using Lemma B.3, we have

$$
\|\bar{\theta}^{(T)} - \bar{\theta}^{(T,ij)}\|_2^2 \leq \sum_{t=0}^{T-1} \left( \frac{\alpha_4\beta_2}{1-\alpha_3}\eta_t + \frac{1+\frac{1}{\beta_1}}{m} \right)\frac{\eta_t^2}{m} \left\| (\nabla\ell(\theta_j^{(t)}; z_{i,j}) - \nabla\ell(\theta_j^{(t,ij)}; \tilde{z}_{i,j})) \right\|_2^2 \mathbb{I}_{\{I_j^t=i\}}
$$

$$
+ \sum_{t=0}^{T-1} \frac{2\eta_t}{m} \left\| \bar{\theta}^{(t)} - \bar{\theta}^{(t,ij)} \right\|_2 \left\| \nabla\ell(\theta_j^{(t)}; z_{i,j}) - \nabla\ell(\theta_j^{(t,ij)}; \tilde{z}_{i,j}) \right\|_2 \mathbb{I}_{\{I_j^t=i\}}. \tag{B.20}
$$

Taking expectation to both sides of Eq. (B.20) and using the fact that $I_j^t$ is independent of $\bar{\theta}^{(t)}$, $\bar{\theta}^{(t,ij)}$, $\theta_j^{(t)}$ and $\theta_j^{(t,ij)}$, we know

$$
\mathbb{E}[\|\bar{\theta}^{(T)} - \bar{\theta}^{(t,ij)}\|_2^2] \leq \frac{1}{mn} \sum_{t=0}^{T-1} \left( \frac{\alpha_4\beta_2}{1-\alpha_3}\eta_t + \frac{1+\frac{1}{\beta_1}}{m} \right)\eta_t^2 \mathbb{E}[\|(\nabla\ell(\theta_j^{(t)}; z_{i,j}) - \nabla\ell(\theta_j^{(t,ij)}; \tilde{z}_{i,j}))\|_2^2]
$$

$$
+ \sum_{t=0}^{T-1} \frac{2\eta_t}{mn} \mathbb{E}[\|\bar{\theta}^{(t)} - \bar{\theta}^{(t,ij)}\|_2 \|\nabla\ell(\theta_j^{(t)}; z_{i,j}) - \nabla\ell(\theta_j^{(t,ij)}; \tilde{z}_{i,j})\|_2]. \tag{B.21}
$$

Introduce

$$
\Delta_t = \left( \mathbb{E}[\|\bar{\theta}^{(t)} - \bar{\theta}^{(t,ij)}\|_2^2] \right)^{\frac{1}{2}}, \quad \forall t \in [T] \quad \textbf{and} \quad \Delta = \max_{t \leq T} \Delta_t.
$$

By the Cauchy–Schwarz inequality, we know

$$
\mathbb{E}\left[ \|\bar{\theta}^{(t)} - \bar{\theta}^{(t,ij)}\|_2 \|\nabla\ell(\theta_j^{(t)}; z_{i,j}) - \nabla\ell(\theta_j^{(t,ij)}; \tilde{z}_{i,j})\|_2 \right] \leq \left( \mathbb{E}\left[ \|\bar{\theta}^{(t)} - \bar{\theta}^{(t,ij)}\|_2^2 \right] \right)^{\frac{1}{2}} \left( \mathbb{E}\left[ \|\nabla\ell(\theta_j^{(t)}; z_{i,j}) - \nabla\ell(\theta_j^{(t,ij)}; \tilde{z}_{i,j})\|_2^2 \right] \right)^{\frac{1}{2}}.
$$

It then follows that

$$
\Delta^2 \leq \frac{1}{mn} \sum_{t=0}^{T-1} \left( \frac{\alpha_4\beta_2}{1-\alpha_3}\eta_t + \frac{1+\frac{1}{\beta_1}}{m} \right)\eta_t^2 \mathbb{E}[\|(\nabla\ell(\theta_j^{(t)}; z_{i,j}) - \nabla\ell(\theta_j^{(t,ij)}; \tilde{z}_{i,j}))\|_2^2]
$$

$$
+ \frac{2}{mn} \sum_{t=0}^{T-1} \eta_t \left( \mathbb{E}\left[ \|\nabla\ell(\theta_j^{(t)}; z_{i,j}) - \nabla\ell(\theta_j^{(t,ij)}; \tilde{z}_{i,j})\|_2^2 \right] \right)^{\frac{1}{2}} \Delta. \tag{B.22}
$$

According to Lemma B.1, this further shows

$$\Delta^2 \leq \frac{2}{mn} \sum_{t=0}^{T-1} \left(\frac{\alpha_4\beta_2}{1-\alpha_3}\eta_t + \frac{1+\frac{1}{\beta_1}}{m}\right)\eta_t^2 \mathbb{E}\left[\left\|\left(\nabla\ell(\theta_j^{(t)};z_{i,j}) - \nabla\ell(\theta_j^{(t,ij)};\tilde{z}_{i,j})\right)\right\|_2^2\right]$$
$$+ \frac{4}{m^2n^2}\left(\sum_{t=0}^{T-1}\eta_t\left(\mathbb{E}\left[\left\|\nabla\ell(\theta_j^{(t)};z_{i,j}) - \nabla\ell(\theta_j^{(t,ij)};\tilde{z}_{i,j})\right\|_2^2\right]\right)^{\frac{1}{2}}\right)^2. \tag{B.23}$$

By the symmetry between $z_{i,j}$ and $\tilde{z}_{i,j}$, we know

$$\mathbb{E}\left[\|\nabla\ell(\theta_j^{(t)};z_{i,j}) - \nabla\ell(\theta_j^{(t,ij)};\tilde{z}_{i,j})\|_2^2\right] \leq 2\mathbb{E}\left[\|\nabla\ell(\theta_j^{(t)};z_{i,j})\|_2^2\right] + 2\mathbb{E}\left[\|\nabla\ell(\theta_j^{(t,ij)};\tilde{z}_{i,j})\|_2^2\right]$$
$$= 4\mathbb{E}\left[\|\nabla\ell(\theta_j^{(t)};z_{i,j})\|_2^2\right]. \tag{B.24}$$

We combine the above two inequalities together, and get

$$\mathbb{E}[\|A(S) - A(S^{(ij)})\|_2^2] \leq \frac{8}{mn}\sum_{t=0}^{T-1}\left(\frac{\alpha_4\beta_2}{1-\alpha_3}\eta_t + \frac{1+\frac{1}{\beta_1}}{m}\right)\eta_t^2\mathbb{E}\left[\|\nabla\ell(\theta_j^{(t)};z_{i,j})\|_2^2\right]$$
$$+ \frac{16}{m^2n^2}\left(\sum_{t=0}^{T-1}\eta_t\left(\mathbb{E}\left[\|\nabla\ell(\theta_j^{(t)};z_{i,j})\|_2^2\right]\right)^{\frac{1}{2}}\right)^2. \tag{B.25}$$

Taking an average over all $i \in [n]$ and $j \in [m]$, we get

$$\frac{1}{mn}\sum_{i=1}^{n}\sum_{j=1}^{m}\mathbb{E}[\|A(S) - A(S^{(ij)})\|_2^2] \leq \frac{8}{m^2n^2}\sum_{i=1}^{n}\sum_{j=1}^{m}\sum_{t=0}^{T-1}\left(\frac{\alpha_4\beta_2}{1-\alpha_3}\eta_t + \frac{1+\frac{1}{\beta_1}}{m}\right)\eta_t^2\mathbb{E}\left[\|\nabla\ell(\theta_j^{(t)};z_{i,j})\|_2^2\right]$$
$$+ \frac{16}{m^3n^3}\sum_{i=1}^{n}\sum_{j=1}^{m}\left(\sum_{t=0}^{T-1}\eta_t\left(\mathbb{E}\left[\|\nabla\ell(\theta_j^{(t)};z_{i,j})\|_2^2\right]\right)^{\frac{1}{2}}\right)^2.$$

The proof is completed. $\qquad\square$

*Proof of Theorem 4.1.* By choosing $\beta = \frac{1-\lambda}{2\lambda}$, we know

$$\frac{\alpha_4}{1-\alpha_3} = \frac{1+\beta^{-1}}{1-(1+\beta)\lambda} = \frac{1+\frac{2\lambda}{1-\lambda}}{1-(1+\frac{1-\lambda}{2\lambda})\lambda} = \frac{\frac{1+\lambda}{1-\lambda}}{\frac{1-\lambda}{2}} = \frac{2(1+\lambda)}{(1-\lambda)^2}.$$

The stated bound then follows from Theorem B.4 by choosing $\beta_1 = 1$ and $\beta_2 = L$ (note Eq. (B.6) and (B.7) hold if Eq. (4.2) holds). $\qquad\square$

*Proof of Corollary 4.4.* By the Cauchy–Schwarz inequality, we know

$$\left(\sum_{t=0}^{T-1}\eta_t\left(\mathbb{E}\left[\|\nabla\ell(\theta_j^{(t)};z_{i,j})\|_2^2\right]\right)^{\frac{1}{2}}\right)^2 \leq \sum_{t=0}^{T-1}\eta_t^2\sum_{t=0}^{T-1}\mathbb{E}\left[\|\nabla\ell(\theta_j^{(t)};z_{i,j})\|_2^2\right]. \tag{B.26}$$

By the elementary inequality $a^2 \leq 2(a-b)^2 + 2b^2$, the $L$-smoothness and self bounding property of the loss function $\ell$ (Lemma A.1), we know

$$\left\|\nabla\ell(\theta_j^{(t)};z_{i,j})\right\|_2^2 \leq 2\left\|\nabla\ell(\theta_j^{(t)};z_{i,j}) - \nabla\ell(\bar{\theta}^{(t)};z_{i,j})\right\|_2^2 + 2\left\|\nabla\ell(\bar{\theta}^{(t)};z_{i,j})\right\|_2^2$$
$$\leq 2L^2\left\|\theta_j^{(t)} - \bar{\theta}^{(t)}\right\|_2^2 + 2\left\|\nabla\ell(\bar{\theta}^{(t)};z_{i,j})\right\|_2^2$$
$$\leq 2L^2\left\|\theta_j^{(t)} - \bar{\theta}^{(t)}\right\|_2^2 + 4L\ell(\bar{\theta}^{(t)}, z_{i,j}). \tag{B.27}$$

Plugging Eq. (B.26) and Eq. (B.27) into Theorem 4.1, we have

$$\frac{1}{mn}\sum_{i=1}^{n}\sum_{j=1}^{m}\mathbb{E}[\|A(S)-A(S^{(ij)})\|_2^2] \lesssim \frac{L^2}{mn}\sum_{t=0}^{T-1}\Big(\frac{L\eta_t}{(1-\lambda)^2}+\frac{1}{m}\Big)\eta_t^2\mathbb{E}\big[\frac{1}{m}\sum_{j=1}^{m}\|\theta_j^{(t)}-\bar{\theta}^{(t)}\|_2^2\big]$$

$$+\frac{L}{mn}\sum_{t=0}^{T-1}\Big(\frac{L\eta_t}{(1-\lambda)^2}+\frac{1}{m}\Big)\eta_t^2\mathbb{E}\big[\frac{1}{mn}\sum_{i=1}^{n}\sum_{j=1}^{m}\ell(\bar{\theta}^{(t)};z_{i,j})\big]$$

$$+\frac{L^2\sum_{t=0}^{T-1}\eta_t^2}{m^2n^2}\sum_{t=0}^{T-1}\mathbb{E}\big[\frac{1}{m}\sum_{j=1}^{m}\|\theta_j^{(t)}-\bar{\theta}^{(t)}\|_2^2\big]$$

$$+\frac{L\sum_{t=0}^{T-1}\eta_t^2}{m^2n^2}\sum_{t=0}^{T-1}\mathbb{E}\big[\frac{1}{mn}\sum_{i=1}^{n}\sum_{j=1}^{m}\ell(\bar{\theta}^{(t)};z_{i,j})\big]. \tag{B.28}$$

From Lemma B.5, we know

$$\mathbb{E}\big[\frac{1}{m}\sum_{j=1}^{m}\|\theta_j^{(t)}-\bar{\theta}^{(t)}\|_2^2\big]=\frac{1}{m}\mathbb{E}[\|\Theta^{(t)}-\bar{\Theta}^{(t)}\|_F^2]\leq \alpha_2 L\sum_{\tau=1}^{t-1}\alpha_1^{\tau-1}\eta_{t-\tau}^2\mathbb{E}[R_S(\bar{\theta}^{(t-\tau)})],$$

from which we know

$$\sum_{t=0}^{T-1}\Big(\frac{L\eta_t}{(1-\lambda)^2}+\frac{1}{m}\Big)\eta_t^2\mathbb{E}\big[\frac{1}{m}\sum_{j=1}^{m}\|\theta_j^{(t)}-\bar{\theta}^{(t)}\|_2^2\big]$$

$$\leq \sum_{t=0}^{T-1}\Big(\frac{L\eta_t}{(1-\lambda)^2}+\frac{1}{m}\Big)\eta_t^2\alpha_2 L\sum_{\tau=1}^{t-1}\alpha_1^{\tau-1}\eta_{t-\tau}^2\mathbb{E}[R_S(\bar{\theta}^{(t-\tau)})]$$

$$\leq \alpha_2 L\sum_{\tau=1}^{t-1}\alpha_1^{\tau-1}\sum_{t=\tau+1}^{T-1}\Big(\frac{L\eta_t}{(1-\lambda)^2}+\frac{1}{m}\Big)\eta_t^2\eta_{t-\tau}^2\mathbb{E}[R_S(\bar{\theta}^{(t-\tau)})]$$

$$\leq \alpha_2 L\sum_{\tau=1}^{T-1}\alpha_1^{\tau-1}\sum_{t=0}^{T-\tau}\Big(\frac{L\eta_{t+\tau}}{(1-\lambda)^2}+\frac{1}{m}\Big)\eta_{t+\tau}^2\eta_t^2\mathbb{E}[R_S(\bar{\theta}^{(t)})]$$

$$\leq \alpha_2 L\sum_{\tau=1}^{T-1}\alpha_1^{\tau-1}\sum_{t=0}^{T-1}\Big(\frac{L\eta_{t+\tau}}{(1-\lambda)^2}+\frac{1}{m}\Big)\eta_{t+\tau}^2\eta_t^2\mathbb{E}[R_S(\bar{\theta}^{(t)})]$$

$$\leq \alpha_2 L\sum_{\tau=1}^{T-1}\alpha_1^{\tau-1}\sum_{t=0}^{T-1}\Big(\frac{L\eta_t}{(1-\lambda)^2}+\frac{1}{m}\Big)\eta_t^4\mathbb{E}[R_S(\bar{\theta}^{(t)})]$$

$$\leq \frac{\alpha_2 L}{1-\alpha_1}\sum_{t=0}^{T-1}\Big(\frac{L\eta_t}{(1-\lambda)^2}+\frac{1}{m}\Big)\eta_t^4\mathbb{E}[R_S(\bar{\theta}^{(t)})]. \tag{B.29}$$

In a similar way, we can show

$$\sum_{t=0}^{T-1}\mathbb{E}\big[\frac{1}{m}\sum_{j=1}^{m}\|\theta_j^{(t)}-\bar{\theta}^{(t)}\|_2^2\big]\leq \frac{\alpha_2 L}{1-\alpha_1}\sum_{t=0}^{T-1}\eta_t^2\mathbb{E}[R_S(\bar{\theta}^{(t)})]. \tag{B.30}$$

Plugging Eq. (B.29), Eq. (B.30) into Eq. (B.28) and using the identity $\frac{1}{mn}\sum_{i=1}^{n}\sum_{j=1}^{m}\ell(\bar{\theta}^{(t)}; z_{i,j}) = R_S(\bar{\theta}^{(t)})$, we get

$$\frac{1}{mn}\sum_{i=1}^{n}\sum_{j=1}^{m}\mathbb{E}[\|A(S) - A(S^{(ij)})\|_2^2] \lesssim \frac{\alpha_2 L^3}{(1-\alpha_1)mn}\sum_{t=0}^{T-1}\left(\frac{L\eta_t}{(1-\lambda)^2} + \frac{1}{m}\right)\eta_t^4 \mathbb{E}[R_S(\bar{\theta}^{(t)})]$$

$$+ \frac{L}{mn}\sum_{t=0}^{T-1}\left(\frac{L\eta_t}{(1-\lambda)^2} + \frac{1}{m}\right)\eta_t^2 \mathbb{E}[R_S(\bar{\theta}^{(t)})]$$

$$+ \frac{\alpha_2 L^3 \sum_{t=0}^{T-1}\eta_t^2}{(1-\alpha_1)m^2 n^2}\sum_{t=0}^{T-1}\eta_t^2 \mathbb{E}[R_S(\bar{\theta}^{(t)})]$$

$$+ \frac{L\sum_{t=0}^{T-1}\eta_t^2}{m^2 n^2}\sum_{t=0}^{T-1}\mathbb{E}[R_S(\bar{\theta}^{(t)})].$$

If $\eta_t \leq \sqrt{\frac{1-\alpha_1}{L^2\alpha_2}}$, we know $\eta_t^2 \alpha_2 L^2/(1-\alpha_1) \leq 1$ and get

$$\frac{1}{mn}\sum_{i=1}^{n}\sum_{j=1}^{m}\mathbb{E}[\|A(S) - A(S^{(ij)})\|_2^2] \lesssim \frac{L}{mn}\sum_{t=0}^{T-1}\left(\frac{L\eta_t}{(1-\lambda)^2} + \frac{1}{m}\right)\eta_t^2 \mathbb{E}[R_S(\bar{\theta}^{(t)})] + \frac{L^2\sum_{t=0}^{T-1}\eta_t^2}{m^2 n^2}\sum_{t=0}^{T-1}\mathbb{E}[R_S(\bar{\theta}^{(t)})].$$

By choosing $\beta = \frac{1-\lambda}{2\lambda}$, we know

$$\frac{1-\alpha_1}{L^2\alpha_2} = \frac{1 - (1+\beta)\lambda + 2(1+\beta^{-1})\eta_1^2 L^2}{4L^2(1+\beta^{-1})} = \frac{1 - (1+\frac{1-\lambda}{2\lambda})\lambda + 2(1+\frac{2\lambda}{1-\lambda})\eta_1^2 L^2}{4L^2(1+\frac{2\lambda}{1-\lambda})}$$

$$= \frac{\frac{1-\lambda}{2} + \frac{2(1+\lambda)}{1-\lambda}\eta_1^2 L^2}{\frac{4L^2(1+\lambda)}{1-\lambda}} = \frac{(1-\lambda)^2}{8L^2(1+\lambda)} + \frac{\eta_1^2}{2}. \tag{B.31}$$

The proof is completed. $\qquad\square$

## B.2. Proof of Stability Bounds in Taheri and Thrampoulidis (2023)

According to Eq. (12) in the proof of Lemma 8 in Taheri and Thrampoulidis (2023) (with $\alpha = 1/2, c = \sqrt{2L}$), we know

$$\frac{1}{mn}\sum_{i=1}^{m}\sum_{j=1}^{n}\mathbb{E}[\|\bar{\theta}^{(T)} - \bar{\theta}^{(T,ij)}\|_2^2] \lesssim \frac{L\eta^2 T}{m^2 n^2}\sum_{t=0}^{T-1}\mathbb{E}[R_S(\bar{\theta}^{(t)})] + \frac{L^2\eta^2}{m}\left(\sum_{t=0}^{T-1}\|\Theta^{(t)} - \bar{\Theta}^{(t)}\|_F\right)^2.$$

From Lemma 11 there, we know

$$\left(\sum_{t=0}^{T-1}\|\Theta^{(t)} - \bar{\Theta}^{(t)}\|_F\right)^2 \leq T\sum_{t=0}^{T-1}\|\Theta^{(t)} - \bar{\Theta}^{(t)}\|_F^2 \leq T\frac{\tilde{\alpha}_2 L\eta^2 m}{1-\tilde{\alpha}_1}\sum_{t=0}^{T-1}\mathbb{E}[R_S(\bar{\theta}^{(t)})],$$

where $\tilde{\alpha}_1 = (3+\lambda)/4$, $\tilde{\alpha}_2 = 4(2/(1-\lambda) - 1)$. Combining the above two inequalities gives

$$\frac{1}{mn}\sum_{i=1}^{m}\sum_{j=1}^{n}\mathbb{E}[\|\bar{\theta}^{(T)} - \bar{\theta}^{(T,ij)}\|_2^2] \lesssim \frac{L\eta^2 T}{m^2 n^2}\sum_{t=0}^{T-1}\mathbb{E}[R_S(\bar{\theta}^{(t)})] + \frac{\tilde{\alpha}_2 L^3 \eta^4 T}{1-\tilde{\alpha}_1}\sum_{t=0}^{T-1}\mathbb{E}[R_S(\bar{\theta}^{(t)})]$$

$$\lesssim \frac{L\eta^2 T}{m^2 n^2}\sum_{t=0}^{T-1}\mathbb{E}[R_S(\bar{\theta}^{(t)})] + \frac{L^3 \eta^4 T}{(1-\lambda)^2}\sum_{t=0}^{T-1}\mathbb{E}[R_S(\bar{\theta}^{(t)})],$$

where we have used the fact that

$$\frac{\tilde{\alpha}_2}{1-\tilde{\alpha}_1} = \frac{4(\frac{2}{1-\lambda} - 1)}{1 - \frac{3+\lambda}{4}} = \frac{\frac{4(1+\lambda)}{1-\lambda}}{\frac{1-\lambda}{4}} = \frac{16(1+\lambda)}{(1-\lambda)^2}.$$

## B.3. Proofs on Convergence Analysis

In this subsection, we study the convergence analysis of D-SGD. Our analysis is motivated by Taheri and Thrampoulidis (2023), who derived convergence rates for DGD. We adapt their analysis to D-SGD, which is computationally efficient for large-scale problems. Following the standard analysis of decentralized optimization, we first present a lemma to control the distance between local models and their average.

**Lemma B.5.** *Assume the loss function $\ell(\cdot; z)$ is L-smooth and Assumption 3.1 holds. Let $\Theta^{(t)}$ and $\bar{\Theta}^{(t)}$ be defined in Eq. (B.1). Assume that $\theta_l^{(0)}$ are the same for different $l \in [m]$. Let $\alpha_1, \alpha_2$ be defined in Eq. (B.3) with $\alpha_1 < 1$. Then, there holds*

$$\mathbb{E}[\|\Theta^{(t)} - \bar{\Theta}^{(t)}\|_F^2] \leq \alpha_2 Lm \sum_{\tau=1}^{t-1} \alpha_1^{\tau-1} \eta_{t-\tau}^2 \mathbb{E}[R_S(\bar{\theta}^{(t-\tau)})].$$

*Proof.* Recall $W^\infty$ defined in Eq. (B.4). For any $\beta > 0$, we have

$$\begin{aligned}
\|\Theta^{(t)} - \bar{\Theta}^{(t)}\|_F^2 &= \|\Theta^{(t)} - \bar{\Theta}^{(t-1)} - \bar{\Theta}^{(t)} + \bar{\Theta}^{(t-1)}\|_F^2 \leq \|\Theta^{(t)} - \bar{\Theta}^{(t-1)}\|_F^2 \\
&= \|W\Theta^{(t-1)} - \eta_{t-1}\nabla\ell(\Theta^{(t-1)}; \mathbf{z}_{I^{t-1}}) - \bar{\Theta}^{(t-1)}\|_F^2 \\
&\leq (1+\beta)\|W\Theta^{(t-1)} - \bar{\Theta}^{(t-1)}\|_F^2 + (1+\frac{1}{\beta})\eta_{t-1}^2\|\nabla\ell(\Theta^{(t-1)}; \mathbf{z}_{I^{t-1}})\|_F^2,
\end{aligned} \quad (B.32)$$

where the first inequality holds due to $\|\Theta - \bar{\Theta}\|_F \leq \|\Theta\|_F$ for all $\Theta \in \mathbb{R}^{m \times d}$ (Taheri and Thrampoulidis, 2023), and the last inequality holds due to $\|a + b\|^2 \leq (1+\beta)\|a\|^2 + (1 + \frac{1}{\beta})\|b\|^2$. For any $\theta \in \mathbb{R}^{m \times d}$, Eq. (B.5) shows that $\|W\Theta - \bar{\Theta}\|_F^2 \leq \lambda\|\Theta - \bar{\Theta}\|_F^2$. Applying this to Eq. (B.32), we know

$$\|\Theta^{(t)} - \bar{\Theta}^{(t)}\|_F^2 \leq (1+\beta)\lambda\|\Theta^{(t-1)} - \bar{\Theta}^{(t-1)}\|_F^2 + (1+\frac{1}{\beta})\eta_{t-1}^2\|\nabla\ell(\Theta^{(t-1)}; \mathbf{z}_{I^{t-1}})\|_F^2. \quad (B.33)$$

By the $L$-smoothness of $\ell(\cdot, z)$ and the self-bounding property, we get

$$\begin{aligned}
\|\nabla\ell(\Theta^{(t-1)}; \mathbf{z}_{I^{t-1}})\|_F^2 &= \|\nabla\ell(\Theta^{(t-1)}; \mathbf{z}_{I^{t-1}}) - \nabla\ell(\bar{\Theta}^{(t-1)}; \mathbf{z}_{I^{t-1}}) + \nabla\ell(\bar{\Theta}^{(t-1)}; \mathbf{z}_{I^{t-1}})\|_F^2 \\
&\leq 2\|\nabla\ell(\Theta^{(t-1)}; \mathbf{z}_{I^{t-1}}) - \nabla\ell(\bar{\Theta}^{(t-1)}; \mathbf{z}_{I^{t-1}})\|_F^2 + 2\|\nabla\ell(\bar{\Theta}^{(t-1)}; \mathbf{z}_{I^{t-1}})\|_F^2 \\
&\leq 2L^2\|\Theta^{(t-1)} - \bar{\Theta}^{(t-1)}\|_F^2 + 4L\sum_{k=1}^m \ell(\bar{\theta}^{(t-1)}; z_{I_k^{t-1},k}).
\end{aligned} \quad (B.34)$$

Taking expectation to both sides of Eq. (B.34), we have

$$\begin{aligned}
\mathbb{E}[\|\nabla\ell(\Theta^{(t-1)}; \mathbf{z}_{I^{t-1}})\|_F^2] &\leq 2L^2\mathbb{E}[\|\Theta^{(t-1)} - \bar{\Theta}^{(t-1)}\|_F^2] + 4L\mathbb{E}[\sum_{k=1}^m R_{S_k}(\bar{\theta}^{(t-1)})] \\
&= 2L^2\mathbb{E}[\|\Theta^{(t-1)} - \bar{\Theta}^{(t-1)}\|_F^2] + 4Lm\mathbb{E}[R_S(\bar{\theta}^{(t-1)})],
\end{aligned} \quad (B.35)$$

where the first inequality holds due to the independence of $\bar{\theta}^{(t-1)}$ and $I_k^{t-1}$, and we use the fact that $R_S(\cdot) = \frac{1}{m}\sum_{k=1}^m R_{S_k}(\cdot)$ for the last equality. Taking expectation to both sides of Eq. (B.33) and plugging Eq. (B.35) into it, we have

$$\mathbb{E}[\|\Theta^{(t)} - \bar{\Theta}^{(t)}\|_F^2] \leq \left((1+\beta)\lambda + 2(1+\frac{1}{\beta})L^2\eta_{t-1}^2\right)\mathbb{E}[\|\Theta^{(t-1)} - \bar{\Theta}^{(t-1)}\|_F^2] + 4(1+\frac{1}{\beta})\eta_{t-1}^2 Lm\mathbb{E}[R_S(\bar{\theta}^{(t-1)})].$$

By the definition of $\alpha_1, \alpha_2$ in Eq. (B.3) and the assumption that the step size is nonincreasing, we further get

$$\mathbb{E}[\|\Theta^{(t)} - \bar{\Theta}^{(t)}\|_F^2] \leq \alpha_1 \mathbb{E}[\|\Theta^{(t-1)} - \bar{\Theta}^{(t-1)}\|_F^2] + \alpha_2\eta_{t-1}^2 Lm\mathbb{E}[R_S(\bar{\theta}^{(t-1)})]. \quad (B.36)$$

Applying Eq. (B.36) repeatedly and using the assumption that $\theta_l^{(0)}$ are the same for $l \in [m]$, we have

$$\mathbb{E}[\|\Theta^{(t)} - \bar{\Theta}^{(t)}\|_F^2] \leq \alpha_2 Lm \sum_{\tau=1}^{t-1} \alpha_1^{\tau-1} \eta_{t-\tau}^2 \mathbb{E}[R_S(\bar{\theta}^{(t-\tau)})]. \quad (B.37)$$

The proof is completed. $\qquad\square$

*Proof of Theorem 4.6.* Let $\Theta^{(t)}$ and $\bar{\Theta}^{(t)}$ be defined in Eq. (B.1). Firstly, we develop a bound for $\frac{1}{mT}\sum_{t=1}^{T}\mathbb{E}[\|\Theta^{(t)} - \bar{\Theta}^{(t)}\|_F^2]$. According to Lemma B.5, we know

$$
\frac{1}{mT}\sum_{t=1}^{T}\mathbb{E}[\|\Theta^{(t)} - \bar{\Theta}^{(t)}\|_F^2] \leq \frac{\alpha_2 L}{T}\sum_{t=2}^{T}\sum_{\tau=1}^{t-1}\alpha_1^{\tau-1}\eta_{t-\tau}^2\mathbb{E}[R_S(\bar{\theta}^{(t-\tau)})] \leq \frac{\alpha_2 L}{T}\sum_{\tau=1}^{T-1}\alpha_1^{\tau-1}\sum_{t=\tau+1}^{T}\eta_{t-\tau}^2\mathbb{E}[R_S(\bar{\theta}^{(t-\tau)})]
$$

$$
\leq \frac{\alpha_2 L}{T}\sum_{\tau=1}^{T-1}\alpha_1^{\tau-1}\sum_{t=1}^{T-\tau}\eta_t^2\mathbb{E}[R_S(\bar{\theta}^{(t)})] \leq \frac{\alpha_2 L}{T}\sum_{\tau=1}^{T-1}\alpha_1^{\tau-1}\sum_{t=1}^{T-1}\eta_t^2\mathbb{E}[R_S(\bar{\theta}^{(t)})]
$$

$$
\leq \frac{\alpha_2 L}{(1-\alpha_1)T}\sum_{t=1}^{T-1}\eta_t^2\mathbb{E}[R_S(\bar{\theta}^{(t)})]. \tag{B.38}
$$

Secondly, for any $\theta \in \mathbb{R}^d$, there holds

$$
\|\bar{\theta}^{(t+1)} - \theta\|_2^2 = \left\|\bar{\theta}^{(t)} - \eta_t\frac{1}{m}\sum_{k=1}^{m}\nabla\ell(\theta_k^{(t)}; z_{I_k^t,k}) - \theta\right\|_2^2
$$

$$
= \|\bar{\theta}^{(t)} - \theta\|_2^2 + \eta_t^2\left\|\frac{1}{m}\sum_{k=1}^{m}\nabla\ell(\theta_k^{(t)}; z_{I_k^t,k})\right\|_2^2 - 2\eta_t\left\langle\bar{\theta}^{(t)} - \theta, \frac{1}{m}\sum_{k=1}^{m}\nabla\ell(\theta_k^{(t)}; z_{I_k^t,k})\right\rangle. \tag{B.39}
$$

By the Cauchy-Schwarz's inequality, we know

$$
\left\|\frac{1}{m}\sum_{k=1}^{m}\nabla\ell(\theta_k^{(t)}; z_{I_k^t,k})\right\|_2^2 \leq 2\left\|\frac{1}{m}\sum_{k=1}^{m}\left(\nabla\ell(\theta_k^{(t)}; z_{I_k^t,k}) - \nabla\ell(\bar{\theta}^{(t)}; z_{I_k^t,k})\right)\right\|_2^2 + 2\left\|\frac{1}{m}\sum_{k=1}^{m}\nabla\ell(\bar{\theta}^{(t)}; z_{I_k^t,k})\right\|_2^2
$$

$$
\leq \frac{2}{m}\sum_{k=1}^{m}\left\|\nabla\ell(\theta_k^{(t)}; z_{I_k^t,k}) - \nabla\ell(\bar{\theta}^{(t)}; z_{I_k^t,k})\right\|_2^2 + \frac{2}{m}\sum_{k=1}^{m}\left\|\nabla\ell(\bar{\theta}^{(t)}; z_{I_k^t,k})\right\|_2^2
$$

$$
\leq \frac{2L^2}{m}\sum_{k=1}^{m}\|\theta_k^{(t)} - \bar{\theta}^{(t)}\|_2^2 + \frac{4L}{m}\sum_{k=1}^{m}\ell(\bar{\theta}^{(t)}; z_{I_k^t,k})
$$

$$
= \frac{2L^2}{m}\|\Theta^{(t)} - \bar{\Theta}^{(t)}\|_F^2 + \frac{4L}{m}\sum_{k=1}^{m}\ell(\bar{\theta}^{(t)}; z_{I_k^t,k}), \tag{B.40}
$$

where the third inequality holds due to the $L$-smoothness of $\ell(\cdot, z)$ and the self-bounding property (Lemma A.1). Taking expectation to both sides of Eq. (B.40), we have

$$
\mathbb{E}\left[\left\|\frac{1}{m}\sum_{k=1}^{m}\nabla\ell(\theta_k^{(t)}; z_{I_k^t,k})\right\|_2^2\right] \leq \frac{2L^2}{m}\mathbb{E}[\|\Theta^{(t)} - \bar{\Theta}^{(t)}\|_F^2] + 4L\mathbb{E}[R_S(\bar{\theta}^{(t)})], \tag{B.41}
$$

where we use the fact that $R_S(\cdot) = \frac{1}{m}\sum_{k=1}^{m}R_{S_k}(\cdot)$. Note

$$
\left\langle\bar{\theta}^{(t)} - \theta, \frac{1}{m}\sum_{k=1}^{m}\nabla\ell(\theta_k^{(t)}; z_{I_k^t,k})\right\rangle = \frac{1}{m}\sum_{k=1}^{m}\left\langle\bar{\theta}^{(t)} - \theta, \nabla\ell(\theta_k^{(t)}; z_{I_k^t,k})\right\rangle
$$

$$
= \frac{1}{m}\sum_{k=1}^{m}\left\langle\bar{\theta}^{(t)} - \theta_k^{(t)}, \nabla\ell(\theta_k^{(t)}; z_{I_k^t,k})\right\rangle + \frac{1}{m}\sum_{k=1}^{m}\left\langle\theta_k^{(t)} - \theta, \nabla\ell(\theta_k^{(t)}; z_{I_k^t,k})\right\rangle
$$

$$
\geq \frac{1}{m}\sum_{k=1}^{m}\left(\ell(\bar{\theta}^{(t)}; z_{I_k^t,k}) - \ell(\theta_k^{(t)}; z_{I_k^t,k}) - \frac{L}{2}\|\theta_k^{(t)} - \bar{\theta}^{(t)}\|_2^2\right) + \frac{1}{m}\sum_{k=1}^{m}\left(\ell(\theta_k^{(t)}; z_{I_k^t,k}) - \ell(\theta; z_{I_k^t,k})\right)
$$

$$
= \frac{1}{m}\sum_{k=1}^{m}\left(\ell(\bar{\theta}^{(t)}; z_{I_k^t,k}) - \ell(\theta; z_{I_k^t,k}) - \frac{L}{2}\|\theta_k^{(t)} - \bar{\theta}^{(t)}\|_2^2\right), \tag{B.42}
$$

where the third inequality holds due to the $L$-smoothness and convexity of $\ell(\cdot, z)$. Taking expectation on both sides of Eq. (B.42), we have

$$\mathbb{E}\Big[\Big\langle \bar{\theta}^{(t)} - \theta, \frac{1}{m}\sum_{k=1}^{m}\nabla\ell(\theta_k^{(t)}; z_{I_k^t, k})\Big\rangle\Big] \geq \mathbb{E}[R_S(\bar{\theta}^{(t)})] - \mathbb{E}[R_S(\theta)] - \frac{L}{2m}\mathbb{E}[\|\Theta^{(t)} - \bar{\Theta}^{(t)}\|_F^2]. \tag{B.43}$$

Taking expectation to both sides of Eq. (B.39) and plugging Eq. (B.41) and Eq. (B.43) into it, we have

$$\mathbb{E}[\|\bar{\theta}^{(t+1)} - \theta\|_2^2]$$
$$\leq \mathbb{E}[\|\bar{\theta}^{(t)} - \theta\|_2^2] + \frac{2L^2\eta_t^2 + L\eta_t}{m}\mathbb{E}[\|\Theta^{(t)} - \bar{\Theta}^{(t)}\|_2^2] + (4L\eta_t^2 - 2\eta_t)\mathbb{E}[R_S(\bar{\theta}^{(t)})] + 2\eta_t\mathbb{E}[R_S(\theta)]$$
$$= \mathbb{E}[\|\bar{\theta}^{(t)} - \theta\|_2^2] + \frac{2L^2\eta_t^2 + L\eta_t}{m}\mathbb{E}[\|\Theta^{(t)} - \bar{\Theta}^{(t)}\|_2^2] + 4L\eta_t^2\mathbb{E}[R_S(\bar{\theta}^{(t)})] + 2\eta_t\mathbb{E}[R_S(\theta) - R_S(\bar{\theta}^{(t)})].$$

Taking a summation of the above inequality gives

$$\frac{1}{T}\sum_{t=1}^{T}\mathbb{E}[R_S(\bar{\theta}^{(t)}) - R_S(\theta)]$$
$$\leq \frac{\mathbb{E}[\|\bar{\theta}^{(1)} - \theta\|_2^2]}{2\eta T} + \frac{2L^2\eta + L}{2mT}\sum_{t=1}^{T}\mathbb{E}[\|\Theta^{(t)} - \bar{\Theta}^{(t)}\|_2^2] + \frac{2L\eta}{T}\sum_{t=1}^{T}\mathbb{E}[R_S(\bar{\theta}^{(t)})]$$
$$\leq \frac{\mathbb{E}[\|\bar{\theta}^{(1)} - \theta\|_2^2]}{2\eta T} + \frac{L}{mT}\sum_{t=1}^{T}\mathbb{E}[\|\Theta^{(t)} - \bar{\Theta}^{(t)}\|_2^2] + \frac{2L\eta}{T}\sum_{t=1}^{T}\mathbb{E}[R_S(\bar{\theta}^{(t)})]$$
$$\leq \frac{\mathbb{E}[\|\bar{\theta}^{(1)} - \theta\|_2^2]}{2\eta T} + \frac{\alpha_2 L^2\eta^2}{(1-\alpha_1)T}\sum_{t=1}^{T}\mathbb{E}[R_S(\bar{\theta}^{(t)})] + \frac{2L\eta}{T}\sum_{t=1}^{T}\mathbb{E}[R_S(\bar{\theta}^{(t)})]$$
$$= \frac{\mathbb{E}[\|\bar{\theta}^{(1)} - \theta\|_2^2]}{2\eta T} + \Big(\frac{\alpha_2 L^2\eta^2}{(1-\alpha_1)} + 2L\eta\Big)\Big(\frac{1}{T}\sum_{t=1}^{T}\mathbb{E}[R_S(\bar{\theta}^{(t)})] - \mathbb{E}[R_S(\theta)]\Big) + \Big(\frac{\alpha_2 L^2\eta^2}{(1-\alpha_1)} + 2L\eta\Big)\mathbb{E}[R_S(\theta)],$$

where we have used the assumption that $\eta \leq 1/(2L)$ for the first inequality and Eq. (B.38). By Eq. (B.31), we know

$$\frac{1-\alpha_1}{L^2\alpha_2} = \frac{(1-\lambda)^2}{8L^2(1+\lambda)} + \frac{\eta_1^2}{2} \geq \frac{(1-\lambda)^2}{8L^2(1+\lambda)}.$$

Therefore, it follows from Eq. (4.11) that

$$\frac{\alpha_2 L^2\eta^2}{(1-\alpha_1)} + 2L\eta \leq \frac{8L^2(1+\lambda)\eta^2}{(1-\lambda)^2} + 2L\eta \leq \frac{1}{2}. \tag{B.44}$$

We now consider two cases. If $\mathbb{E}[R_S(\theta)] \leq \frac{1}{T}\sum_{t=1}^{T}\mathbb{E}[R_S(\bar{\theta}^{(t)})]$, then we combine the above discussions to derive that

$$\frac{1}{T}\sum_{t=1}^{T}\mathbb{E}[R_S(\bar{\theta}^{(t)}) - R_S(\theta)] \leq \frac{\mathbb{E}[\|\bar{\theta}^{(1)} - \theta\|_2^2]}{2\eta T} + \frac{1}{2T}\sum_{t=1}^{T}\mathbb{E}[R_S(\bar{\theta}^{(t)}) - R_S(\theta)] + \Big(\frac{\alpha_2 L^2\eta^2}{(1-\alpha_1)} + 2L\eta\Big)\mathbb{E}[R_S(\theta)],$$

from which and Eq. (B.44) we get

$$\frac{1}{T}\sum_{t=1}^{T}\mathbb{E}[R_S(\bar{\theta}^{(t)}) - R_S(\theta)] \leq \frac{\mathbb{E}[\|\bar{\theta}^{(1)} - \theta\|_2^2]}{\eta T} + 2\Big(\frac{\alpha_2 L^2\eta^2}{(1-\alpha_1)} + 2L\eta\Big)\mathbb{E}[R_S(\theta)]$$
$$\leq \frac{\mathbb{E}[\|\bar{\theta}^{(1)} - \theta\|_2^2]}{\eta T} + \Big(\frac{16L^2(1+\lambda)\eta^2}{(1-\lambda)^2} + 4L\eta\Big)\mathbb{E}[R_S(\theta)].$$

If $\mathbb{E}[R_S(\theta)] > \frac{1}{T}\sum_{t=1}^{T}\mathbb{E}[R_S(\bar{\theta}^{(t)})]$, the above inequality still holds. The proof is completed. $\square$

### B.4. Proofs on Excess Risk Analysis

In this subsection, we present the proof of Theorem 4.8 on excess risk bounds.

*Proof of Theorem 4.8.* By Lemma 3.5 and Eq. (4.5), we know

$$\mathbb{E}_{S,A}[R(A(S)) - R_S(A(S))] \lesssim \frac{L}{\gamma}\mathbb{E}[R_S(\bar{\theta}^{(T)})] + \frac{(L+\gamma)L}{mn}\sum_{t=0}^{T-1}\Big(\frac{L\eta_t}{(1-\lambda)^2} + \frac{1}{m}\Big)\eta_t^2\mathbb{E}\big[R_S(\bar{\theta}^{(t)})\big]$$
$$+ \frac{(L+\gamma)L^2\sum_{t=0}^{T-1}\eta_t^2}{m^2n^2}\sum_{t=0}^{T-1}\mathbb{E}\big[R_S(\bar{\theta}^{(t)})\big].$$

By noting that $R_S(\bar{\theta}^{(T)}) \leq \frac{1}{T}\sum_{t=0}^{T-1}R_S(\bar{\theta}^{(t)})$ and choosing a constant step size $\eta_t = \eta$, this further implies

$$\mathbb{E}_{S,A}[R(A(S)) - R_S(A(S))] \lesssim \Big(\frac{L}{\gamma T} + \frac{(L+\gamma)L\big(\frac{L\eta}{(1-\lambda)^2} + \frac{1}{m}\big)\eta^2}{mn} + \frac{(L+\gamma)L^2\eta^2T}{m^2n^2}\Big)\sum_{t=0}^{T-1}\mathbb{E}\big[R_S(\bar{\theta}^{(t)})\big]. \quad \text{(B.45)}$$

Eq. (4.12) and Eq. (4.11) imply that

$$\frac{1}{T}\sum_{t=0}^{T-1}\mathbb{E}\big[R_S(\bar{\theta}^{(t)})\big] \lesssim R(\theta^*) + \frac{\|\theta^*\|_2^2}{\eta T}. \quad \text{(B.46)}$$

We combine Eq. (B.45) and Eq. (B.46) together, and derive

$$\mathbb{E}_{S,A}[R(A(S)) - R_S(A(S))] \lesssim \Big(R(\theta^*) + \frac{\|\theta^*\|_2^2}{\eta T}\Big)\Big(\frac{L}{\gamma} + \frac{T(L+\gamma)L\big(\frac{L\eta}{(1-\lambda)^2} + \frac{1}{m}\big)\eta^2}{mn} + \frac{(L+\gamma)L^2\eta^2T^2}{m^2n^2}\Big). \quad \text{(B.47)}$$

Plugging Eq. (B.47) and Eq. (4.12) into Eq. (3.1), we have

$$\mathbb{E}[R(A(S))] - R(\theta^*) \lesssim \Big(R(\theta^*) + \frac{\|\theta^*\|_2^2}{\eta T}\Big)\Big(\frac{L}{\gamma} + \frac{T(L+\gamma)L\big(\frac{L\eta}{(1-\lambda)^2} + \frac{1}{m}\big)\eta^2}{mn} + \frac{(L+\gamma)L^2\eta^2T^2}{m^2n^2}\Big)$$
$$+ \frac{\|\theta^*\|_2^2}{\eta T} + \Big(\frac{L^2\eta^2}{(1-\lambda)^2} + L\eta\Big)\mathbb{E}[R_S(\theta^*)].$$

The proof is completed. □

## C. Proofs for Convex and Nonsmooth Problems

### C.1. Proofs on Stability Analysis

In this subsection, we present the proof on the stability analysis of D-SGD for convex and nonsmooth problems. An useful inequality for our analysis is the Young's inequality

$$ab \leq p^{-1}|a|^p + q^{-1}|b|^q, \quad a, b \in \mathbb{R}, p, q > 0 \text{ with } p^{-1} + q^{-1} = 1. \quad \text{(C.1)}$$

*Proof of Theorem 5.1.* Eq. (B.9) shows that

$$\|\bar{\theta}^{(t+1)} - \bar{\theta}^{(t+1,ij)}\|_2^2 = \|\bar{\theta}^{(t)} - \bar{\theta}^{(t,ij)}\|_2^2 + \frac{\eta_t^2}{m^2}\Big\|\sum_{k=1}^m \big(\nabla\ell(\theta_k^{(t)}; Z_{I_k^t,k}) - \nabla\ell(\theta_k^{(t,ij)}; \tilde{Z}_{I_k^t,k}^{(ij)})\big)\Big\|_2^2$$
$$- \frac{2\eta_t}{m}\sum_{k=1}^m \big\langle\bar{\theta}^{(t)} - \bar{\theta}^{(t,ij)}, \nabla\ell(\theta_k^{(t)}; Z_{I_k^t,k}) - \nabla\ell(\theta_k^{(t,ij)}; \tilde{Z}_{I_k^t,k}^{(ij)})\big\rangle. \quad \text{(C.2)}$$

**Part 1: For the first term in Eq.** (C.2), Eq. (B.10) shows that for any $\beta_1 > 0$,

$$\big\|\sum_{k=1}^{m}\big(\nabla\ell(\theta_k^{(t)};Z_{I_k^t,k})-\nabla\ell(\theta_k^{(t,ij)};\tilde{Z}_{I_k^t,k}^{(ij)})\big)\big\|_2^2 \leq (1+\beta_1)(m-1)\sum_{k\neq j}\big\|\big(\nabla\ell(\theta_k^{(t)};Z_{I_k^t,k})-\nabla\ell(\theta_k^{(t,ij)};\tilde{Z}_{I_k^t,k}^{(ij)})\big)\big\|_2^2$$

$$+ (1+\frac{1}{\beta_1})\big\|\nabla\ell(\theta_j^{(t)};Z_{I_j^t,j})-\nabla\ell(\theta_j^{(t,ij)};\tilde{Z}_{I_j^t,j}^{(ij)})\big\|_2^2. \quad \text{(C.3)}$$

**Part 2: For the last term in Eq.** (C.2), we first consider two cases to control

$$\big\langle \bar{\theta}^{(t)}-\bar{\theta}^{(t,ij)},\nabla\ell(\theta_k^{(t)};Z_{I_k^t,k})-\nabla\ell(\theta_k^{(t,ij)};\tilde{Z}_{I_k^t,k}^{(ij)})\big\rangle.$$

(1) If $k=j$ and $I_j^t=i$, then it is clear

$$\big\langle \bar{\theta}^{(t)}-\bar{\theta}^{(t,ij)},\nabla\ell(\theta_j^{(t)};z_{i,j})-\nabla\ell(\theta_j^{(t,ij)};\tilde{z}_{i,j})\big\rangle \geq -\big\|\bar{\theta}^{(t)}-\bar{\theta}^{(t,ij)}\big\|_2\big\|\nabla\ell(\theta_j^{(t)};z_{i,j})-\nabla\ell(\theta_j^{(t,ij)};\tilde{z}_{i,j})\big\|_2. \quad \text{(C.4)}$$

(2) If ($k=j$ and $I_j^t\neq i$) or if $k\neq j$, then

$$\big\langle \bar{\theta}^{(t)}-\bar{\theta}^{(t,ij)},\nabla\ell(\theta_k^{(t)};Z_{I_k^t,k})-\nabla\ell(\theta_k^{(t,ij)};\tilde{Z}_{I_k^t,k}^{(ij)})\big\rangle = \big\langle \theta_k^{(t)}-\theta_k^{(t,ij)},\nabla\ell(\theta_k^{(t)};Z_{I_k^t,k})-\nabla\ell(\theta_k^{(t,ij)};\tilde{Z}_{I_k^t,k}^{(ij)})\big\rangle$$

$$+ \big\langle \bar{\theta}^{(t)}-\theta_k^{(t)}-\bar{\theta}^{(t,ij)}+\theta_k^{(t,ij)},\nabla\ell(\theta_k^{(t)};Z_{I_k^t,k})-\nabla\ell(\theta_k^{(t,ij)};\tilde{Z}_{I_k^t,k}^{(ij)})\big\rangle. \quad \text{(C.5)}$$

The convexity and $(\alpha,L)$-Hölder continuity imply that the gradients are co-coercive (Lemma B.2), namely,

$$\big\langle \theta_k^{(t)}-\theta_k^{(t,ij)},\nabla\ell(\theta_k^{(t)};Z_{I_k^t,k})-\nabla\ell(\theta_k^{(t,ij)};\tilde{Z}_{I_k^t,k}^{(ij)})\big\rangle \geq \frac{2L^{-\frac{1}{\alpha}}\alpha}{1+\alpha}\big\|\nabla\ell(\theta_k^{(t)};Z_{I_k^t,k})-\nabla\ell(\theta_k^{(t,ij)};\tilde{Z}_{I_k^t,k}^{(ij)})\big\|_2^{\frac{1+\alpha}{\alpha}},$$

which further yields the following inequality for any $b>0$

$$\big\|\nabla\ell(\theta_k^{(t)};Z_{I_k^t,k})-\nabla\ell(\theta_k^{(t,ij)};\tilde{Z}_{I_k^t,k}^{(ij)})\big\|_2^2$$

$$\leq \Big(\frac{1+\alpha}{2L^{-\frac{1}{\alpha}}\alpha}\big\langle \theta_k^{(t)}-\theta_k^{(t,ij)},\nabla\ell(\theta_k^{(t)};Z_{I_k^t,k})-\nabla\ell(\theta_k^{(t,ij)};\tilde{Z}_{I_k^t,k}^{(ij)})\big\rangle\Big)^{\frac{2\alpha}{1+\alpha}}$$

$$= \Big(\frac{L(1+\alpha)}{2b\eta_t\alpha}\big\langle \theta_k^{(t)}-\theta_k^{(t,ij)},\nabla\ell(\theta_k^{(t)};Z_{I_k^t,k})-\nabla\ell(\theta_k^{(t,ij)};\tilde{Z}_{I_k^t,k}^{(ij)})\big\rangle\Big)^{\frac{2\alpha}{1+\alpha}}\Big(b^{\frac{2\alpha}{1+\alpha}}\eta_t^{\frac{2\alpha}{1+\alpha}}L^{\frac{2(1-\alpha)}{1+\alpha}}\Big)$$

$$\leq \frac{L}{b\eta_t}\big\langle \theta_k^{(t)}-\theta_k^{(t,ij)},\nabla\ell(\theta_k^{(t)};Z_{I_k^t,k})-\nabla\ell(\theta_k^{(t,ij)};\tilde{Z}_{I_k^t,k}^{(ij)})\big\rangle + \frac{1-\alpha}{1+\alpha}b^{\frac{2\alpha}{1-\alpha}}\eta_t^{\frac{2\alpha}{1-\alpha}}L^2,$$

where we use Young's inequality with $p=\frac{1+\alpha}{2\alpha}$ and $q=\frac{1+\alpha}{1-\alpha}$ for the last inequality above. Therefore,

$$\big\langle \theta_k^{(t)}-\theta_k^{(t,ij)},\nabla\ell(\theta_k^{(t)};Z_{I_k^t,k})-\nabla\ell(\theta_k^{(t,ij)};\tilde{Z}_{I_k^t,k}^{(ij)})\big\rangle \geq \frac{b\eta_t}{L}\big\|\nabla\ell(\theta_k^{(t)};Z_{I_k^t,k})-\nabla\ell(\theta_k^{(t,ij)};\tilde{Z}_{I_k^t,k}^{(ij)})\big\|_2^2 - \frac{1-\alpha}{(1+\alpha)}(b\eta_t)^{1+\frac{2\alpha}{1-\alpha}}L.$$

$$\text{(C.6)}$$

For any $\beta_2 > 0$, we know

$$\big\langle \bar{\theta}^{(t)}-\theta_k^{(t)}-\bar{\theta}^{(t,ij)}+\theta_k^{(t,ij)},\nabla\ell(\theta_k^{(t)};Z_{I_k^t,k})-\nabla\ell(\theta_k^{(t,ij)};\tilde{Z}_{I_k^t,k}^{(ij)})\big\rangle$$

$$\geq -\big\|\bar{\theta}^{(t)}-\theta_k^{(t)}-\bar{\theta}^{(t,ij)}+\theta_k^{(t,ij)}\big\|_2\big\|\nabla\ell(\theta_k^{(t)};Z_{I_k^t,k})-\nabla\ell(\theta_k^{(t,ij)};\tilde{Z}_{I_k^t,k}^{(ij)})\big\|_2$$

$$\geq -\frac{\beta_2}{2}\big\|\bar{\theta}^{(t)}-\theta_k^{(t)}-\bar{\theta}^{(t,ij)}+\theta_k^{(t,ij)}\big\|_2^2 - \frac{1}{2\beta_2}\big\|\nabla\ell(\theta_k^{(t)};Z_{I_k^t,k})-\nabla\ell(\theta_k^{(t,ij)};\tilde{Z}_{I_k^t,k}^{(ij)})\big\|_2^2. \quad \text{(C.7)}$$

Plugging Eq. (C.6) and Eq. (C.7) into Eq. (C.5), we have, for ($k=j$ and $I_j^t\neq i$) or $k\neq j$:

$$\big\langle \bar{\theta}^{(t)}-\bar{\theta}^{(t,ij)},\nabla\ell(\theta_k^{(t)};Z_{I_k^t,k})-\nabla\ell(\theta_k^{(t,ij)};\tilde{Z}_{I_k^t,k}^{(ij)})\big\rangle$$

$$\geq -\frac{\beta_2}{2}\big\|\bar{\theta}^{(t)}-\theta_k^{(t)}-\bar{\theta}^{(t,ij)}+\theta_k^{(t,ij)}\big\|_2^2 + \big(\frac{b\eta_t}{L}-\frac{1}{2\beta_2}\big)\big\|\nabla\ell(\theta_k^{(t)};Z_{I_k^t,k})-\nabla\ell(\theta_k^{(t,ij)};\tilde{Z}_{I_k^t,k}^{(ij)})\big\|_2^2 - \frac{1-\alpha}{1+\alpha}(b\eta_t)^{\frac{1+\alpha}{1-\alpha}}L.$$

$$\text{(C.8)}$$

Then, we combine the results above to control $-\frac{2\eta_t}{m}\sum_{k=1}^m \big\langle \bar{\theta}^{(t)} - \bar{\theta}^{(t,ij)}, \nabla\ell(\theta_k^{(t)}; Z_{I_k^t,k}) - \nabla\ell(\theta_k^{(t,ij)}; \tilde{Z}_{I_k^t,k}^{(ij)})\big\rangle$.

(1) If $I_j^t \neq i$, by plugging in Eq. (C.8), we have

$$
-\frac{2\eta_t}{m}\sum_{k=1}^m \big\langle \bar{\theta}^{(t)} - \bar{\theta}^{(t,ij)}, \nabla\ell(\theta_k^{(t)}; Z_{I_k^t,k}) - \nabla\ell(\theta_k^{(t,ij)}; \tilde{Z}_{I_k^t,k}^{(ij)})\big\rangle \leq \frac{\beta_2\eta_t}{m}\sum_{k=1}^m \big\| \bar{\theta}^{(t)} - \theta_k^{(t)} - \bar{\theta}^{(t,ij)} + \theta_k^{(t,ij)} \big\|_2^2
$$
$$
- \Big(\frac{2b\eta_t}{L} - \frac{1}{\beta_2}\Big)\frac{\eta_t}{m}\sum_{k=1}^m \big\| \nabla\ell(\theta_k^{(t)}; Z_{I_k^t,k}) - \nabla\ell(\theta_k^{(t,ij)}; \tilde{Z}_{I_k^t,k}^{(ij)}) \big\|_2^2 + \frac{2L(1-\alpha)}{1+\alpha} b^{\frac{1+\alpha}{1-\alpha}} \eta_t^{\frac{2}{1-\alpha}}. \quad (C.9)
$$

(2) If $I_j^t = i$, we know

$$
-\frac{2\eta_t}{m}\sum_{k=1}^m \big\langle \bar{\theta}^{(t)} - \bar{\theta}^{(t,ij)}, \nabla\ell(\theta_k^{(t)}; Z_{I_k^t,k}) - \nabla\ell(\theta_k^{(t,ij)}; \tilde{Z}_{I_k^t,k}^{(ij)})\big\rangle
$$
$$
= -\frac{2\eta_t}{m}\sum_{k\neq j} \big\langle \bar{\theta}^{(t)} - \bar{\theta}^{(t,ij)}, \nabla\ell(\theta_k^{(t)}; Z_{I_k^t,k}) - \nabla\ell(\theta_k^{(t,ij)}; \tilde{Z}_{I_k^t,k}^{(ij)})\big\rangle - \frac{2\eta_t}{m}\big\langle \bar{\theta}^{(t)} - \bar{\theta}^{(t,ij)}, \nabla\ell(\theta_j^{(t)}; Z_{I_j^t,j}) - \nabla\ell(\theta_j^{(t,ij)}; \tilde{Z}_{I_j^t,j}^{(ij)})\big\rangle
$$
$$
\leq -\frac{2\eta_t}{m}\sum_{k\neq j} \big\langle \bar{\theta}^{(t)} - \bar{\theta}^{(t,ij)}, \nabla\ell(\theta_k^{(t)}; Z_{I_k^t,k}) - \nabla\ell(\theta_k^{(t,ij)}; \tilde{Z}_{I_k^t,k}^{(ij)})\big\rangle + \frac{2\eta_t}{m}\big\| \bar{\theta}^{(t)} - \bar{\theta}^{(t,ij)} \big\|_2 \big\| \nabla\ell(\theta_j^{(t)}; z_{i,j}) - \nabla\ell(\theta_j^{(t,ij)}; \tilde{z}_{i,j}) \big\|_2.
$$

From Eq. (C.8), we know

$$
-\frac{2\eta_t}{m}\sum_{k\neq j} \big\langle \bar{\theta}^{(t)} - \bar{\theta}^{(t,ij)}, \nabla\ell(\theta_k^{(t)}; Z_{I_k^t,k}) - \nabla\ell(\theta_k^{(t,ij)}; \tilde{Z}_{I_k^t,k}^{(ij)})\big\rangle \leq \frac{\beta_2\eta_t}{m}\sum_{k=1}^m \big\| \bar{\theta}^{(t)} - \theta_k^{(t)} - \bar{\theta}^{(t,ij)} + \theta_k^{(t,ij)} \big\|_2^2
$$
$$
- \Big(\frac{2b\eta_t}{L} - \frac{1}{\beta_2}\Big)\frac{\eta_t}{m}\sum_{k\neq j} \big\| \nabla\ell(\theta_k^{(t)}; Z_{I_k^t,k}) - \nabla\ell(\theta_k^{(t,ij)}; \tilde{Z}_{I_k^t,k}^{(ij)}) \big\|_2^2 + \frac{2L(1-\alpha)}{1+\alpha} b^{\frac{1+\alpha}{1-\alpha}} \eta_t^{\frac{2}{1-\alpha}}.
$$

Combining these two inequalities above, for the case when $I_j^t = i$, we have

$$
-\frac{2\eta_t}{m}\sum_{k=1}^m \big\langle \bar{\theta}^{(t)} - \bar{\theta}^{(t,ij)}, \nabla\ell(\theta_k^{(t)}; Z_{I_k^t,k}) - \nabla\ell(\theta_k^{(t,ij)}; \tilde{Z}_{I_k^t,k}^{(ij)})\big\rangle
$$
$$
\leq \frac{\beta_2\eta_t}{m}\sum_{k=1}^m \big\| \bar{\theta}^{(t)} - \theta_k^{(t)} - \bar{\theta}^{(t,ij)} + \theta_k^{(t,ij)} \big\|_2^2 - \Big(\frac{2b\eta_t}{L} - \frac{1}{\beta_2}\Big)\frac{\eta_t}{m}\sum_{k\neq j} \big\| \nabla\ell(\theta_k^{(t)}; Z_{I_k^t,k}) - \nabla\ell(\theta_k^{(t,ij)}; \tilde{Z}_{I_k^t,k}^{(ij)}) \big\|_2^2
$$
$$
+ \frac{2\eta_t}{m}\big\| \bar{\theta}^{(t)} - \bar{\theta}^{(t,ij)} \big\|_2 \big\| \nabla\ell(\theta_j^{(t)}; z_{i,j}) - \nabla\ell(\theta_j^{(t,ij)}; \tilde{z}_{i,j}) \big\|_2 + \frac{2L(1-\alpha)}{1+\alpha} b^{\frac{1+\alpha}{1-\alpha}} \eta_t^{\frac{2}{1-\alpha}}. \quad (C.10)
$$

Therefore, combining the cases when $I_j^t \neq i$ and $I_j^t = i$, i.e., Eq. (C.9) and Eq. (C.10), we have

$$
-\frac{2\eta_t}{m}\sum_{k=1}^m \big\langle \bar{\theta}^{(t)} - \bar{\theta}^{(t,ij)}, \nabla\ell(\theta_k^{(t)}; Z_{I_k^t,k}) - \nabla\ell(\theta_k^{(t,ij)}; \tilde{Z}_{I_k^t,k}^{(ij)})\big\rangle
$$
$$
\leq \frac{\beta_2\eta_t}{m}\sum_{k=1}^m \big\| \bar{\theta}^{(t)} - \theta_k^{(t)} - \bar{\theta}^{(t,ij)} + \theta_k^{(t,ij)} \big\|_2^2 - \Big(\frac{2b\eta_t}{L} - \frac{1}{\beta_2}\Big)\frac{\eta_t}{m}\sum_{k\neq j} \big\| \nabla\ell(\theta_k^{(t)}; Z_{I_k^t,k}) - \nabla\ell(\theta_k^{(t,ij)}; \tilde{Z}_{I_k^t,k}^{(ij)}) \big\|_2^2
$$
$$
- \Big(\frac{2\eta_t}{L} - \frac{1}{\beta_2}\Big)\frac{\eta_t}{m}\big\| \nabla\ell(\theta_j^{(t)}; Z_{I_j^t,j}) - \nabla\ell(\theta_j^{(t,ij)}; \tilde{Z}_{I_j^t,j}^{(ij)}) \big\|_2^2 \mathbb{I}_{\{I_j^t\neq i\}}
$$
$$
+ \frac{2\eta_t}{m}\big\| \bar{\theta}^{(t)} - \bar{\theta}^{(t,ij)} \big\|_2 \big\| \nabla\ell(\theta_j^{(t)}; z_{i,j}) - \nabla\ell(\theta_j^{(t,ij)}; \tilde{z}_{i,j}) \big\|_2 \mathbb{I}_{\{I_j^t=i\}} + \frac{2L(1-\alpha)}{1+\alpha} b^{\frac{1+\alpha}{1-\alpha}} \eta_t^{\frac{2}{1-\alpha}}. \quad (C.11)
$$

**Iteration form:** Plugging Eq. (C.3) and Eq. (C.11) into Eq.(C.2), we have

$$\|\bar{\theta}^{(t+1)} - \bar{\theta}^{(t+1,ij)}\|_2^2 \leq \|\bar{\theta}^{(t)} - \bar{\theta}^{(t,ij)}\|_2^2 + \frac{\beta_2 \eta_t}{m} \sum_{k=1}^m \|\bar{\theta}^{(t)} - \theta_k^{(t)} - \bar{\theta}^{(t,ij)} + \theta_k^{(t,ij)}\|_2^2$$

$$+ \left( (1+\beta_1)(m-1)\frac{\eta_t^2}{m^2} - \left(\frac{2b\eta_t}{L} - \frac{1}{\beta_2}\right)\frac{\eta_t}{m} \right) \sum_{k \neq j} \|(\nabla\ell(\theta_k^{(t)}; Z_{I_k^t,k}) - \nabla\ell(\theta_k^{(t,ij)}; \tilde{Z}_{I_k^t,k}^{(ij)}))\|_2^2$$

$$+ \left( (1 + \frac{1}{\beta_1})\frac{\eta_t^2}{m^2} - \left(\frac{2b\eta_t}{L} - \frac{1}{\beta_2}\right)\frac{\eta_t}{m} \right) \|\nabla\ell(\theta_j^{(t)}; Z_{I_j^t,j}) - \nabla\ell(\theta_j^{(t,ij)}; \tilde{Z}_{I_j^t,j}^{(ij)})\|_2^2 \mathbb{I}_{\{I_j^t \neq i\}}$$

$$+ (1 + \frac{1}{\beta_1})\frac{\eta_t^2}{m^2} \|(\nabla\ell(\theta_j^{(t)}; z_{i,j}) - \nabla\ell(\theta_j^{(t,ij)}; \tilde{z}_{i,j}))\|_2^2 \mathbb{I}_{\{I_j^t = i\}} + \frac{2L(1-\alpha)}{1+\alpha} b^{\frac{1+\alpha}{1-\alpha}} \eta_t^{\frac{2}{1-\alpha}}$$

$$+ \frac{2\eta_t}{m} \|\bar{\theta}^{(t)} - \bar{\theta}^{(t,ij)}\|_2 \|\nabla\ell(\theta_j^{(t)}; z_{i,j}) - \nabla\ell(\theta_j^{(t,ij)}; \tilde{z}_{i,j})\|_2 \mathbb{I}_{\{I_j^t = i\}}. \tag{C.12}$$

By Lemma B.3, we get

$$\sum_{t=0}^{T-1} \frac{\beta_2 \eta_t}{m} \sum_{k=1}^m \|\bar{\theta}^{(t)} - \theta_k^{(t)} - \bar{\theta}^{(t,ij)} + \theta_k^{(t,ij)}\|_2^2$$

$$+ \sum_{t=0}^{T-1} \left( (1+\beta_1)(m-1)\frac{\eta_t^2}{m^2} - \left(\frac{2b\eta_t}{L} - \frac{1}{\beta_2}\right)\frac{\eta_t}{m} \right) \sum_{k \neq j} \|(\nabla\ell(\theta_k^{(t)}; Z_{I_k^t,k}) - \nabla\ell(\theta_k^{(t,ij)}; \tilde{Z}_{I_k^t,k}^{(ij)}))\|_2^2$$

$$+ \sum_{t=0}^{T-1} \left( (1 + \frac{1}{\beta_1})\frac{\eta_t^2}{m^2} - \left(\frac{2b\eta_t}{L} - \frac{1}{\beta_2}\right)\frac{\eta_t}{m} \right) \|\nabla\ell(\theta_j^{(t)}; Z_{I_j^t,j}) - \nabla\ell(\theta_j^{(t,ij)}; \tilde{Z}_{I_j^t,j}^{(ij)})\|_2^2 \mathbb{I}_{\{I_j^t \neq i\}}$$

$$\leq \sum_{t=0}^{T-1} \frac{\eta_t}{m} \left( \left(\frac{\alpha_4 \beta_2}{(1-\alpha_3)}\eta_t + \frac{(1+\beta_1)(m-1)}{m}\right)\eta_t - \left(\frac{2b\eta_t}{L} - \frac{1}{\beta_2}\right) \right) \sum_{k \neq j} \|(\nabla\ell(\theta_k^{(t)}; Z_{I_k^t,k}) - \nabla\ell(\theta_k^{(t,ij)}; \tilde{Z}_{I_k^t,k}^{(ij)}))\|_2^2$$

$$+ \sum_{t=0}^{T-1} \frac{\eta_t}{m} \left( \left(\frac{\alpha_4 \beta_2}{(1-\alpha_3)}\eta_t + \frac{1+\frac{1}{\beta_1}}{m}\right)\eta_t - \left(\frac{2b\eta_t}{L} - \frac{1}{\beta_2}\right) \right) \|\nabla\ell(\theta_j^{(t)}; Z_{I_j^t,j}) - \nabla\ell(\theta_j^{(t,ij)}; \tilde{Z}_{I_j^t,j}^{(ij)})\|_2^2 \mathbb{I}_{\{I_j^t \neq i\}}$$

$$+ \sum_{t=0}^{T-1} \frac{\eta_t^3 \alpha_4 \beta_2 \mathbb{I}_{\{I_j^t = i\}}}{(1-\alpha_3)m} \|\nabla\ell(\theta_j^{(t)}; Z_{I_j^t,j}) - \nabla\ell(\theta_j^{(t,ij)}; \tilde{Z}_{I_j^t,j}^{(ij)})\|_2^2$$

$$\leq \sum_{t=0}^{T-1} \frac{\eta_t^2 \alpha_4 L \mathbb{I}_{\{I_j^t = i\}}}{(1-\alpha_3)bm} \|\nabla\ell(\theta_j^{(t)}; Z_{I_j^t,j}) - \nabla\ell(\theta_j^{(t,ij)}; \tilde{Z}_{I_j^t,j}^{(ij)})\|_2^2,$$

where we choose $\beta_1 = 1, \beta_2 = L/(b\eta_t)$, and $b = \frac{\alpha_4 L}{1-\alpha_3} + 2L$ to use $(b \geq L)$

$$\left(\frac{\alpha_4 \beta_2}{(1-\alpha_3)}\eta_t + \frac{(1+\beta_1)(m-1)}{m}\right)\eta_t - \left(\frac{2b\eta_t}{L} - \frac{1}{\beta_2}\right) = \left(\frac{\alpha_4 L}{b(1-\alpha_3)} + \frac{2(m-1)}{m} - \frac{b}{L}\right)\eta_t$$

$$\leq \left(\frac{\alpha_4}{1-\alpha_3} + 2 - \frac{\frac{\alpha_4 L}{1-\alpha_3} + 2L}{L}\right)\eta_t = 0$$

and similarly

$$\left(\frac{\alpha_4 \beta_2}{1-\alpha_3}\eta_t + \frac{1+\frac{1}{\beta_1}}{m}\right)\eta_t - \left(\frac{2b\eta_t}{L} - \frac{1}{\beta_2}\right) \leq 0.$$

Therefore, taking a summation for $t = 0, \ldots, T-1$ to both sides of Eq. (C.12) and using Lemma B.3, we have

$$\|\bar{\theta}^{(T)} - \bar{\theta}^{(t,ij)}\|_2^2 \leq \sum_{t=0}^{T-1} \Big(\frac{\alpha_4 L}{(1-\alpha_3)b} + \frac{2}{m}\Big)\frac{\eta_t^2}{m}\big\|\big(\nabla\ell(\theta_j^{(t)}; z_{i,j}) - \nabla\ell(\theta_j^{(t,ij)}; \tilde{z}_{i,j})\big)\big\|_2^2 \mathbb{I}_{\{I_j^t = i\}}$$
$$+ \sum_{t=0}^{T-1} \frac{2\eta_t}{m}\big\|\bar{\theta}^{(t)} - \bar{\theta}^{(t,ij)}\big\|_2\big\|\nabla\ell(\theta_j^{(t)}; z_{i,j}) - \nabla\ell(\theta_j^{(t,ij)}; \tilde{z}_{i,j})\big\|_2 \mathbb{I}_{\{I_j^t = i\}} + \frac{2L(1-\alpha)}{1+\alpha}\sum_{t=0}^{T-1} b^{\frac{1+\alpha}{1-\alpha}}\eta_t^{\frac{2}{1-\alpha}}. \tag{C.13}$$

Taking expectation to both sides of Eq. (C.13) and use the fact that $I_j^t$ is independent of $\bar{\theta}^{(t)}$, $\bar{\theta}^{(t,ij)}$, $\theta_j^{(t)}$ and $\theta_j^{(t,ij)}$, we know

$$\mathbb{E}[\|\bar{\theta}^{(T)} - \bar{\theta}^{(t,ij)}\|_2^2] \leq \frac{1}{mn}\sum_{t=0}^{T-1}\Big(\frac{\alpha_4 L}{(1-\alpha_3)b} + \frac{2}{m}\Big)\eta_t^2\mathbb{E}\big[\big\|\big(\nabla\ell(\theta_j^{(t)}; z_{i,j}) - \nabla\ell(\theta_j^{(t,ij)}; \tilde{z}_{i,j})\big)\big\|_2^2\big]$$
$$+ \sum_{t=0}^{T-1}\frac{2\eta_t}{mn}\mathbb{E}\big[\big\|\bar{\theta}^{(t)} - \bar{\theta}^{(t,ij)}\big\|_2\big\|\nabla\ell(\theta_j^{(t)}; z_{i,j}) - \nabla\ell(\theta_j^{(t,ij)}; \tilde{z}_{i,j})\big\|_2\big] + \frac{2L(1-\alpha)}{1+\alpha}\sum_{t=0}^{T-1} b^{\frac{1+\alpha}{1-\alpha}}\eta_t^{\frac{2}{1-\alpha}}. \tag{C.14}$$

Introduce

$$\Delta_t = \big(\mathbb{E}[\|\bar{\theta}^{(t)} - \bar{\theta}^{(t,ij)}\|_2^2]\big)^{\frac{1}{2}}, \quad \forall t \in [T], \quad \text{and} \quad \Delta = \max_{t \leq T}\Delta_t.$$

By the Cauchy–Schwarz inequality, we know

$$\mathbb{E}\Big[\big\|\bar{\theta}^{(t)} - \bar{\theta}^{(t,ij)}\big\|_2\big\|\nabla\ell(\theta_j^{(t)}; z_{i,j}) - \nabla\ell(\theta_j^{(t,ij)}; \tilde{z}_{i,j})\big\|_2\Big] \leq \Big(\mathbb{E}\Big[\big\|\bar{\theta}^{(t)} - \bar{\theta}^{(t,ij)}\big\|_2^2\Big]\Big)^{\frac{1}{2}}\Big(\mathbb{E}\Big[\big\|\nabla\ell(\theta_j^{(t)}; z_{i,j}) - \nabla\ell(\theta_j^{(t,ij)}; \tilde{z}_{i,j})\big\|_2^2\Big]\Big)^{\frac{1}{2}}.$$

It then follows that

$$\Delta^2 \leq \frac{1}{mn}\sum_{t=0}^{T-1}\Big(\frac{\alpha_4 L}{(1-\alpha_3)b} + \frac{2}{m}\Big)\eta_t^2\mathbb{E}\big[\big\|\big(\nabla\ell(\theta_j^{(t)}; z_{i,j}) - \nabla\ell(\theta_j^{(t,ij)}; \tilde{z}_{i,j})\big)\big\|_2^2\big]$$
$$+ \frac{2}{mn}\sum_{t=0}^{T-1}\eta_t\Big(\mathbb{E}\big[\big\|\nabla\ell(\theta_j^{(t)}; z_{i,j}) - \nabla\ell(\theta_j^{(t,ij)}; \tilde{z}_{i,j})\big\|_2^2\big]\Big)^{\frac{1}{2}}\Delta + \frac{2L(1-\alpha)}{1+\alpha}\sum_{t=0}^{T-1} b^{\frac{1+\alpha}{1-\alpha}}\eta_t^{\frac{2}{1-\alpha}}. \tag{C.15}$$

According to Lemma B.1, this further shows

$$\Delta^2 \leq \frac{2}{mn}\sum_{t=0}^{T-1}\Big(\frac{\alpha_4 L}{(1-\alpha_3)b} + \frac{2}{m}\Big)\eta_t^2\mathbb{E}\big[\big\|\big(\nabla\ell(\theta_j^{(t)}; z_{i,j}) - \nabla\ell(\theta_j^{(t,ij)}; \tilde{z}_{i,j})\big)\big\|_2^2\big]$$
$$+ \frac{4}{m^2 n^2}\Big(\sum_{t=0}^{T-1}\eta_t\Big(\mathbb{E}\big[\big\|\nabla\ell(\theta_j^{(t)}; z_{i,j}) - \nabla\ell(\theta_j^{(t,ij)}; \tilde{z}_{i,j})\big\|_2^2\big]\Big)^{\frac{1}{2}}\Big)^2 + \frac{4L(1-\alpha)}{1+\alpha}\sum_{t=0}^{T-1} b^{\frac{1+\alpha}{1-\alpha}}\eta_t^{\frac{2}{1-\alpha}}. \tag{C.16}$$

Eq. (B.24) shows

$$\mathbb{E}\big[\|\nabla\ell(\theta_j^{(t)}; z_{i,j}) - \nabla\ell(\theta_j^{(t,ij)}; \tilde{z}_{i,j})\|_2^2\big] \leq 4\mathbb{E}\big[\|\nabla\ell(\theta_j^{(t)}; z_{i,j})\|_2^2\big].$$

We combine the above two inequalities together, and get

$$\mathbb{E}[\|A(S) - A(S^{(ij)})\|_2^2] \leq \frac{8}{mn}\sum_{t=0}^{T-1}\Big(\frac{\alpha_4 L}{(1-\alpha_3)b} + \frac{2}{m}\Big)\eta_t^2\mathbb{E}\big[\big\|\nabla\ell(\theta_j^{(t)}; z_{i,j})\big\|_2^2\big]$$
$$+ \frac{16}{m^2 n^2}\Big(\sum_{t=0}^{T-1}\eta_t\Big(\mathbb{E}\big[\big\|\nabla\ell(\theta_j^{(t)}; z_{i,j})\big\|_2^2\big]\Big)^{\frac{1}{2}}\Big)^2 + \frac{4L(1-\alpha)}{1+\alpha}\sum_{t=0}^{T-1} b^{\frac{1+\alpha}{1-\alpha}}\eta_t^{\frac{2}{1-\alpha}}. \tag{C.17}$$

Taking an average over all $i \in [n]$ and $j \in [m]$, we get

$$\frac{1}{mn}\sum_{i=1}^{n}\sum_{j=1}^{m}\mathbb{E}[\|A(S)-A(S^{(ij)})\|_2^2] \le \frac{8}{m^2n^2}\sum_{i=1}^{n}\sum_{j=1}^{m}\sum_{t=0}^{T-1}\Big(\frac{\alpha_4 L}{(1-\alpha_3)b}+\frac{2}{m}\Big)\eta_t^2\mathbb{E}\big[\|\nabla\ell(\theta_j^{(t)};z_{i,j})\|_2^2\big]$$

$$+\frac{16}{m^3n^3}\sum_{i=1}^{n}\sum_{j=1}^{m}\Big(\sum_{t=0}^{T-1}\eta_t\Big(\mathbb{E}\big[\|\nabla\ell(\theta_j^{(t)};z_{i,j})\|_2^2\big]\Big)^{\frac{1}{2}}\Big)^2 + \frac{4L(1-\alpha)}{1+\alpha}\sum_{t=0}^{T-1}b^{\frac{1+\alpha}{1-\alpha}}\eta_t^{\frac{2}{1-\alpha}}.$$

By choosing $\beta = \frac{1-\lambda}{2\lambda}$, we know

$$\frac{\alpha_4}{1-\alpha_3} = \frac{1+\beta^{-1}}{1-(1+\beta)\lambda} = \frac{1+\frac{2\lambda}{1-\lambda}}{1-(1+\frac{1-\lambda}{2\lambda})\lambda} = \frac{\frac{1+\lambda}{1-\lambda}}{\frac{1-\lambda}{2}} = \frac{2(1+\lambda)}{(1-\lambda)^2}$$

and further get

$$\frac{1}{mn}\sum_{i=1}^{n}\sum_{j=1}^{m}\mathbb{E}[\|A(S)-A(S^{(ij)})\|_2^2] \le \frac{8}{m^2n^2}\sum_{i=1}^{n}\sum_{j=1}^{m}\sum_{t=0}^{T-1}\Big(\frac{2(1+\lambda)L}{(1-\lambda)^2 b}+\frac{2}{m}\Big)\eta_t^2\mathbb{E}\big[\|\nabla\ell(\theta_j^{(t)};z_{i,j})\|_2^2\big]$$

$$+\frac{16}{m^3n^3}\sum_{i=1}^{n}\sum_{j=1}^{m}\Big(\sum_{t=0}^{T-1}\eta_t\Big(\mathbb{E}\big[\|\nabla\ell(\theta_j^{(t)};z_{i,j})\|_2^2\big]\Big)^{\frac{1}{2}}\Big)^2 + \frac{4L(1-\alpha)}{1+\alpha}\sum_{t=0}^{T-1}b^{\frac{1+\alpha}{1-\alpha}}\eta_t^{\frac{2}{1-\alpha}}.$$

Furthermore, we know

$$b = \frac{\alpha_4 L}{1-\alpha_3} + 2L = \frac{2L(1+\lambda)}{(1-\lambda)^2} + 2L.$$

The proof is completed. □

*Proof of Corollary 5.2.* From the $G$-Lipschitz continuity of the loss function $\ell(\cdot;z)$ (Definition 3.4), we know

$$\|\nabla\ell(\theta,z)-\nabla\ell(\theta',z)\|_2 \le \|\nabla\ell(\theta,z)\| + \|\nabla\ell(\theta',z)\|_2 \le 2G, \quad \forall\theta,\theta' \in \mathbb{R}^d,$$

i.e., the loss function has $(0,2G)$-Hölder continuous gradients (Definition 3.3). Then, by choosing $\alpha = 0$ in Theorem 5.1, we know

$$\frac{1}{mn}\sum_{i=1}^{n}\sum_{j=1}^{m}\mathbb{E}[\|A(S)-A(S^{(ij)})\|_2^2] \le \frac{8}{m^2n^2}\sum_{i=1}^{n}\sum_{j=1}^{m}\sum_{t=0}^{T-1}\Big(\frac{1}{2G}+\frac{2}{m}\Big)\eta_t^2\mathbb{E}\big[\|\nabla\ell(\theta_j^{(t)};z_{i,j})\|_2^2\big]$$

$$+\frac{16}{m^3n^3}\sum_{i=1}^{n}\sum_{j=1}^{m}\Big(\sum_{t=0}^{T-1}\eta_t\Big(\mathbb{E}\big[\|\nabla\ell(\theta_j^{(t)};z_{i,j})\|_2^2\big]\Big)^{\frac{1}{2}}\Big)^2 + 8Gb\sum_{t=0}^{T-1}\eta_t^2. \qquad \text{(C.18)}$$

From the $G$-Lipschitz continuity of the loss function $\ell(\cdot;z)$ (Definition 3.4), we know

$$\|\nabla\ell(\theta,z)\|_2 \le G, \quad \forall\theta \in \mathbb{R}^d, z \in \mathcal{Z}.$$

Plugging this into Eq. (C.18) and noting $b = \frac{4(1+\lambda)G}{(1-\lambda)^2} + 4G$ give the stated bound. □

## C.2. Proofs on Convergence Analysis

For the convergence analysis, we require the following two lemmas on functions with Hölder continuous gradients. Lemma C.1 shows the self-bounding property, while Lemma C.2 presents a bound on the first-order approximation for functions with Hölder continuous gradients.

**Lemma C.1** (Lei and Ying 2020). *Assume the loss function $\ell(\cdot; z)$ is nonnegative and has $(\alpha, L)$-Hölder continuous gradients with $\alpha \in [0, 1]$. Then,*

$$\|\nabla \ell(\theta, z)\|_2 \leq c_{\alpha,1} \ell^{\frac{\alpha}{1+\alpha}}(\theta, z), \quad \forall \theta \in \mathbb{R}^d, z \in \mathcal{Z},$$

*where $c_{\alpha,1}$ is defined as*

$$c_{\alpha,1} = \begin{cases} (1 + 1/\alpha)^{\frac{\alpha}{1+\alpha}} L^{\frac{1}{1+\alpha}}, & \text{if } \alpha > 0 \\ \sup_z \|\nabla \ell(0; z)\|_2 + L, & \text{if } \alpha = 0. \end{cases}$$

**Lemma C.2** (Nesterov 2015). *Suppose $f$ has $(\alpha, L)$-Hölder continuous gradients over a given convex set $D$, where $\alpha \in [0, 1]$. Then for any $\mathbf{x}, \mathbf{y} \in D$,*

$$f(\mathbf{y}) \leq f(\mathbf{x}) + \langle \nabla f(\mathbf{x}), \mathbf{y} - \mathbf{x} \rangle + \frac{L}{\alpha + 1} \|\mathbf{y} - \mathbf{x}\|^{\alpha+1}.$$

*Proof of Lemma 5.4.* Let $\Theta^{(t)}$ and $\bar{\Theta}^{(t)}$ be defined in Eq. (B.1). For any $\theta \in \mathbb{R}^d$, there holds

$$\|\bar{\theta}^{(t+1)} - \theta\|_2^2 = \left\| \bar{\theta}^{(t)} - \eta_t \frac{1}{m} \sum_{k=1}^m \nabla \ell(\theta_k^{(t)}; z_{I_k^t, k}) - \theta \right\|_2^2$$

$$= \|\bar{\theta}^{(t)} - \theta\|_2^2 + \eta_t^2 \left\| \frac{1}{m} \sum_{k=1}^m \nabla \ell(\theta_k^{(t)}; z_{I_k^t, k}) \right\|_2^2 - 2\eta_t \left\langle \bar{\theta}^{(t)} - \theta, \frac{1}{m} \sum_{k=1}^m \nabla \ell(\theta_k^{(t)}; z_{I_k^t, k}) \right\rangle. \tag{C.19}$$

By the Cauchy-Schwarz's inequality, we know

$$\left\| \frac{1}{m} \sum_{k=1}^m \nabla \ell(\theta_k^{(t)}; z_{I_k^t, k}) \right\|_2^2 \leq 2 \left\| \frac{1}{m} \sum_{k=1}^m \left( \nabla \ell(\theta_k^{(t)}; z_{I_k^t, k}) - \nabla \ell(\bar{\theta}^{(t)}; z_{I_k^t, k}) \right) \right\|_2^2 + 2 \left\| \frac{1}{m} \sum_{k=1}^m \nabla \ell(\bar{\theta}^{(t)}; z_{I_k^t, k}) \right\|_2^2$$

$$\leq \frac{2}{m} \sum_{k=1}^m \left\| \nabla \ell(\theta_k^{(t)}; z_{I_k^t, k}) - \nabla \ell(\bar{\theta}^{(t)}; z_{I_k^t, k}) \right\|_2^2 + \frac{2}{m} \sum_{k=1}^m \left\| \nabla \ell(\bar{\theta}^{(t)}; z_{I_k^t, k}) \right\|_2^2$$

$$\leq \frac{2L^2}{m} \sum_{k=1}^m \|\theta_k^{(t)} - \bar{\theta}^{(t)}\|_2^{2\alpha} + \frac{2c_{\alpha,1}^2}{m} \sum_{k=1}^m \ell^{\frac{2\alpha}{\alpha+1}}(\bar{\theta}^{(t)}; z_{I_k^t, k})$$

$$\leq \frac{2L^2}{m^\alpha} \|\Theta^{(t)} - \bar{\Theta}^{(t)}\|_F^{2\alpha} + \frac{2c_{\alpha,1}^2}{m} \sum_{k=1}^m \ell^{\frac{2\alpha}{\alpha+1}}(\bar{\theta}^{(t)}; z_{I_k^t, k}), \tag{C.20}$$

where the second inequality holds due to the $(\alpha, L)$-Hölder continuity of $\nabla \ell(\cdot, z)$ and the self-bounding property (Lemma C.1), and the last inequality holds due to the concavity of $x \to x^\alpha$.

$$\frac{1}{m} \sum_{k=1}^m \|\theta_k^{(t)} - \bar{\theta}^{(t)}\|_2^{2\alpha} \leq \left( \frac{1}{m} \sum_{k=1}^m \|\theta_k^{(t)} - \bar{\theta}^{(t)}\|_2^2 \right)^\alpha. \tag{C.21}$$

Taking expectation to both sides of Eq. (C.20), and using the fact that

$$\mathbb{E}\left[ \frac{1}{m} \sum_{k=1}^m \ell^{\frac{2\alpha}{\alpha+1}}(\bar{\theta}^{(t)}; z_{I_k^t, k}) \right] \leq \left( \mathbb{E}\left[ \frac{1}{m} \sum_{k=1}^m \ell(\bar{\theta}^{(t)}; z_{I_k^t, k}) \right] \right)^{\frac{2\alpha}{\alpha+1}} = \left( \mathbb{E}[R_S(\bar{\theta}^{(t)})] \right)^{\frac{2\alpha}{\alpha+1}},$$

we have

$$\mathbb{E}\left[ \left\| \frac{1}{m} \sum_{k=1}^m \nabla \ell(\theta_k^{(t)}; z_{I_k^t, k}) \right\|_2^2 \right] \leq \frac{2L^2}{m^\alpha} \mathbb{E}[\|\Theta^{(t)} - \bar{\Theta}^{(t)}\|_F^{2\alpha}] + 2c_{\alpha,1}^2 \left( \mathbb{E}[R_S(\bar{\theta}^{(t)})] \right)^{\frac{2\alpha}{\alpha+1}}. \tag{C.22}$$

Since

$$\left\langle \bar{\theta}^{(t)} - \theta, \frac{1}{m} \sum_{k=1}^{m} \nabla \ell(\theta_k^{(t)}; z_{I_k^t, k}) \right\rangle = \frac{1}{m} \sum_{k=1}^{m} \left\langle \bar{\theta}^{(t)} - \theta, \nabla \ell(\theta_k^{(t)}; z_{I_k^t, k}) \right\rangle$$

$$= \frac{1}{m} \sum_{k=1}^{m} \left\langle \bar{\theta}^{(t)} - \theta_k^{(t)}, \nabla \ell(\theta_k^{(t)}; z_{I_k^t, k}) \right\rangle + \frac{1}{m} \sum_{k=1}^{m} \left\langle \theta_k^{(t)} - \theta, \nabla \ell(\theta_k^{(t)}; z_{I_k^t, k}) \right\rangle$$

$$\geq \frac{1}{m} \sum_{k=1}^{m} \left( \ell(\bar{\theta}^{(t)}; z_{I_k^t, k}) - \ell(\theta_k^{(t)}; z_{I_k^t, k}) - \frac{L}{\alpha + 1} \|\theta_k^{(t)} - \bar{\theta}^{(t)}\|_2^{\alpha+1} \right) + \frac{1}{m} \sum_{k=1}^{m} \left( \ell(\theta_k^{(t)}; z_{I_k^t, k}) - \ell(\theta; z_{I_k^t, k}) \right)$$

$$= \frac{1}{m} \sum_{k=1}^{m} \left( \ell(\bar{\theta}^{(t)}; z_{I_k^t, k}) - \ell(\theta; z_{I_k^t, k}) - \frac{L}{\alpha + 1} \|\theta_k^{(t)} - \bar{\theta}^{(t)}\|_2^{\alpha+1} \right), \tag{C.23}$$

where the third inequality holds due to the $(\alpha, L)$-Hölder continuity (Lemma C.2) and convexity. Taking expectation on both sides of Eq. (C.23), we have

$$\mathbb{E}\left[ \left\langle \bar{\theta}^{(t)} - \theta, \frac{1}{m} \sum_{k=1}^{m} \nabla \ell(\theta_k^{(t)}; z_{I_k^t, k}) \right\rangle \right] \geq \mathbb{E}[R_S(\bar{\theta}^{(t)})] - \mathbb{E}[R_S(\theta)] - \frac{L}{(\alpha + 1)m} \mathbb{E}[\|\Theta^{(t)} - \bar{\Theta}^{(t)}\|_F^{\alpha+1}], \tag{C.24}$$

where we have used the following inequality since $\alpha \in [0, 1]$ (i.e., if $p \leq q$, then $\|\mathbf{a}\|_p \geq \|\mathbf{a}\|_q$ for $\mathbf{a} \in \mathbb{R}^m$)

$$\left( \sum_{k=1}^{m} \|\theta_k^{(t)} - \bar{\theta}^{(t)}\|_2^{\alpha+1} \right)^{\frac{1}{\alpha+1}} \geq \left( \sum_{k=1}^{m} \|\theta_k^{(t)} - \bar{\theta}^{(t)}\|_2^2 \right)^{\frac{1}{2}} = \|\Theta^{(t)} - \bar{\Theta}^{(t)}\|_F.$$

Taking expectation to both sides of Eq. (C.19) and plugging Eq. (C.22) and Eq. (C.24) into it, we have

$$\mathbb{E}[\|\bar{\theta}^{(t+1)} - \theta\|_2^2] \leq \mathbb{E}[\|\bar{\theta}^{(t)} - \theta\|_2^2] + \frac{2L^2 \eta_t^2}{m^\alpha} \mathbb{E}[\|\Theta^{(t)} - \bar{\Theta}^{(t)}\|_F^{2\alpha}] + \frac{2L\eta_t}{(\alpha + 1)m} \mathbb{E}[\|\Theta^{(t)} - \bar{\Theta}^{(t)}\|_F^{\alpha+1}]$$

$$+ 2c_{\alpha,1}^2 \eta_t^2 \left( \mathbb{E}[R_S(\bar{\theta}^{(t)})] \right)^{\frac{2\alpha}{\alpha+1}} + 2\eta_t \mathbb{E}[R_S(\theta) - R_S(\bar{\theta}^{(t)})].$$

Taking a summation of the above inequality gives

$$\frac{1}{T} \sum_{t=1}^{T} \mathbb{E}[R_S(\bar{\theta}^{(t)}) - R_S(\theta)] \leq \frac{\mathbb{E}[\|\bar{\theta}^{(1)} - \theta\|_2^2]}{2\eta T} + \frac{L^2 \eta}{m^\alpha T} \sum_{t=1}^{T} \mathbb{E}[\|\Theta^{(t)} - \bar{\Theta}^{(t)}\|_F^{2\alpha}]$$

$$+ \frac{L}{(\alpha + 1)mT} \sum_{t=1}^{T} \mathbb{E}[\|\Theta^{(t)} - \bar{\Theta}^{(t)}\|_F^{\alpha+1}] + \frac{c_{\alpha,1}^2 \eta}{T} \sum_{t=1}^{T} \left( \mathbb{E}[R_S(\bar{\theta}^{(t)})] \right)^{\frac{2\alpha}{\alpha+1}}.$$

The proof is completed. $\qquad\square$

The following lemma presents a bound on the distance between the D-SGD iterates around their mean over the nodes.

**Lemma C.3.** *Assume the loss function $\ell(\cdot; z)$ has $(\alpha, L)$-Hölder continuous gradients and Assumption 3.1 holds. Let $\Theta^{(t)}$ and $\bar{\Theta}^{(t)}$ be defined in Eq. (B.1). For any $x \in (0, 1]$, $\beta > 0$ and $a \in \mathbb{R}$, if we choose $\eta_t$ such that*

$$a_{\alpha,x} := (1 + \beta)^x \lambda^x + \alpha (2(1 + 1/\beta)L^2)^x m^{x(1-\alpha)} \eta_1^a < 1,$$

*then*

$$\mathbb{E}[\|\Theta^{(t)} - \bar{\Theta}^{(t)}\|_F^{2x}] \leq \left( 2c_{\alpha,1}^2 (1 + \frac{1}{\beta}) \right)^x m^x \sum_{\tau=1}^{t-1} a_{\alpha,x}^{\tau-1} \eta_{t-\tau}^{2x} \left( \mathbb{E}[R_S(\bar{\theta}^{(t-\tau)})] \right)^{\frac{2\alpha x}{1+\alpha}}$$

$$+ (1 - \alpha) \left( 2(1 + \frac{1}{\beta})L^2 \right)^x m^{x(1-\alpha)} \sum_{\tau=1}^{t-1} a_{\alpha,x}^{\tau-1} \eta_{t-\tau}^{\frac{2x-a\alpha}{1-\alpha}}. \tag{C.25}$$

*Proof.* Recall $W^\infty$ defined in Eq. (B.4). For any $0 < x \leq 1$ and $\beta > 0$, we have

$$
\begin{aligned}
\|\Theta^{(t)} - \bar{\Theta}^{(t)}\|_F^{2x} &= \|\Theta^{(t)} - \bar{\Theta}^{(t-1)} - \bar{\Theta}^{(t)} + \bar{\Theta}^{(t-1)}\|_F^{2x} \leq \|\Theta^{(t)} - \bar{\Theta}^{(t-1)}\|_F^{2x} \\
&= \|W\Theta^{(t-1)} - \eta_{t-1}\nabla\ell(\Theta^{(t-1)}; \mathbf{z}_{I^{t-1}}) - \bar{\Theta}^{(t-1)}\|_F^{2x} \\
&\leq \left((1+\beta)\|W\Theta^{(t-1)} - \bar{\Theta}^{(t-1)}\|_F^2 + (1+\frac{1}{\beta})\eta_{t-1}^2\|\nabla\ell(\Theta^{(t-1)}; \mathbf{z}_{I^{t-1}})\|_F^2\right)^x \\
&\leq (1+\beta)^x\|W\Theta^{(t-1)} - \bar{\Theta}^{(t-1)}\|_F^{2x} + (1+1/\beta)^x\eta_{t-1}^{2x}\|\nabla\ell(\Theta^{(t-1)}; \mathbf{z}_{I^{t-1}})\|_F^{2x}, \qquad \text{(C.26)}
\end{aligned}
$$

where the first inequality holds due to $\|\Theta - \bar{\Theta}\|_F \leq \|\Theta\|_F$ for all $\Theta \in \mathbb{R}^{m \times d}$ (Taheri and Thrampoulidis, 2023), the second inequality holds due to $\|a + b\|^2 \leq (1+\beta)\|a\|^2 + \left(1 + \frac{1}{\beta}\right)\|b\|^2$, and the last inequality holds due to the fact that for any $a_1 > 0$, $a_2 > 0$, and $x \in (0, 1]$,

$$
(a_1 + a_2)^x \leq a_1^x + a_2^x.
$$

For any $\Theta \in \mathbb{R}^{m \times d}$, Eq. (B.5) shows that $\|W\Theta - \bar{\Theta}\|_F^2 \leq \lambda\|\Theta - \bar{\Theta}\|_F^2$. Therefore,

$$
\|W\Theta - \bar{\Theta}\|_F^{2x} \leq \lambda^x\|\Theta - \bar{\Theta}\|_F^{2x}.
$$

Applying this to Eq. (C.26), we know

$$
\|\Theta^{(t)} - \bar{\Theta}^{(t)}\|_F^{2x} \leq (1+\beta)^x\lambda^x\|\Theta^{(t-1)} - \bar{\Theta}^{(t-1)}\|_F^{2x} + (1+1/\beta)^x\eta_{t-1}^{2x}\|\nabla\ell(\Theta^{(t-1)}; \mathbf{z}_{I^{t-1}})\|_F^{2x}. \qquad \text{(C.27)}
$$

By the $(\alpha, L)$-Hölder continuity of $\nabla\ell(\cdot, z)$ and the self-bounding property in Lemma C.1, we get

$$
\begin{aligned}
\|\nabla\ell(\Theta^{(t-1)}; \mathbf{z}_{I^{t-1}})\|_F^2 &= \|\nabla\ell(\Theta^{(t-1)}; \mathbf{z}_{I^{t-1}}) - \nabla\ell(\bar{\Theta}^{(t-1)}; \mathbf{z}_{I^{t-1}}) + \nabla\ell(\bar{\Theta}^{(t-1)}; \mathbf{z}_{I^{t-1}})\|_F^2 \\
&\leq 2\|\nabla\ell(\Theta^{(t-1)}; \mathbf{z}_{I^{t-1}}) - \nabla\ell(\bar{\Theta}^{(t-1)}; \mathbf{z}_{I^{t-1}})\|_F^2 + 2\|\nabla\ell(\bar{\Theta}^{(t-1)}; \mathbf{z}_{I^{t-1}})\|_F^2 \\
&\leq 2L^2\sum_{k=1}^m\|\theta_k^{(t-1)} - \bar{\theta}_k^{(t-1)}\|_2^{2\alpha} + 2\sum_{k=1}^m\|\ell(\bar{\theta}^{(t-1)}; z_{I_k^{t-1},k})\|_2^2 \\
&\leq 2L^2 m^{1-\alpha}\|\Theta^{(t-1)} - \bar{\Theta}^{(t-1)}\|_F^{2\alpha} + 2c_{\alpha,1}^2\sum_{k=1}^m\ell^{\frac{2\alpha}{\alpha+1}}(\bar{\theta}^{(t-1)}; z_{I_k^{t-1},k}),
\end{aligned}
$$

where we have used Eq. (C.21). Therefore,

$$
\begin{aligned}
\|\nabla\ell(\Theta^{(t-1)}; \mathbf{z}_{I^{t-1}})\|_F^{2x} &= \left(\|\nabla\ell(\Theta^{(t-1)}; \mathbf{z}_{I^{t-1}})\|_F^2\right)^x \\
&\leq \left(2L^2 m^{1-\alpha}\right)^x\|\Theta^{(t-1)} - \bar{\Theta}^{(t-1)}\|_F^{2\alpha x} + \left(2c_{\alpha,1}^2\sum_{k=1}^m\ell^{\frac{2\alpha}{\alpha+1}}(\bar{\theta}^{(t-1)}; z_{I_k^{t-1},k})\right)^x. \qquad \text{(C.28)}
\end{aligned}
$$

Due to the concavity of $y \to y^{\frac{2\alpha}{1+\alpha}}$, the independence of $\bar{\theta}^{(t-1)}$ and $I_k^{t-1}$, and the fact that $R_S(\cdot) = \frac{1}{m}\sum_{k=1}^m R_{S_k}(\cdot)$, we know

$$
\begin{aligned}
\mathbb{E}\Big[\frac{1}{m}\sum_{k=1}^m\ell^{\frac{2\alpha}{\alpha+1}}(\bar{\theta}^{(t-1)}; z_{I_k^{t-1},k})\Big] &\leq \Big(\mathbb{E}\Big[\frac{1}{m}\sum_{k=1}^m\ell(\bar{\theta}^{(t-1)}; z_{I_k^{t-1},k})\Big]\Big)^{\frac{2\alpha}{\alpha+1}} \\
&= \Big(\mathbb{E}\Big[\frac{1}{m}\sum_{k=1}^m R_{S_k}(\bar{\theta}^{(t-1)})\Big]\Big)^{\frac{2\alpha}{\alpha+1}} = \left(\mathbb{E}\big[R_S(\bar{\theta}^{(t-1)})\big]\right)^{\frac{2\alpha}{\alpha+1}},
\end{aligned}
$$

by using which we get

$$
\begin{aligned}
\mathbb{E}\Big[\Big(\sum_{k=1}^m\ell^{\frac{2\alpha}{\alpha+1}}(\bar{\theta}^{(t-1)}; z_{I_k^{t-1},k})\Big)^x\Big] &\leq \Big(\mathbb{E}\Big[\Big(\sum_{k=1}^m\ell^{\frac{2\alpha}{\alpha+1}}(\bar{\theta}^{(t-1)}; z_{I_k^{t-1},k})\Big)\Big]\Big)^x \\
&= m^x\Big(\mathbb{E}\Big[\Big(\frac{1}{m}\sum_{k=1}^m\ell^{\frac{2\alpha}{\alpha+1}}(\bar{\theta}^{(t-1)}; z_{I_k^{t-1},k})\Big)\Big]\Big)^x \leq m^x\left(\mathbb{E}\big[R_S(\bar{\theta}^{(t-1)})\big]\right)^{\frac{2\alpha x}{\alpha+1}},
\end{aligned}
$$

where we use the concavity of $y \to y^x$ for the first inequality above. Therefore, taking expectation to both sides of Eq. (C.28), we have

$$\mathbb{E}[\|\nabla \ell(\Theta^{(t-1)}; \mathbf{z}_{I^{t-1}})\|_F^{2x}] \leq \left(2L^2 m^{1-\alpha}\right)^x \mathbb{E}[\|\Theta^{(t-1)} - \bar{\Theta}^{(t-1)}\|_F^{2\alpha x}] + \left(2c_{\alpha,1}^2\right)^x m^x \left(\mathbb{E}\left[R_S(\bar{\theta}^{(t-1)})\right]\right)^{\frac{2\alpha x}{\alpha+1}}. \quad (C.29)$$

Taking expectation to both sides of Eq. (C.27) and plugging Eq. (C.29) into it, we have

$$\mathbb{E}[\|\Theta^{(t)} - \bar{\Theta}^{(t)}\|_F^{2x}] \leq (1+\beta)^x \lambda^x \mathbb{E}[\|\Theta^{(t-1)} - \bar{\Theta}^{(t-1)}\|_F^{2x}] + \left(2(1+\frac{1}{\beta})L^2\right)^x m^{x(1-\alpha)} \eta_{t-1}^{2x} \mathbb{E}[\|\Theta^{(t-1)} - \bar{\Theta}^{(t-1)}\|_F^{2\alpha x}]$$

$$+ \left(2c_{\alpha,1}^2(1+\frac{1}{\beta})\right)^x m^x \eta_{t-1}^{2x} \left(\mathbb{E}[R_S(\bar{\theta}^{(t-1)})]\right)^{\frac{2\alpha x}{1+\alpha}}. \quad (C.30)$$

By Young's inequality (i.e., Eq. (C.1)) with $p = \frac{1}{\alpha}$ and $q = \frac{1}{1-\alpha}$, we know for any $a \in \mathbb{R}$,

$$\eta_{t-1}^{2x} \mathbb{E}[\|\Theta^{(t-1)} - \bar{\Theta}^{(t-1)}\|_F^{2\alpha x}] \leq \left(\eta_{t-1}^a \mathbb{E}[\|\Theta^{(t-1)} - \bar{\Theta}^{(t-1)}\|_F^{2x}]\right)^\alpha \eta_{t-1}^{2x-a\alpha}$$

$$\leq \alpha \eta_{t-1}^a \mathbb{E}[\|\Theta^{(t-1)} - \bar{\Theta}^{(t-1)}\|_F^{2x}] + (1-\alpha) \eta_{t-1}^{\frac{2x-a\alpha}{1-\alpha}}.$$

Therefore,

$$\mathbb{E}[\|\Theta^{(t)} - \bar{\Theta}^{(t)}\|_F^{2x}] \leq \left((1+\beta)^x \lambda^x + \alpha(2(1+1/\beta)L^2)^x m^{x(1-\alpha)} \eta_{t-1}^a\right) \mathbb{E}[\|\Theta^{(t-1)} - \bar{\Theta}^{(t-1)}\|_F^{2x}]$$

$$+ \left(2c_{\alpha,1}^2(1+\frac{1}{\beta})\right)^x m^x \eta_{t-1}^{2x} \left(\mathbb{E}[R_S(\bar{\theta}^{(t-1)})]\right)^{\frac{2\alpha x}{1+\alpha}} + (1-\alpha)(2(1+1/\beta)L^2)^x m^{x(1-\alpha)} \eta_{t-1}^{\frac{2x-a\alpha}{1-\alpha}}. \quad (C.31)$$

By our definition of $a_{\alpha,x}$ we know

$$(1+\beta)^x \lambda^x + \alpha(2(1+1/\beta)L^2)^x m^{x(1-\alpha)} \eta_{t-1}^a \leq a_{\alpha,x} < 1.$$

Plugging this into Eq. (C.31), we get

$$\mathbb{E}[\|\Theta^{(t)} - \bar{\Theta}^{(t)}\|_F^{2x}] \leq a_{\alpha,x} \mathbb{E}[\|\Theta^{(t-1)} - \bar{\Theta}^{(t-1)}\|_F^{2x}] +$$

$$\left(2c_{\alpha,1}^2(1+\frac{1}{\beta})\right)^x m^x \eta_{t-1}^{2x} \left(\mathbb{E}[R_S(\bar{\theta}^{(t-1)})]\right)^{\frac{2\alpha x}{1+\alpha}} + (1-\alpha)(2(1+1/\beta)L^2)^x m^{x(1-\alpha)} \eta_{t-1}^{\frac{2x-a\alpha}{1-\alpha}}. \quad (C.32)$$

Applying Eq. (C.32) repeatedly gives

$$\mathbb{E}[\|\Theta^{(t)} - \bar{\Theta}^{(t)}\|_F^{2x}] \leq \left(2c_{\alpha,1}^2(1+\frac{1}{\beta})\right)^x m^x \sum_{\tau=1}^{t-1} a_{\alpha,x}^{\tau-1} \eta_{t-\tau}^{2x} \left(\mathbb{E}[R_S(\bar{\theta}^{(t-\tau)})]\right)^{\frac{2\alpha x}{1+\alpha}}$$

$$+ (1-\alpha)\left(2(1+\frac{1}{\beta})L^2\right)^x m^{x(1-\alpha)} \sum_{\tau=1}^{t-1} a_{\alpha,x}^{\tau-1} \eta_{t-\tau}^{\frac{2x-a\alpha}{1-\alpha}}.$$

The proof is completed. $\qquad \square$

Now, we are ready to present the proof of Theorem 5.5 on convergence of D-SGD for convex and nonsmooth problems.

*Proof of Theorem 5.5.* From Lemma 5.4, choosing $\alpha = 0$, we know for any $\theta \in \mathbb{R}^d$,

$$\frac{1}{T} \sum_{t=1}^T \mathbb{E}[R_S(\bar{\theta}^{(t)}) - R_S(\theta)] \leq \frac{\mathbb{E}[\|\bar{\theta}^{(1)} - \theta\|_2^2]}{2\eta T} + (L^2 + c_{0,1}^2)\eta + \frac{L}{mT} \sum_{t=1}^T \mathbb{E}[\|\Theta^{(t)} - \bar{\Theta}^{(t)}\|_F]. \quad (C.33)$$

From the $G$-Lipschitz continuity of the loss function $\ell(\cdot; z)$ (Definition 3.4), we know

$$\|\nabla \ell(\theta, z) - \nabla \ell(\theta', z)\|_2 \leq \|\nabla \ell(\theta, z)\| + \|\nabla \ell(\theta', z)\|_2 \leq 2G, \quad \forall \theta, \theta' \in \mathbb{R}^d,$$

i.e., the loss function has $(0, 2G)$-Hölder continuous gradients (Definition 3.3). Then, from Lemma C.3, by choosing $\alpha = 0$, $x = 1/2$, and $L = 2G$ there and noting that $c_{0,1} \leq 3G$, then there holds

$$\mathbb{E}[\|\Theta^{(t)} - \bar{\Theta}^{(t)}\|_F] \leq 3\sqrt{2}(1+1/\beta)^{\frac{1}{2}}Gm^{\frac{1}{2}}\sum_{\tau=1}^{t-1}a_{0,\frac{1}{2}}^{\tau-1}\eta_{t-\tau} + 2\sqrt{2}(1+1/\beta)^{\frac{1}{2}}G\sqrt{m}\sum_{\tau=1}^{t-1}a_{0,\frac{1}{2}}^{\tau-1}\eta_{t-\tau}$$

$$= 5\sqrt{2m}(1+1/\beta)^{\frac{1}{2}}G\sum_{\tau=1}^{t-1}a_{0,\frac{1}{2}}^{\tau-1}\eta_{t-\tau}. \tag{C.34}$$

Plugging Eq. (C.34) into Eq. (C.33) and choosing a constant step size $\eta_t = \eta$, we have

$$\frac{1}{T}\sum_{t=1}^{T}\mathbb{E}[R_S(\bar{\theta}^{(t)}) - R_S(\theta)] \leq \frac{\mathbb{E}[\|\bar{\theta}^{(1)} - \theta\|_2^2]}{2\eta T} + 13G^2\eta + \frac{10\sqrt{2m}(1+1/\beta)^{\frac{1}{2}}G^2}{mT}\sum_{t=1}^{T}\sum_{\tau=1}^{t-1}a_{0,\frac{1}{2}}^{\tau-1}\eta_{t-\tau}$$

$$\leq \frac{\mathbb{E}[\|\bar{\theta}^{(1)} - \theta\|_2^2]}{2\eta T} + 13G^2\eta + \frac{10\sqrt{2m}(1+1/\beta)^{\frac{1}{2}}G^2\eta}{(1 - a_{0,\frac{1}{2}})m},$$

where we use the following fact for the second inequality above

$$\sum_{t=1}^{T}\sum_{\tau=1}^{t-1}a_{0,\frac{1}{2}}^{\tau-1}\eta_{t-\tau} \leq \sum_{\tau=1}^{T-1}a_{0,\frac{1}{2}}^{\tau-1}\sum_{t=1}^{T-\tau}\eta_t \leq \sum_{\tau=1}^{T-1}a_{0,\frac{1}{2}}^{\tau-1}\sum_{t=1}^{T-1}\eta_t \leq \frac{1}{1 - a_{0,\frac{1}{2}}}\sum_{t=1}^{T-1}\eta_t.$$

Furthermore, we know $a_{0,\frac{1}{2}} = (1+\beta)^{\frac{1}{2}}\lambda^{\frac{1}{2}}$. The proof is completed. $\qquad\square$

### C.3. Proofs on Excess Risk Analysis

*Proof of Theorem 5.6.* By Lemma 3.5 and Jensen's inequality, we know

$$|\mathbb{E}_{S,A}[R(A(S)) - R_S(A(S))]| \leq \frac{G}{mn}\sum_{i=1}^{n}\sum_{j=1}^{m}\mathbb{E}_{S,\tilde{S},A}[\|A(S^{(ij)}) - A(S)\|_2]$$

$$\leq G\left(\frac{1}{mn}\sum_{i=1}^{n}\sum_{j=1}^{m}\mathbb{E}_{S,\tilde{S},A}[\|A(S^{(ij)}) - A(S)\|_2^2]\right)^{\frac{1}{2}}. \tag{C.35}$$

Plugging Eq. (5.2) into Eq. (C.35), by choosing a constant step size $\eta_t = \eta$, we know

$$|\mathbb{E}_{S,A}[R(A(S)) - R_S(A(S))]| \lesssim G^2\left(\frac{\eta^2 T^2}{m^2 n^2} + \frac{\eta^2 T}{(1-\lambda)^2}\right)^{\frac{1}{2}}. \tag{C.36}$$

Plugging Eq. (C.36) and Eq. (5.4) into Eq. (3.1), we have

$$\mathbb{E}[R(A(S))] - R(\theta^*) \lesssim G^2\left(\frac{\eta^2 T^2}{m^2 n^2} + \frac{\eta^2 T}{(1-\lambda)^2}\right)^{\frac{1}{2}} + \frac{\|\theta^*\|_2^2}{\eta T} + G^2\eta + \frac{(1+1/\beta)^{\frac{1}{2}}G^2\eta}{(1 - (1+1/\beta)^{\frac{1}{2}}\lambda^{\frac{1}{2}})m^{\frac{1}{2}}}.$$

By choosing $\beta = \frac{1-\lambda}{2\lambda}$ we know

$$\frac{(1+1/\beta)^{\frac{1}{2}}}{1 - (1+\beta)^{\frac{1}{2}}\lambda^{\frac{1}{2}}} = \frac{\left(1 + \frac{1}{\frac{1-\lambda}{2\lambda}}\right)^{\frac{1}{2}}}{1 - (1 + \frac{1-\lambda}{2\lambda})^{\frac{1}{2}}\lambda^{\frac{1}{2}}} = \frac{(\frac{1+\lambda}{1-\lambda})^{\frac{1}{2}}}{1 - (\frac{1+\lambda}{2})^{\frac{1}{2}}} = \frac{\sqrt{2}(1+\lambda)^{\frac{1}{2}}}{(1-\lambda)^{\frac{1}{2}}(\sqrt{2} - (1+\lambda)^{\frac{1}{2}})}$$

$$= \frac{\sqrt{2}(1+\lambda)^{\frac{1}{2}}(\sqrt{2} + (1+\lambda)^{\frac{1}{2}})}{(1-\lambda)^{\frac{1}{2}}(2 - (1+\lambda))} \lesssim \frac{1}{(1-\lambda)^{\frac{3}{2}}}.$$

Therefore,

$$\mathbb{E}[R(A(S))] - R(\theta^*) \lesssim \frac{G^2\eta T}{mn} + \frac{G^2\eta T^{\frac{1}{2}}}{1-\lambda} + \frac{\|\theta^*\|_2^2}{\eta T} + G^2\eta + \frac{G^2\eta}{(1-\lambda)^{\frac{3}{2}}m^{\frac{1}{2}}}.$$

The proof is completed. $\qquad\square$

