# OpenReview forum: "Stability and Generalization Analysis of Decentralized SGD: Sharper Bounds Beyond Lipschitzness and Smoothness"
_ICML.cc/2025/Conference — ICML 2025 poster_

### Official Review · Reviewer_wK2k · 2025-03-10

**Overall Recommendation:** 5

**Summary:**

This work establishes sharper stability and generalization bounds for decentralized SGD (D-SGD) under weaker assumptions. The analysis primarily builds on the on-average model stability, with a key innovation lying in the novel decomposition of neighboring consensus errors in decentralized settings.

**Claims And Evidence:**

Yes

**Essential References Not Discussed:**

Not to my knowledge.

**Experimental Designs Or Analyses:**

This paper does not contain experiments.

**Methods And Evaluation Criteria:**

Yes

**Other Comments Or Suggestions:**

See the questions.

**Other Strengths And Weaknesses:**

Strengths
* For smooth convex problems, this paper analyzes the convergence and generalization of D-SGD without relying on the Lipschitzness assumption.

* The paper investigates the generalization performance of D-SGD in non-smooth problems (satisfying the Holder continuous gradients condition).
* The analysis utilizes the co-coercivity property of the gradient for a novel decomposition of the neighboring consensus error.

Weaknesses
* The theoretical analysis in this paper is limited to the convex problem.

**Questions For Authors:**

1. ~~Can the analysis in this paper be extended to non-convex problems, e.g., training deep neural networks, and does the neighboring consensus error behave differently in this case?~~

2. ~~In line 22, the authors say that this paper develops optimal generalization bounds, how to show that the results in the paper are optimal?~~

**Relation To Broader Scientific Literature:**

Prior work was constrained by strict assumptions (Richards et al., 2020; Sun et al., 2021; Zhu et al., 2022; Le Bars et al. 2024); this work relaxes the assumptions and yields improved generalization results.

**Theoretical Claims:**

Yes, I have checked the theoretical results in the main text.

---

> ### Author Rebuttal · Authors · 2025-03-29
>
> We thank you for taking the time to review our paper and greatly appreciate your valuable feedback.
>
> **Q1**: The theoretical analysis in this paper is limited to the convex problem. Can the analysis in this paper be extended to non-convex problems, e.g., training deep neural networks, and does the neighboring consensus error behave differently in this case?
>
> **A**: Thanks for the suggestion. We agree that generalization analysis for nonconvex problems is interesting for understanding the practical performance of D-SGD. We will explore this direction in our future studies. For example, it is interesting to study the stability and generalization analysis of decentralized algorithms for training overparameterized neural networks, where we can exploit some weak-convexity [2, 3] and self-bounding weak-convexity [4, 5] to develop meaningful stability bounds.
>
> **Q2**: In line 22, the authors say that this paper develops optimal generalization bounds, how to show that the results in the paper are optimal?
>
> **A**: The minimax statistical error for learning with a convex and smooth function is $O(1/\sqrt{n})$ (e.g., Theorem 7 in [1]), where $n$ is the sample size. As we have $mn$ training examples in total, the minimax optimal error bound for the decentralized setting is $O(1/\sqrt{mn})$. Since we achieve excess risk bounds of order $O(1/\sqrt{mn})$ (Remark 4.9), we derive the minimax optimal risk bounds. We will clarify this in the revision.
>
> [1] Stability and convergence trade-off of iterative optimization algorithms. arXiv preprint, 2018.
>
> [2] Stability & generalisation of gradient descent for shallow neural networks without the neural tangent kernel. NeurIPS, 2021.
>
> [3] Generalization guarantees of gradient descent for shallow neural networks. Neural Computation, 2024.
>
> [4] Sharper guarantees for learning neural network classifiers with gradient methods. arXiv preprint, 2024.
>
> [5] On the optimization and generalization of multi-head attention. TMLR, 2024.

---

### Official Review · Reviewer_PhLr · 2025-03-12

**Overall Recommendation:** 4

**Summary:**

This work studies decentralized stochastic gradient descent where a network of agents collaborate to minimize an aggregate of local cost functions privately available to each agent. It focuses on the generalization analysis, which is different from the convergence rate analysis. It improves the generalization analysis compared to previous works under more general settings such as removing the Lipschitzness assumption and also considering nonsmooth settings. It also provides optimal generalization bounds.

**Claims And Evidence:**

Yes. All proofs are provided.

**Essential References Not Discussed:**

No. The work discusses all relevant works.

**Experimental Designs Or Analyses:**

There are no numerical experiments.

**Methods And Evaluation Criteria:**

No simulations are provided.

**Other Comments Or Suggestions:**

I suggest clearly discussing the novelty in this work and how you were able to provide tighter bounds with less restrictive assumption. A clear explanation of the theoretical novelty would be useful.

It would be useful to provide experimental results backing the theoretical findings.

**Other Strengths And Weaknesses:**

The paper has sufficient novelty but lacks clarity in presenting the problem and results. Notation is quite confusing, and clearer discussion should be provided regarding the main results.

Another weakness is that the analysis applies to convex setting.

**Questions For Authors:**

Can you please discuss Table 1 in details and clearly explain how your result is tighter. The bounds involve the stepsize and it is not clear to me how you result is tighter than Taheri and Thrampoulidis (2023). Are your results comparable when considering deterministic gradient?

**Relation To Broader Scientific Literature:**

The contributions are novel as the theoretical results give tighter bounds with less restrictive assumptions.

**Theoretical Claims:**

I did not check the proofs. The results seem reasonable. However, the writing and presentation of the results needs working on.

---

> ### Author Rebuttal · Authors · 2025-03-29
>
> We thank you for taking the time to review our paper and greatly appreciate your valuable feedback.
>
> **Q1**: Lack of clarity in presenting the problem and results. Notation is quite confusing.
>
> **A**: Thanks for your valuable feedback. We will reorganize the problem statement more clearly, systematically sort out the notations and present our results in a more readable way.
>
> **Q2**: The analysis applies to convex setting.
>
> **A**: Thanks for the suggestion. We agree that generalization analysis for nonconvex problems is interesting for understanding the practical performance of decentralized SGD. We will explore this direction in our future work. For example, it is interesting to study the stability and generalization analysis of decentralized algorithms for training overparameterized neural networks, where we can exploit some weak-convexity [2, 3] and self-bounding weak convexity [4, 5] to develop meaningful stability bounds.
>
> **Q3**: Explain novelty and how you provided tighter bounds with less assumption.
>
> **A**: We highlight our novelty here. First, we remove the Lipschitzness assumptions used in [1, 6] and replace the uniform gradient bound by function value via the self-bounding property, i.e., $\lVert \ell(\theta, z) \rVert_2^2 \leq 2 L \ell(\theta, z)$. In this way, we build stability bounds involving training errors, which shows the benefit of optimization in stability and implies fast rates under low noise conditions. Second, instead of decomposing the neighboring consensus error $\mathbb{E}[\lVert \bar{\theta}^{(t)}-\theta_k^{(t)}-\bar{\theta}^{(t, i j)}+\theta_k^{(t, i j)}\rVert_2^2]$ into two consensus errors as in [6, 7], we show that it can be offset using the co-coercivity of functions, which improves the stability analysis.
>
> **Q4**: Discuss Table 1 in detail and explain how your result is tighter.
>
> **A**: Our results improve the discussions in [1, 6] by removing the Lipschitzness assumptions and involving training errors, which implies faster rates under low noise conditions. Our results improve the results in [6] and [8] by removing the term $G\eta T/(1-\lambda)$ and $\lambda$ respectively, which do not involve $m$ and $n$. Our results improve [9] by removing both the bounded variance assumption and the term $C_W$. Furthermore, our stability analysis implies fast rates under a low noise condition. As a comparison, the stability bound in [9] involves $(\sigma\eta T+GT\eta C_W)/(mn)$, which does not imply fast rates in the low noise case. The stability bound in [7] involves a dominant factor $\frac{\eta^2\sqrt{T}}{1-\lambda}$ (if $\eta\gtrsim \frac{1-\lambda}{mn}$ and we ignore $L$ for brevity), which is replaced by $\frac{\eta^{\frac{3}{2}}}{\sqrt{m n}(1-\lambda)}+\frac{\eta}{m\sqrt{n}}$ in our bound. If $\eta \gtrsim \frac{1}{mnT}$ and $\eta\gtrsim \frac{1-\lambda}{\sqrt{Tn}m}$, then we have $\frac{\eta^2\sqrt{T}}{1-\lambda}\gtrsim \frac{\eta^{\frac{3}{2}}}{\sqrt{m n}(1-\lambda)}+\frac{\eta}{m\sqrt{n}}$ and our stability bound is better. Our analysis suggests $\eta\asymp 1/\sqrt{mn}$ in Remark 4.9, and in this case we have $\frac{\eta^2\sqrt{T}}{1-\lambda}\gg \frac{\eta^{\frac{3}{2}}}{\sqrt{m n}(1-\lambda)}+\frac{\eta}{m\sqrt{n}}$.
>
> **Q5**: How your result is tighter than [7]? Is it comparable when considering deterministic gradient?
>
> **A**: Yes, our technique can still imply better stability bounds when applied to decentralized gradient descent. Indeed, the discussions in [7] control the neighboring consensus errors by two consensus errors (Remark 4.3), which fail to use the property that neighboring consensus errors consider the difference of models produced on neighboring datasets. We introduce new techniques to control the neighboring consensus errors with an error decomposition and the coercivity of smooth functions. We can apply this technique to decentralized GD and improve the existing stability analysis.
>
> **Q6**: Providing experimental results is useful.
>
> **A**: Thanks for your suggestions. We agree that empirical analysis is helpful to validate our theoretical findings. We will leave it as future work.
>
> [1] Graph-dependent implicit regularisation for distributed stochastic subgradient descent. JMLR, 2020.
>
> [2] Stability & generalisation of gradient descent for shallow neural networks without the neural tangent kernel. NeurIPS, 2021.
>
> [3] Generalization guarantees of gradient descent for shallow neural networks. NeurIPS, 2024.
>
> [4] Sharper guarantees for learning neural network classifiers with gradient methods. arXiv preprint, 2024.
>
> [5] On the optimization and generalization of multi-head attention. TMLR, 2024.
>
> [6] Stability and generalization of decentralized stochastic gradient descent. AAAI, 2021.
>
> [7] On generalization of decentralized learning with separable data. AISTATS, 2023.
>
> [8] Topology-aware generalization of decentralized sgd. ICML, 2022.
>
> [9] Improved stability and generalization guarantees of the decentralized sgd algorithm. ICML, 2024.

---

### Official Review · Reviewer_5LWF · 2025-03-13

**Overall Recommendation:** 3

**Summary:**

This paper studies the stability of D-SGD, and presents new and sharper convergence bounds on the assumption that the functions are convex and L-smooth. The removal of the function Lipschitzness and the bounded variance assumptions highlight the novelty of the work. Theoretical analysis shows an improved stability bound compared to Zhu et al. under similar conditions. In addition, the theoretical results this paper offers also interpolated many known tight bounds under certain conditions.

**Claims And Evidence:**

The major contribution of this paper is theoretical, please see Theoretical Claims.

**Essential References Not Discussed:**

I am not very familiar with the related literature. There are no more related works that should be referenced to the best of my knowledge.

**Experimental Designs Or Analyses:**

There are no experiments provided in this paper.

The analysis seems straightforward and the authors were able to highlight the difference in analysis compared to prior works.

**Methods And Evaluation Criteria:**

There are no numerical evaluations in this paper. The author provided a good comparison of the stability bounds of related literature in Table 1.

**Other Comments Or Suggestions:**

misspelling of "related" on line 60.

**Other Strengths And Weaknesses:**

N/A

**Questions For Authors:**

N/A

**Relation To Broader Scientific Literature:**

The author provided a comparison of the stability bounds of related literature in Table 1. Although the comparison is far from complete, it was able to highlight the comparison of the current paper with some closely related papers.

**Theoretical Claims:**

The theoretical claims are mostly to be expected and seem to be either an extension of previous D-SGD analysis or from the stability/generalization of SGD in the centralized setting.

The authors were able to highlight the novelty of this work in Remark 4.3. However, I have some questions on the inequality shown on the left hand side of line 226, the consensus error is upper bounded by a norm of difference between $\nabla l$ and $l$. This seems to be a typo since $l$ denotes a scalar loss function.

---

> ### Author Rebuttal · Authors · 2025-03-29
>
> We thank you for taking the time to review our paper and greatly appreciate your valuable feedback.
>
> **Q1**: The theoretical claims are mostly to be expected and seem to be either an extension of previous D-SGD analysis or from the stability/generalization of SGD in the centralized setting.
>
> **A**: We highlight our novelty as follows. As compared to SGD in the centralized setting, the analysis of decentralized SGD is more challenging as this introduces the neighboring consensus error $\mathbb{E}[\\|\bar{\theta}^{(t)}-\theta_k^{(t)}-\bar{\theta}^{(t,i,j)}+\theta_k^{(t,ij)}\\|^2]$. The existing analysis of neighboring consensus error is a bit crude since it directly decompose it into two consensus errors $\mathbb{E}[\\|\bar{\theta}^{(t)}-\theta_k^{(t)}\\|^2]$ and $\mathbb{E}[\\|\bar{\theta}^{(t,i,j)}-\theta_k^{(t,ij)}\\|^2]$ [1, 2], which ignores the important property that $\theta_k^{(t)}$ and $\theta_k^{(t,ij)}$ are produced based on two neighboring datasets and should be close. A novelty of our analysis is to show that the neighboring consensus error can be offset by using the co-coercivity of a gradient map, which is achieved via a new error decomposition. In this way, we improve the stability analysis of decentralized algorithms [2,3]. Please see Remark 4.3 for the novelty of our analysis. Our stability analysis also improves the existing analysis by removing the Lipschitzness condition [1, 3], removing the Gaussian weight difference assumption [4], and removing the bounded variance assumption [5]. We also remove some terms  $G\eta T/(1-\lambda)$ and $\lambda$ in [3] and [4], respectively. Finally, our analysis shows the benefit of optimization in improving the stability, and gives the first fast rates of order $1/(mn)$ for decentralized SGD under a low-noise condition.
>
> **Q2**: However, I have some questions on the inequality shown on the left hand side of line 226, the consensus error is upper bounded by a norm of difference between $\nabla l$ and $l$. This seems to be a typo since $l$ denotes a scalar loss function.
>
> **A**: Thanks for indicating it. This should be the difference of two $\nabla$. We will revise these typos in the revision.
>
> **Q3**: Misspelling of "related" on line 60.
>
> **A**: Thanks for indicating it. We will modify it in the revision.
>
> [1] Graph-dependent implicit regularisation for distributed stochastic subgradient descent. JMLR, 2020.
>
> [2] On generalization of decentralized learning with separable data. AISTATS, 2023.
>
> [3] Stability and generalization of decentralized stochastic gradient descent. AAAI, 2021.
>
> [4] Topology-aware generalization of decentralized sgd. ICML, 2022.
>
> [5] Improved stability and generalization guarantees of the decentralized sgd algorithm. ICML, 2024.

---

> > ### Comment · Reviewer_5LWF · 2025-04-02
> >
> > I appreciate the authors' clarification on the novelty within the analysis. The finer-grained analysis with consensus contributed to the tighter bounds as well as the stability analysis. However, I am not entirely sure about the impact of this submission. I have modified my review accordingly.

---

### Official Review · Reviewer_f9L5 · 2025-03-16

**Overall Recommendation:** 3

**Summary:**

This paper presents the generalization and excess risk analysis for decentralized stochastic gradient descent (D-SGD) on smooth (including Hölder continuous, which generalizes smoothness) and convex problems. The key contribution is the removal of the standard Lipschitzness assumption in the analysis. The authors derive generalization bounds under the relaxed condition and compare them with existing results. The paper is well-written, with clear notations and explicit order analysis of the derived bounds.

## update after rebuttal
No remaining concerns.

**Claims And Evidence:**

The main claims of the paper are well-supported by theoretical derivations. The generalization bounds for D-SGD under non-Lipschitz assumptions are derived and compared with prior work.

**Essential References Not Discussed:**

No essential references not discussed.

**Experimental Designs Or Analyses:**

No experiments provided.

**Methods And Evaluation Criteria:**

The comparison with existing results is reasonable. No experiments provided.

**Other Comments Or Suggestions:**

Theorem 4.1 imposes a G-Lipschitz condition and establishes the bound
$\frac{G^2 \eta^2}{m n} \left(\frac{T}{m} + \frac{T^2}{m n}\right)$.
The authors state that this matches the serial case
$\frac{G^2 \eta^2}{n} \left(T + \frac{T^2}{n}\right)$.
Notably, this bound implies that increasing the number of agents/nodes ($m$) enhances generalization, as the first term in the parentheses decreases with larger $m$. The authors could provide a more detailed explanation and analysis of this effect.

The authors could consider mentioning "non-Lipschitz" in the title to better highlight the key contribution of the paper.

In Theorem 4.1 (Stability bound), there is an extra space after "Stability bound."

**Other Strengths And Weaknesses:**

**Weaknesses**:
The study focuses on **convex problems**, which provides valuable theoretical insights. However, exploring nonconvex settings could further enhance the practical relevance of the results. Additionally, the paper only analyzes the generalization of the average iterate but does not study local iterates.

**Questions For Authors:**

In Theorem 4.1, the assumption
   $\left(\frac{2(1+\lambda) L \eta_t}{(1-\lambda)^2} + 2\right) \eta_t - \frac{1}{L} \leq 0$
   is not carefully analyzed. The authors state that it holds when
   $\eta_t \lesssim \frac{(1-\lambda)}{L}$.
   However, in a ring topology with a large number of nodes (where $\lambda$ is close to 1), this assumption seems difficult to satisfy. This contradicts empirical results, such as Figure 1 (right) in [1], where experiments suggest that larger node sizes in a ring topology actually allow for a larger learning rate.

[1] Beyond spectral gap: The role of the topology in decentralized learning. NeurIPS, 2022.

**Relation To Broader Scientific Literature:**

The discussion of related work the comparison with prior literature is through.

**Theoretical Claims:**

The reviewer checked the theoretical claims but did not check the proof carefully.

---

> ### Author Rebuttal · Authors · 2025-03-29
>
> We thank you for taking the time to review our paper and greatly appreciate your valuable feedback.
>
> **Q1**: Exploring nonconvex settings could further enhance the practical relevance of the results.
>
> **A**: Thanks for the suggestion. We agree that generalization analysis for nonconvex problems is interesting for understanding the practical performance of decentralized SGD. We will explore this direction in our future studies. For example, it is interesting to study the stability and generalization analysis of decentralized algorithms for training overparameterized neural networks, where we can exploit some weak-convexity [2, 3] and self-bounding weak-convexity [4, 5] to develop meaningful stability bounds.
>
> **Q2**: The paper only analyzes the generalization of the average iterate but does not study local iterates.
>
> **A**:  We consider the average of iterates since most of the existing convergence analyses focus on the average of iterates [6, 7, 8]. Then, we can combine our generalization analysis and existing convergence analyses to derive excess risk bounds. We note that the recent work [9] gave interesting discussions on the stability analysis for local iterates. Their studies consider the $\ell_1$-version of on-average model stability. It is interesting to apply our techniques to study the $\ell_2$-version of on-average model stability for local iterates. We will consider this in our future studies.
>
> **Q3**: Theorem 4.1 imposes a G-Lipschitz condition and establishes the bound $\frac{G^2 \eta^2}{m n}\left(\frac{T}{m}+\frac{T^2}{m n}\right)$. The authors state that this matches the serial case $\frac{G^2 \eta^2}{n}\left(T+\frac{T^2}{n}\right)$. Notably, this bound implies that increasing the number of agents/nodes $(m)$ enhances generalization, as the first term in the parentheses decreases with larger $m$. The authors could explain more about this effect.
>
> **A**: Note that each agent has $n$ examples in our setting, and therefore we have $mn$ examples in total. Then, the effect of perturbing a single example diminishes as $m$ increases, which implies improved stability. We will add discussions to explain this more clearly.
>
> **Q4**: The authors could consider mentioning "non-Lipschitz" in the title to better highlight the key contribution of the paper. In Theorem 4.1 (Stability bound), there is an extra space after "Stability bound."
>
> **A**: Thanks for your valuable suggestions. We will add "non-Lipschitz" in the title and fix this formatting issue in the revision.
>
> **Q5**: In Theorem 4.1, the assumption $\left(\frac{2(1+\lambda) L \eta_t}{(1-\lambda)^2}+2\right) \eta_t-\frac{1}{L} \leq 0$ is not carefully analyzed. The authors state that it holds when $\eta_t \lesssim \frac{(1-\lambda)}{L}$. However, in a ring topology with a large number of nodes (where $\lambda$ is close to 1), this assumption seems difficult to satisfy. This contradicts empirical results, such as Figure 1 (right) in [1], where experiments suggest that larger node sizes in a ring topology actually allow for a larger learning rate.
>
> **A**: Thanks for your intuitive comment on the learning rate. We note that the assumption $\eta_t\lesssim (1-\lambda)/L$ is also used in both the existing convergence analysis [6] and stability analysis [8] of decentralized algorithms. Indeed, the estimation of consensus error often leads to a term $1/(1-\lambda)$, which becomes infinite if $\lambda$ is close to $1$. Furthermore, in Remark 9 we set $\eta_t\asymp 1/\sqrt{T}$ and $T\asymp mn$ to get risk bounds of order $1/\sqrt{mn}$. Then, the assumption $\eta_t\lesssim (1-\lambda)/L$ roughly becomes $1/\sqrt{mn}\lesssim (1-\lambda)/L$, which is a mild assumption if $n$ is large. While $1-\lambda$ becomes smaller as $m$ increases, our suggested step size $\eta_t\asymp 1/\sqrt{mn}$ also decreases with $m$. It is very interesting to further relax the requirement $\eta_t\lesssim (1-\lambda)/L$. We will leave this as an interesting question for further investigation.
>
> [1] Beyond spectral gap: The role of the topology in decentralized learning. NeurIPS, 2022.
>
> [2] Stability & generalisation of gradient descent for shallow neural networks without the neural tangent kernel. NeurIPS, 2021.
>
> [3] Generalization guarantees of gradient descent for shallow neural networks. Neural Computation, 2024.
>
> [4] Sharper guarantees for learning neural network classifiers with gradient methods. arXiv preprint, 2024.
>
> [5] On the optimization and generalization of multi-head attention. TMLR, 2024.
>
> [6] Graph-dependent implicit regularisation for distributed stochastic subgradient descent. JMLR, 2020.
>
> [7] Stability and generalization of decentralized stochastic gradient descent. AAAI, 2021.
>
> [8] On generalization of decentralized learning with separable data. AISTATS, 2023.
>
> [9] Improved stability and generalization guarantees of the decentralized sgd algorithm. ICML, 2024.

---

> > ### Comment · Reviewer_f9L5 · 2025-04-04
> >
> > The reviewer thanks the authors for their detailed response.
> >
> > In Theorem 4.1, the term
> > $\frac{G^2 \eta^2}{mn} ( \frac{T}{m} + \frac{T^2}{mn} )$
> > can also be written as
> > $\frac{G^2 \eta^2}{N} ( \frac{T}{m} + \frac{T^2}{N} )$
> > where $N = mn$. This  suggests that increasing $m$ actually helps stability and generalization.
> >
> > This differs from previous results, where improving generalization by increasing $m$ was typically attributed to a larger $N$, since $N = mn$. However, Theorem 4.1 indicates that stability can be improved by increasing $m$ even if $N$ is set fixed.
> >
> > This is a surprising result. Are there any related works that have discussed a similar theoretical observation? Could the authors elaborate more on this point?

---

> > > ### Author Response · Authors · 2025-04-05
> > >
> > > We thank the reviewer for the further clarifications of the comment. We think the underlying reason is that we consider an average of the iterates and the $\ell_2$ version of the on-average model stability. Please note the $\ell_2$ model stability is a second moment of a random variable,  which can be decomposed as a bias and a variance term. The average operator considered in this submission reduces the variance by a factor of m. This explains why there is a factor of $1/m$ in our stability bound (Thm 4.1 gives a bound on the square of the $\ell_2$ on-average model stability). As a comparison, the previous discussions either consider local iterates or the $\ell_1$ version of stability, which may not show the effect of variance reduction of decentralized SGD with m local machines.
> > >
> > > This recent stability analysis of minibatch/local SGD also shows a similar phenomenon [1]. It was shown there the variance decreases by a factor of batch size or local machines. Therefore, the square of the $\ell_2$ model stability improves by a factor of batch size or local machines.
> > >
> > > We will add more discussions on in the revision. Thanks again and please let us know if you have further comments.
> > >
> > > [1] Stability and Generalization for Minibatch SGD and Local SGD. arXiv 2023

---

### Decision · Program_Chairs · 2025-05-01

**Decision:**

Accept (poster)

**Comment:**

This paper presents a refined generalization and excess risk analysis for decentralized stochastic gradient descent (D-SGD) in convex settings, where the authors remove the standard Lipschitzness assumption and extend the analysis to include smooth and nonsmooth problems. A novel error decomposition of neighboring consensus errors is introduced, leading to tighter stability bounds and fast excess risk rates under low noise conditions.

All reviewers appreciate the paper's strong theoretical contributions and clear exposition. Therefore, I recommend acceptance.